

# Atmospheric Dust and Air Quality over large-cities and megacities of the World

Emmanouil Proestakis[1], Kyriakoula Papachristopoulou[2], Thanasis Georgiou[1,3], Sofia Eirini Chatoutsidou[4], Mihalis Lazaridis[4], Antonis Gkikas[5], Ilias Fountoulakis[5], Ioanna Tsikoudi[1,6], Manolis P. Petrakis[4], Vassilis Amiridis[1]

[1]Institute for Astronomy, Astrophysics, Space Applications and Remote Sensing, National Observatory of Athens, Athens, 15236, Greece.

[2]Physikalisch-Meteorologisches Observatorium Davos, World Radiation Center (PMOD/WRC), Davos 7260, Switzerland.

[3]School of Physics, Faculty of Sciences, Aristotle University of Thessaloniki, Thessaloniki, Greece.

[4]School of Chemical and Environmental Engineering, Technical University of Crete, Chania, 73100, Greece

[5]Research Centre for Atmospheric Physics and Climatology, Academy of Athens, Athens, Greece.

[6]Department of Physics, Section of Environmental Physics-Meteorology, National and Kapodistrian University of Athens, 15784, Greece.

Correspondence to: Emmanouil Proestakis (proestakis@noa.gr)

## Abstract

Urbanization is accelerating, with over 55% of the global population residing in urban areas as of 2018, projected to reach
60% by 2030. This growth intensifies environmental pressures, notably air quality degradation, which adversely affects human health and socioeconomic systems. Among air pollutants, atmospheric mineral dust plays a critical role in Earth's climate system and poses direct health risks. This study assesses the mass concentration of atmospheric dust within the planetary boundary layer (PBL)—the atmospheric region where most anthropogenic activity occurs—across global large cities and megacities (population >5 million). Using the European Space Agency's LIdar climatology of Vertical Aerosol Structure
(LIVAS) climate data record, which offers multiyear, four-dimensional Earth Observation-based dust mass concentration data, we analyze both fine-mode and coarse-mode dust components. Results show that current dust levels exceed World Health Organization (WHO) annual mean air quality guidelines (AQGs) for $PM_{2.5}$ and $PM_{10}$ in 49.4% and 87.7% of urban areas studied, respectively, exposing approximately 701.4 million people to hazardous dust concentrations. Regions particularly affected include the Middle East, Indian subcontinent, East Asia, North Africa, and the Sahel. Projections indicate a 21.7%
increase in the exposed population by 2030, totaling 856.5 million, despite a general decline in dust severity in over 70% of cities. Epidemiological models are employed to estimate associated health risks. This work provides essential insights for evidence-based air quality management and public health strategies, supporting mitigation efforts in the face of urban expansion and climate change.

## 1. Introduction

According to the United Nations (UN) Population Division of the Department of Economic and Social Affairs (UN, 2018a, 2019) more than 55% of the global population resided in urban areas in 2018. Driven by various factors, including urbanization, migration, and higher birth rates than death rates (Lerch, 2017) urban areas are projected to house an estimated 60% of the
world's population by 2030 and more than two-thirds (~68%) of the global population by mid-century. As urbanization accelerates and population density rises, degradation of air quality becomes an increasingly pressing issue. Accounting for this environmental pressure, the World Health Organization (WHO) has highlighted the severe impact of poor air quality on public



health, particularly in urban areas (Shairsingh et al., 2024). According to the WHO poor air quality is the second highest risk factor for noncommunicable diseases, with an estimated seven million premature deaths globally every year due to exposure

to both indoor and outdoor air pollution (https://www.who.int/news-room/fact-sheets/detail/ambient-(outdoor)-air-quality-and-health; last access: 13/04/2025).

Among the aerosol species resulting in degradation of air quality are mineral dust particles, especially over densely populated and heavily industrialized areas (Papachristopoulou et al., 2022; Proestakis et al., 2024). More specifically, atmospheric dust is recognized as one of the most important aerosol types, both in terms of mass and optical depth, and the dominant component

of atmospheric aerosol over large areas of the Earth (Gliß et al., 2021; Kok et al., 2017; 2021; 2023). Once suspended in the atmosphere, dust exerts a multifaceted and complex role in the Earth's climate system, while simultaneously posing considerable challenges to anthropogenic activities. More specifically, upon entering the atmosphere, dust particles are subject to aeolian transport, in many cases over distances of thousands of kilometres downwind (e.g. Prospero, 1999a, 1999b; Dey et al., 2004; Schepanski et al., 2009; Kanitz et al., 2014; Weinzierl et al., 2016; Marinou et al., 2017; Proestakis et al., 2018;

2024; Ramaswamy et al., 2018; Adebiyi and Kok, 2020; Aslanoğlu et al., 2022; Drakaki et al., 2022; Gkikas et al., 2022). While airborne, dust particles affect several atmospheric processes, spanning from short- (weather) to long- (climate) term temporal scales, via their interactions with the shortwave (SW) and longwave (LW) radiation. Dust aerosols serve as effective cloud condensation nuclei (CCN; Hatch et al., 2008) and/or ice-nucleating particles (INPs; DeMott et al., 2009). Atmospheric dust layers modify clouds' microphysical, macrophysical and optical properties, precipitation patterns, atmospheric stability,

cloud formation, lifetime, and coverage (Twomey, 1977; Albrecht, 1989; Rosenfeld et al., 2008). Dust is considered a significant parameter related to aviation safety (Papagiannopoulos et al., 2020; Nickovic et al., 2021; Ryder et al., 2024) while, by reducing the amount of SW radiation reaching the Earth's surface, dust layers affect solar energy production (Kosmopoulos et al., 2018; Masoom et al., 2021; Fountoulakis et al., 2021). Eventually, upon their removal from the atmosphere, through wet or dry deposition (Gao et al., 2003; Hand et al., 2004; Prospero et al., 2010; Mahowald et al., 2011; Van der Does et al., 2018;

2021; Proestakis et al., 2025), dust particles enrich with micro nutrients the marine and terrestrial ecosystems (Okin et al., 2004; Jickells et al., 2005; Li et al., 2018).

However, the multifaceted role of dust extends beyond its effects on biogeochemistry, the radiation budget, weather, and climate; it also poses a major threat to human health. To date, a key aspect governing the association between aerosols, air quality (AQ), and negative effects induced on human health is considered the amount of the airborne particulate matter (PM).

More specifically, according to the WHO air quality guidelines (WHO, 2021), PM is generally divided into $PM_{10}$ (coarse), $PM_{2.5}$ (fine) and UFP (ultra-fine) classes, referring to categories of airborne particles with aerodynamic diameter ≤10 μm, ≤2.5 μm, and ≤0.1 μm, respectively (Table 1).

Table 1: World Health Organization Air Quality Guidelines for $PM_{2.5}$ and $PM_{10}$.

| $PM_{2.5}$: | 5 μg m$^{-3}$ annual mean |
| | 15 μg m$^{-3}$ 24-hour mean |
| $PM_{10}$: | 15 μg m$^{-3}$ annual mean |
| | 45 μg m$^{-3}$ 24-hour mean |


Several epidemiological studies have revealed a strong connection between high concentrations of airborne dust and adverse health effects. In general, the health risk attributed to coarse-size mineral particles of the order of ~10 μm or larger is considered low, referring mainly to mild skin irritation and/or allergic responses, even under conditions of high dust concentrations and long-term exposure (Sandstrom, 2008; Pérez García-Pando et al., 2014). However, dust $PM_{2.5}$ particles, due to their small size,

can penetrate deeper into the lungs and alveoli (Martinelli et al., 2013), leading among others to allergic responses (Bousquet et al., 2003; Kellogg et al. 2004; Chang et al., 2006; Ezeamuzie et al., 2008; Smith et al. 2012; Watanabe et al., 2011), cardiovascular diseases (Kwon et al., 2002; Meng and Lu, 2007; Middleton et al., 2008; Prospero et al., 2008; Sandstrom and





Forsberg; 2008; Pérez et al., 2012; De Longueville et al., 2010; Martinelli et al., 2013; Goudie, 2014; Zhang et al., 2016;
Achakulwisut et al., 2018; Querol et al., 2019), respiratory diseases (Kwon et al., 2002; Wiggs et al., 2003; Chen et al., 2004;

Veranth et al., 2004; Park et al., 2005; Derbyshire, 2007; Meng and Lu, 2007; Cheng et al., 2008; Yoo et al., 2008; De
Longueville et al., 2010; 2013; Leski et al., 2011; Goudie, 2014; Katra et al., 2014; Mueller et al., 2017; Middleton, 2020),
related due to silica with lung cancer (Giannadaki et al., 2014; Steenland and Ward, 2014), even to Valley Fever and Meningitis
epidemic outbreaks in the Sahel during the Harmattan seasons (Tobías et al., 2011; Agier et al., 2013; Deroubaix et al., 2013;
Pérez García-Pando et al., 2014; Martiny and Chiapello, 2013; Carc et al., 2014; Ceccato et al., 2014; Goudie et al., 2014;

Jusot et al., 2017; Mueller et al., 2017; Nakazawa and Matsueda, 2017; Mazamay et al., 2020; Middleton et al., 2020; Woringer
et al., 2022).

Despite the increasing number of scientific studies indicating that airborne mineral dust constitutes a significant environmental
hazard and risk factor for human health, current knowledge on the dust health impacts is still characterized by large
uncertainties, primarily attributed to three key challenges. The first challenge arises from the reliance of many studies on Earth

Observation (EO) using passive remote sensing (RS) techniques aiming to investigate the dust load at regional and global
scales and to establish associations with dust-induced disorders on human health (i.e., De Longueville et al., 2010; Deroubaix
et al., 2013; Katra et al., 2014; Prospero et al., 2014; Querol et al., 2019). However, passive RS techniques mainly retrieve and
provide column-integrated aerosol properties and not acquire vertically resolved aerosol profiles, thereby limiting their
capability to accurately quantify the aerosol load within the planetary boundary layer (PBL) (McGrath-Spangler and Denning,

2013; Luo et al., 2014) where the majority of anthropogenic activities take place. The second challenge pertains to the
dependence of dust-related health disorders primarily on the dust PSD and, to a lesser extent, on total dust mass. large scale
intensive experimental campaigns employing in-situ instrumentation report on dust particle size distribution (PSD) ranging
from 0.1 (UFP upper limit) to more than 100 µm in diameter (Weinzierl et al., 2017; Ryder et al., 2018; Van der Does et al.,
2018), with the fine-mode fraction of dust particles been the particularly hazardous component as can penetrate deep into the

lungs and alveoli, increasing morbidity and mortality rates. However, limited research has focused on the specific relationship
between the inhalable fraction of dust PSD and associated health effects at regional and global scales using EO data. The third
challenge arises from the use of atmospheric aerosol models which are extensively employed to provide four-dimensional
spatiotemporal insight into dust emission, transport, deposition, and vertical structure (Textor et al., 2006; Astitha et al., 2012;
Randles et al., 2017; Konsta et al., 2018; Inness et al., 2019). Models often employ static land cover classifications to classify

arid and semi-arid regions as dust sources (Ginoux et al., 2001). However, implementation of static emission inventories
introduces substantial uncertainties, particularly by neglecting anthropogenic dust emissions (Ginoux et al., 2012; Huang et
al., 2015; Chen et al., 2019), leading to significant underestimations of the human health risks associated with dust exposure
especially over densely populated and highly industrialized urban areas of the Earth (Proestakis et al., 2018; 2024;
Papachristopoulou et al., 2022).

According to the Agenda for Sustainable Development Goals (SDGs) 2030, and more specifically SDG11, the UN are
committed to make cities and human settlements inclusive, safe, resilient, and sustainable, addressing in a holistic approach
environmental, economic, and social urban development (Weiland et al., 2021). As emphasized in the UN "World Urbanization
Prospects (WUP): The 2018 Revision", urban sustainable development depends increasingly on unravelling and understanding
the key trends in urbanization, population, and air quality likely to unfold over the coming years (UN, 2019). Urban sustainable

development critically depends on high quality demographic and air quality data translated into information on future
projection for evidence-based policy making, to adhere to the UN "urban agglomeration" concept of cities essential for
managing urbanization, mitigating air pollution, and safeguarding public health in rapidly growing urban areas worldwide.

The present study aims to address the following scientific questions: To what extent the submicrometer (fine) and
supermicrometer (coarse) modes of mineral dust entrained into the atmosphere within the PBL has changed over the highly-

industrialized and densely-populated areas/Megacities of the world over the last two decades? Is it feasible to identify



statistically significant trends? Which areas experience fine-mode and coarse-mode dust mass concentrations within the PBL exceeding WHO air quality guidelines and over which areas it is foreseen the dust modes to exceed WHO air quality guidelines in the near-future?

The study is based on the well-established European Space Agency (ESA) "LIdar climatology of Vertical Aerosol Structure" (LIVAS) climate data record (Amiridis et al., 2015), and more specifically on the Earth Observation (EO) -based products of the total atmospheric dust (Amiridis et al., 2013; Marinou et al., 2017; Proestakis et al., 2018) and the fine-mode and coarse-mode components of the total atmospheric dust, described in detail in Proestakis et al. (2024). In a nutshell, the ESA-LIVAS atmospheric dust products are enabled through a conceptual combination of (i) the outcomes of extensive laboratory experiments reporting on the distinct light-depolarizing properties of the fine-mode and coarse-mode components of atmospheric dust (Sakai et al., 2010; Järvinen et al., 2016), (ii) a well-established in the framework of the European Aerosol Research Lidar Network (EARLINET; http://www.earlinet.org/; last access: 13/04/2025) activities (Pappalardo et al., 2014) sophisticated technique, namely the two-step POlarization LIdar PHOtometer Networking (two-step POLIPHON) algorithm (Mamouri and Ansmann, 2014, 2017) allowing decoupling of the two modes, and (iii) extensive implementation of near-global atmospheric aerosol profiling observations acquired by the Cloud-Aerosol Lidar with Orthogonal Polarization (CALIOP) (Hunt et al., 2009) satellite-based system (Winker et al., 2010). The ESA-LIVAS atmospheric fine-mode and coarse-mode dust climate data record (CDR) presents several innovative features, such as (i) full global coverage, enabling the assessment of air quality across major cities and megacities, (ii) EO-based dust products established entirely on experimentally-parameterized and observational-based conditions, (iii) a four-dimensional (horizontal, vertical, and temporal) representation of atmospheric dust conditions in terms of fine-mode and coarse-mode mass concentrations, with a particular focus on the lower troposphere (PBL) where low air quality strongly influences human health, and (iv) sixteen full years of temporal coverage (12/2006–11/2022), leveraging the latest CALIPSO data collection to facilitate time-series and trend analyses for projecting future PBL dust load variations..

Overall, the study through the analysis of the ESA-LIVAS atmospheric fine-mode and coarse-mode dust CDR aims to quantify the intensity of PBL dust PM with reference on the WHO air quality guidelines and determine over which areas the dust modes are expected to exceed WHO air quality guidelines in the near-future. By doing so, the study aims to provide a stepping-stone to enhance our fundamental scientific understanding of the complex role of inhalable dust particles to the induced disorders of human health under, allowing new pathways to support development of adaptation and mitigation strategies to preserve human health under the ongoing climate change.

The paper is organized as follows. Section 2 provides a description of the datasets used (Sect. 2.1) and an overview of the applied methodology (Sect. 2.2), both designed to address the scientific questions of the study. In Sect. 3, a consistency assessment of increasing and decreasing trends in the PBL dust aerosol load is conducted using long-term AERONET observations as reference datasets. Section 4 examines the extent to which the fine-mode and coarse-mode components of dust aerosols, entrained into the atmosphere and confined within the PBL, have changed over large cities and megacities worldwide over the past two decades. Section 5 provides insights into the anticipated changes in atmospheric dust in the third decade of the 21st century, based on quantified fine-mode and coarse-mode PBL dust mass concentrations and the identified trends during the EO-based reference period. Section 6 translates the computed dust mass concentrations into environmental risk factors for human health disorders associated with atmospheric dust exposure. Finally, Section 7 summarizes the study and presents the main conclusions.

## 2 Datasets and methodology

### 2.1 Datasets



### 2.1.1 Atmospheric Dust


Our study is based on the four-dimensional, multiyear, and near-global climate data record (CDR) of the fine-mode and coarse-mode components of atmospheric dust (Proestakis et al., 2024). A brief discussion and overview of the CDR is hereinafter provided on the basis of an indicative CALIPSO nighttime overpass on the 16th of April 2018-04-16 between 18:29:57 and 18:35:30 UTC in the proximity of the Beijing megacity area (Fig.1-left). During the overpass, according to the NOAA Hybrid

Single-Particle Lagrangian Integrated Trajectory (HYSPLIT) model (Stein et. al., 2015), air masses in the broader Beijing area where a mixture of emissions residing close to the surface within the PBL, probably attributed to the extensive anthropogenic and industrial activity, and higher altitude dust layers advected from the vast Gobi Desert extending Northwest of the Beijing megacity area mainly across Mongolia (Fig.1-right).

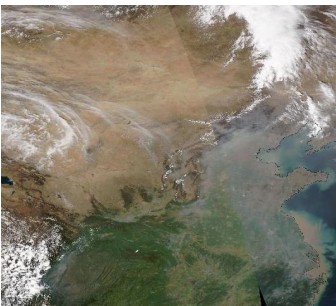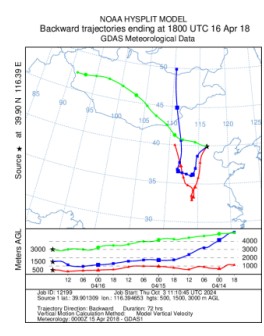


Figure 1: Beijing megacity area during the CALIPSO nighttime overpass on the 16th of April 2018-04-16 between 18:29:57 and 18:35:30 UTC (Source: NASA Worldview: https://worldview.earthdata.nasa.gov/; last access: 13/04/2025) (a) and NOAA HYSPLIT (https://www.ready.noaa.gov/HYSPLIT.php; last access: 13/04/2025) model back-trajectories at 500 m (red line), 1500 m (blue line), and 3000 m (green line) above ground level.


As described in detail in Proestakis et al. (2024), the CDR is established on the basis of Cloud-Aerosol Lidar with Orthogonal Polarization (CALIOP) (Hunt et al., 2009) quality-assured (Winker et al., 2013; Tackett et al., 2018) profiles of backscatter coefficient at 532 nm (Fig.2b) and particulate depolarization ratio at 532 nm along the orbit-track of the satellite Cloud-Aerosol Lidar and Infrared Pathfinder Satellite Observations (CALIPSO) (Winker et al., 2010) (Fig.2a). Decoupling of the atmospheric

dust load into the two modes, a submicrometer and a supermicrometer in terms of diameter, is performed on the basis of the two-step POlarization LIdar PHOtometer Networking (two-step POLIPHON) technique (Mamouri and Ansmann, 2014, 2017), developed within the framework of the European Aerosol Research Lidar Network (EARLINET; http://www.earlinet.org/; last access: 13/04/2025) activities (Pappalardo et al., 2014), applied on CALIOP Level 2 (L2) Version 4.51 (V.51) products and implementing CALIPSO layer detection (Vaughan et al., 2009), cloud–aerosol discrimination (Liu et al., 2009, 2019; Zeng et

al., 2019) (Fig.2c), and aerosol subtype classification (Omar et al., 2009; Kim et al., 2018; Kar et al., 2019) (Fig.2d). The scientific cornerstones of both the two-step POLIPHON technique and the dust fine- and coarse- mode EO-based CDR are the extensive chamber laboratory experiments performed by Sakai et al. (2010) and Järvinen et al. (2016) reporting on distinct light-depolarizing properties of the two dust modes, essentially characterizing dust size distribution (Weinzierl et al., 2016; Ryder et al., 2018). Adapted and applied on CALIOP quality-assured optical products, the two-step POLIPHON technique

yields quality-assured backscatter coefficient at 532 nm profiles of the atmospheric fine-mode (Fig.2g), coarse-mode (Fig.2f), and total dust (Fig.2e) along the CALIPSO orbit-path (Fig.2a). Accordingly, conversion of the backscatter coefficient at 532 nm profiles into extinction coefficient at 532 nm profiles (Fig.2h/i/j) is performed on the basis of suitable geographically-



dependent backscatter-to-extinction conversion factors (i.e. lidar ratio at 532 nm), as provided by the database of lidar-derived aerosol intensive optical properties DeLiAn (Floutsi et al., 2023) and source-attribution classification adopted by the well-established European Space Agency (ESA) database LIdar climatology of Vertical Aerosol Structure (LIVAS) (Amiridis et al., 2013; Marinou et al., 2017; Proestakis et al., 2018). As a next step, the obtained quality-assured extinction coefficient at 532 nm profiles of the atmospheric coarse-mode and total dust are converted into coarse-mode and total dust mass concentration (MC) profiles (Fig.2k/l) on the basis of representative dust extinction-to-volume concentration conversion factors provided by Ansmann et al. (2019) and assumed dust particle density ($\rho_d$) of 2.6 g cm$^{-3}$ (Ansmann et al., 2012). Finally, the atmospheric fine-mode dust mass concentration profiles (Fig.2m) are computed as the residual between the total dust mass concentration and the coarse-mode dust mass concentration profiles along the CALIPSO orbit track.



Figure 2: Illustration of the established methodology towards extracting the atmospheric dust component from the total aerosol
load and accordingly separating the fine-mode and coarse-mode components of atmospheric dust, provided on the basis of an
indicative CALIPSO-Beijing nighttime overpass on the 16th of April 2018-04-16 between 18:29:57 and 18:35:30 UTC (Fig.
2a). On the second row, the quality-assured total backscatter coefficient at 532 nm (Fig.2b), Feature Type (Fig.2c), and Aerosol
Subtype (Fig.2d) are provided. The third row provides the final products of total dust (Fig.2e), coarse-mode dust (Fig.2f), and
fine-mode dust (Fig.2g) in terms of quality-assured profiles of backscatter coefficient at 532 nm. On the fourth row, the total
dust (Fig.2h), coarse-mode dust (Fig.2i), and fine-mode dust (Fig.2j) components of the total aerosol load in terms of quality-
assured extinction coefficient at 532 nm profiles are shown. The fifth row provides the total dust (Fig.2k), coarse-mode dust
(Fig.2l), and fine-mode dust (Fig.2m) components of the total aerosol load in terms of profiles of mass concentration ($\mu g/m^3$).
Layer background: © Google Maps.

The final products consist of the fine-mode and coarse-mode of atmospheric dust, quality-assured profiles of backscatter
coefficient at 532 nm and extinction coefficient at 532 nm, and mass concentration for each component, with the original L2
horizontal and vertical resolution of CALIOP, of 5 km and 60 m respectively, along the CALIPSO orbit path (Winker et al.,
2009). Accordingly, the EO-based dust fine-mode and coarse-mode CDR is established in monthly-mean and seasonal-mean
profiles of backscatter coefficient at 532 nm and extinction coefficient at 532 nm. These profiles are established in 1° × 1°
spatial resolution and 60 m vertical resolution, cover the latitudinal band between 70° S and 70° N and the entire lifetime of
the CALIPSO mission. Moreover, vertical integration of the total aerosol, total dust, coarse-mode dust, and fine-mode dust
mean quality-assured extinction coefficient at 532 nm profiles (Fig.2h/i/j) with respect to height -between the top of atmosphere
(TOA) and the mean surface elevation- yield columnar AOD (Fig.3a), DOD (Fig.3b), DOD$_{\text{fine-mode}}$ (Fig.3c) and DOD$_{\text{coarse-mode}}$
(Fig.3d) at 532 nm values, facilitating implementation of the CDR climatological studies (Amiridis et al., 2013; Marinou et
al., 2017; Aslanoğlu et al., 2022; Proestakis et al., 2018, 2024).

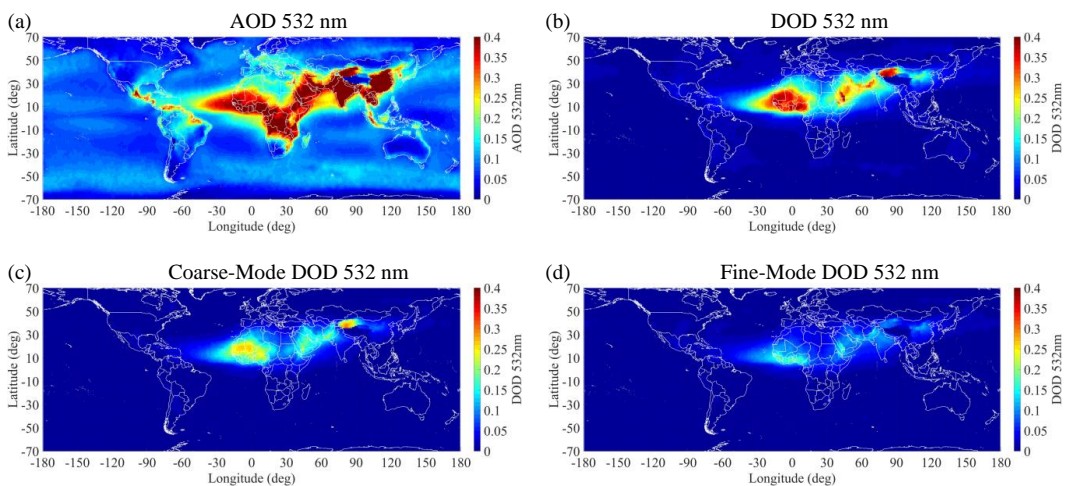

Figure 3: CALIOP-based AOD at 532 nm (Fig.3a), ESA-LIVAS DOD at 532 nm (Fig.3b), and coarse-mode DOD at 532 nm
(Fig.3c) and fine-mode DOD at 532 nm (Fig.3d), gridded at 1°×1° spatial resolution and for the temporal period extending
between June 2006 and July 2023.

In Proestakis et al. (2024), the four-dimensional, multiyear, and near-global CDR of the fine-mode and coarse-mode
components of atmospheric dust expressed in terms of columnar integrated extinction coefficient at 532 nm (DOD at 532 nm)



was evaluated on the basis of AErosol RObotic NETwork (AERONET; https://aeronet.gsfc.nasa.gov/; last access: 13/04/2025)

(Holben et al., 1998) fine-mode and coarse-mode aerosol optical thickness (AOT) at 500 nm derived during atmospheric conditions characterized by dust presence via the Spectral Deconvolution Algorithm (SDA; Eck et al., 1999; O'Neill et al., 2001a, b, 2003) and interpolated to 532 nm. The evaluation analysis between the EO-based $DOD_{fine}$ at 532 nm products and AERONET $AOT_{fine}$ at 532 nm retrievals revealed a fairly good agreement between the two datasets for the fine-mode class, with a slope of 0.652, an offset of 0.018, and a Pearson's correlation coefficient of 0.692. However, an increasing

underestimation of fine-mode dust was observed with higher dust aerosol concentrations (Fig. 6a in Proestakis et al., 2024). For the coarse-mode category, the evaluation demonstrated a substantially better agreement between the EO-based CDR and AERONET retrievals, with a slope of 0.779, an intercept of −0.002, and a Pearson's correlation coefficient of 0.916 (Fig. 6b in Proestakis et al., 2024). The EO-based fine- and coarse- mode atmospheric dust CDR was further verified in terms of profiles of mass concentration against airborne in situ dust aerosol size distributions (PSD) acquired by the United Kingdom (UK)

Facility for Airborne Atmospheric Measurements (FAAM) BAe-146 research aircraft. These reference measurements were obtained in the framework of the "AERosol properties – Dust" (AER–D) experiment conducted in the tropical Atlantic Ocean, in the region extending between Cabo Verde and the Canary Islands, in August 2015 (Marenco et al., 2018; Ryder et al., 2018). With respect to the coarse-mode category, the validation intercomparison revealed agreement within 10 % between the EO-based coarse-mode dust mass concentration product and the reference in situ supermicrometer mass concentration

measurements (Fig.9c in Proestakis et al., 2024). With respect to the fine-mode category, the validation intercomparison activities revealed a fairly good agreement between the two datasets, although a noticeable underestimation was apparent (Fig.9d in Proestakis et al., 2024). Overall, the quality assessment activities corroborated on the capacity and effectiveness of the lidar-based techniques established with the overarching objective to decouple the fine-mode and coarse-mode components from the total atmospheric dust load (Shimizu et al., 2004; Mamouri and Ansmann, 2014, 2017; Tesche et al., 2009) and

revealed the high quality of the established EO-based products of the submicrometer and the supermicrometer dust components in terms of extinction coefficient at 532 nm and mass concentration profiles and DODs at 532 nm.

In addition, the four-dimensional, multiyear, and near-global CDR of the fine-mode and coarse-mode components of atmospheric dust provides monthly-mean mass concentration profiles, on regular grids with 1°×1° spatial resolution spanning 70° S and 70° N, maintaining the original 60 m vertical resolution of CALIOP, and for the entire lifetime of the CALIPSO

mission. These profiles enable the investigation of the study's central scientific question: (i) to what extent near-surface dust mass concentrations have changed over highly industrialized and densely populated large cities and megacities worldwide over the past two decades, (ii) which of these regions experience dust mass concentrations exceeding WHO air quality guidelines, and (iii) where dust mass concentrations are projected to surpass WHO air quality guidelines in the near future. More specifically, near-surface total (Fig.4b), fine-mode (Fig.4c), and coarse-mode (Fig.4d) dust mass concentration retrievals

have the capacity to provide critical insights into the potential impact of airborne particulate matter from mineral dust on human health, particularly in regions where concentrations exceed WHO-recommended safety thresholds for annual-mean and/or hourly-mean $PM_{2.5}$ and $PM_{10}$ (Table 1). This necessity of high-quality information on the spatial distribution of global surface mineral dust concentrations is reflected in the collaborative efforts of several national and international initiatives, including, among others, the World Meteorological Organization (WMO) and Global Atmosphere Watch (GAW) Sand and

Dust Storm-Warning Advisory and Assessment System (SDS-WAS; https://community.wmo.int/en/activity-areas/gaw/science-for-services/sds-was; last access: 13/04/2025), the United Nations Environment Programme (UNEP) Global Environment Outlook (GEO; https://www.unep.org/resources/global-environment-outlook; last access: 13/04/2025), the Dust Alliance for North America (DANA; https://dust.cira.colostate.edu/; last access: 13/04/2025), the European Space Agency (ESA), and the Climate Change Initiative (CCI; https://climate.esa.int/en/; last access: 13/04/2025), aiming to address the

multifaceted challenges posed by atmospheric dust hazards.





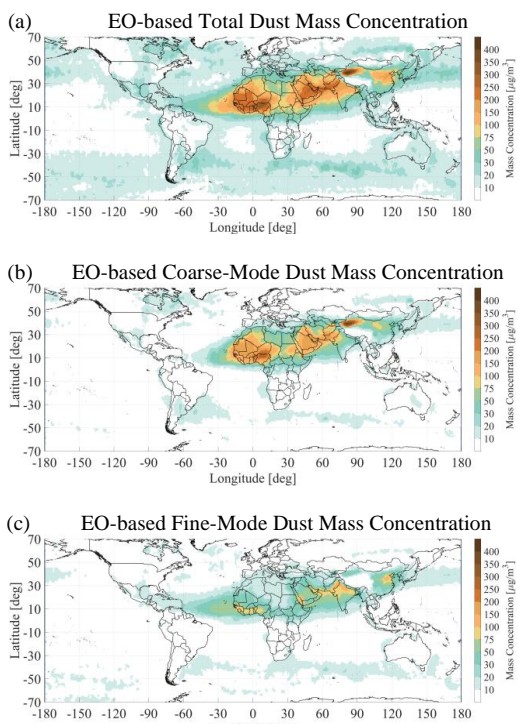

Figure 4: EO-based total (Fig.4a), coarse-mode (Fig.4b), and fine-mode dust (Fig.4c) dust mean mass concentration (μg/m³), corresponding to the altitudinal range 0 - 500 m above mean surface elevation, gridded at 1°×1° spatial resolution and for the temporal period extending between June 2006 and July 2023.

**2.1.2 ERA5 PBL**

Satellite-based lidar remote sensing is capable of providing the vertical structure of aerosol layers in the atmosphere with high accuracy. However, the retrieved profiles exhibit significant signal uncertainties near the surface, as described in Winker et al. (2013) and Tackett et al. (2018). More specifically, elevated uncertainties in backscatter signals within range bins close to the surface arise from multiple factors, including the "negative signal anomaly" (NSA), which generates substantial negative signals near the surface (Vaughan et al., 2018). In addition, the "aerosol-base extension" algorithm fails to account for clear-air gaps below 90 m above the local surface or in cases where negative integrated attenuated backscatter signals occur near the surface (Vaughan et al., 2010). Other contributing factors include strong surface returns, particularly in regions characterized by high variability in local surface elevation within L2 5 km segments (Proestakis et al., 2019) or over high-albedo surfaces (Hunt et al., 2009), which can lead to unrealistic large values. Furthermore, a lower signal-to-noise ratio (SNR), primarily due to solar noise during daytime illumination conditions, reduces the detection sensitivity of atmospheric faint layers (Winker et al., 2013) and contributes to higher uncertainties (Young et al., 2013). These sources of uncertainties produce large negative or positive signals near the surface and bias level 3 (L3) EO-based products low or high, respectively. To compensate for situations like this, near surface values the mean backscatter coefficient at 532 nm, extinction coefficient at 532 nm, and mass concentration values of range bins between the minimum surface elevation and the planetary boundary layer (PBL) height are considered.





The underlying assumption is that the atmosphere due to turbulent mixing is well mixed and relatively homogenized within
the PBL both for marine and continental conditions (McGrath-Spangler and Denning, 2013; Luo et al., 2014). More
specifically, the boundary layer as a part of the lower troposphere contains a substantial portion of the air's mass and a
significant concentration of aerosols, influencing air quality, climate, and weather (Stull, 1988). In addition, strong interactions
between aerosols, radiation, and clouds take place within the boundary layer, shaping atmospheric processes and affecting
local pollution dispersion to global climate patterns (Li et al., 2017). The boundary layer over land, particularly in urban areas,
can reach considerable heights during the daytime due to the combined effects of surface heating, anthropogenic heat
emissions, and turbulence generated by buildings and other structures (von Engeln and Teixeira, 2013; Barlow, 2014), in
addition characterized by strong seasonality (Medeiros et al., 2005). These processes affect the overall structure and dynamics,
leading to a deeper and more convective boundary layer (Hildebrand and Ackerman, 1984), enhancing at the same time the
boundary layer vertical mixing and resulting to more homogeneous conditions in terms of aerosol mixing (Hägeli et al., 2000;
Pal et al., 2010).

The PBL data used for this study were sourced for the ERA5 Reanalysis dataset (Hersbach et al., 2020), which represents the
fifth generation of atmospheric reanalyses produced by the European Centre for Medium-Range Weather Forecasts (ECMWF).
The ERA5 dataset provides global coverage at a horizontal resolution of 0.25°×0.25° (approximately 28 km at the equator)
and contains hourly estimates of various atmospheric, land, and oceanic climate variables derived from model simulations with
assimilated observations. The data used in this study were extracted from the monthly averages of single level data collection,
accessible through ECMWF's Climate Data Store (CDS; https://cds.climate.copernicus.eu/datasets/reanalysis-era5-single-
levels-monthly-means?tab=overview; last access: 13/04/2025). To produce the timeseries for the urban locations of interest,
the PBL data was spatially interpolated from the native ERA5 grid to the geographical coordinates of each city under
investigation.

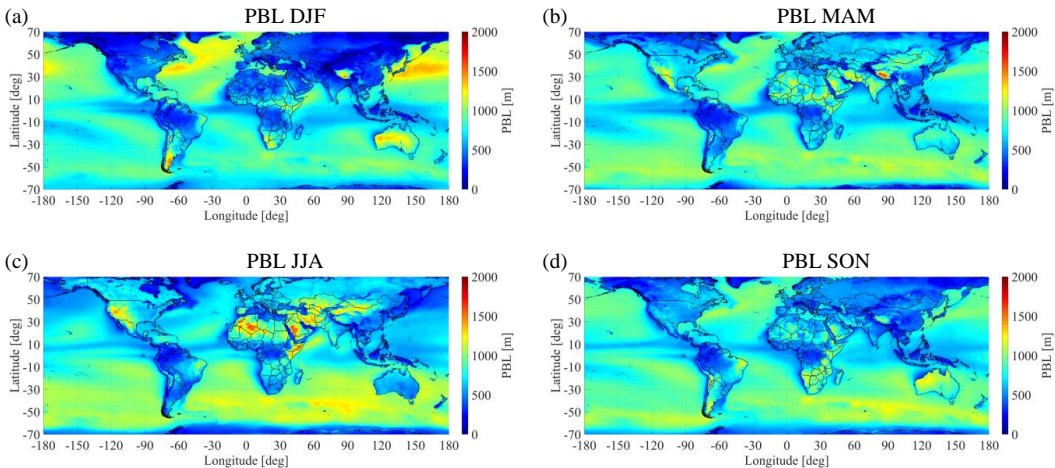

Figure 5: ERA5 mean PBL height (m) above mean surface elevation, seasonally grouped for December–January–February
(DJF; Fig.5a), March–April–May (MAM; Fig.5b), June–July–August (JJA; Fig.5c), and September–October–November
(SON; Fig.5d) and for the year-period 2006 to 2023.

### 2.1.3 AERONET fine-mode and coarse-mode AOT

AERONET is composed of sunphotometers that have been manufactured by CIMEL (type CIMEL CE318, commonly referred
as CIMEL sunphotometers) and are deployed at more than 600 stations world-wide, providing global coverage. The CIMEL



measures sun and sky radiance at nine distinct wavelengths (340 nm, 380 nm, 440 nm, 500 nm, 675 nm, 870 nm, 940 nm, and 1020 nm). Spectral aerosol optical properties are retrieved for eight of the overall nine channels (the 940 nm channel is used to retrieve the columnar water vapor) (Holben et al., 1998). AERONET products are widely used for satellite validation (e.g., Fan et al., 2023; Cheng et al., 2012; Tripathi et al., 2005) and climatological studies (e.g., Holben et al., 2001; Toledano et al., 2007; Kaskaoutis et al., 2007). Sky radiance measurements are performed for solar zenith angles 50° - 75° and are then

processed using an inversion algorithm (Dubovik and King, 2000) to derive, among other products, a bi-modal particle size distribution. The fine and the coarse-mode optical depths are subsequently derived from the size distribution and the spectral shape of the total aerosol optical depth (O' Neill et al., 2003). The fine and coarse-mode AOT at 500 nm is available at the AERONET web-page (https://aeronet.gsfc.nasa.gov/; last access: 13/04/2025). An automated quality control and cloud screening algorithm has been applied to produce the AERONET version 3, level 2 (V3L2) product that has been used in this

study (Giles et al., 2019). The uncertainties in the V3L2 AERONET products are discussed analytically in Sinyuk et al. (2020).

### 2.1.4 Megacities and Population

In this study, we focus on megacities (with populations exceeding 10 million), while also including cities with population of

5-10 million, as they have the potential to become megacities in the future, based on accurate population data and projections from the United Nations (UN) Population Division of the Department of Economic and Social Affairs (UN, 2018a, 2019). The UN's collection of datasets (https://population.un.org/wup/; last access: 13/04/2025) provides city population estimates and projections, in thousands of inhabitants, on an annual basis over a long time period (1950-2035) (UN 2018b), allowing us to identify cities with the highest population changes. Table 2 summarizes the number of cities based on their population in 2007

and future projections. According to UN projections, 26 cities with populations between 5 and 10 million are expected to become megacities by 2035. Here, we focus on 81 cities with more than 5 million inhabitants as reported in 2018 (UN, 2018a, 2019). These cities are listed in Tables 3 and 4, and their geographical locations are shown in figure 6. This figure presents the population of the 81 cities in 2007, along with future projections for 2025 and 2035, highlighting cities with the highest population growth (red points in Fig.6c).


Table 2: Number of large cities (5-10 million) and megacities (more than 10 million) according to their population for the years 2007 (reference year), 2025, 2030, and 2035 (adopted by UN, 2018a, 2019).

|  | Year | | | |
|---|---|---|---|---|
|  | 2007 | 2025 | 2030 | 2035 |
| Megacities (≥ 10 million): | 21 | 36 | 42 | 47 |
| Large cities (5-10 million): | 60 | 45 | 39 | 34 |

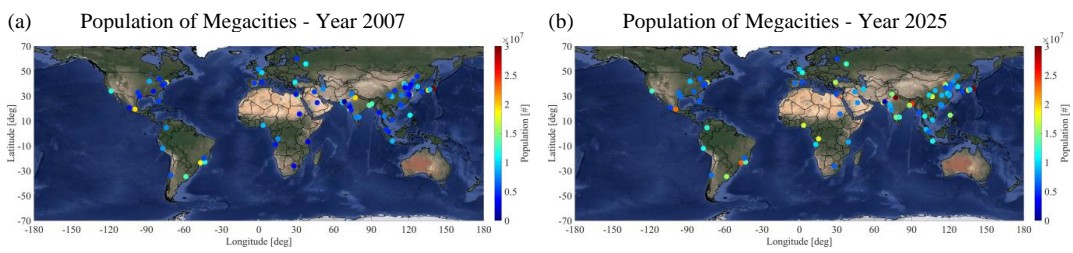

(a)    Population of Megacities - Year 2007          (b)    Population of Megacities - Year 2025

(c)    Population of Megacities - Year 2035          (d)    Change of Population - Years: 2006-2035





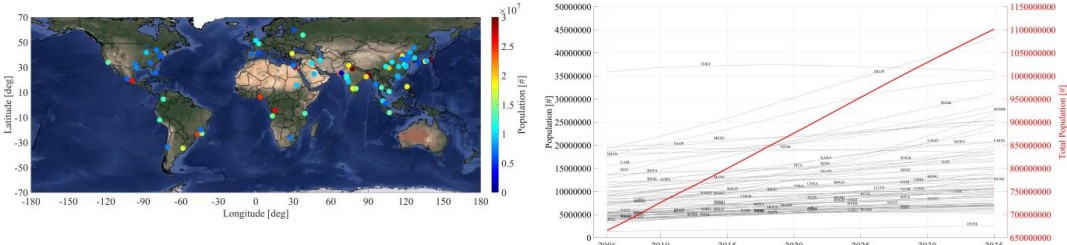

Figure 6: Cities of the world with population greater than 5 million up to 2018 (adopted by UN, 2018a, b, 2019). Focusing on population change, the figure reports on the population of the largest cities, as documented for the years 2007 (Fig.6a), 2025 (Fig.6b), for the projected population of the largest cities on the year 2035 (Fig.6c), including the total increase of the population of all cities between 2007 and present day (2025) and as projected to increase in the years till 2035 (Fig.6d). Layer background: © Google Maps.

### 2.1.5 Health risk Assessment

The health risk assessment due to exposure to air pollution is achieved through the estimation of the Relative Risk. The latter expresses the risk of a population for a health effect (e.g. all-cause mortality, lung cancer etc.) when exposure to certain levels of air pollution takes place. Given a city or an area, exposure to PM10 or PM2.5 levels can be assessed using concentration-response functions obtained from epidemiological studies. These functions use a target concentration which represents a baseline where no health effect is observed.

For short-term exposure to PM$_{10}$, the Relative Risk for all-cause mortality and for all ages is estimated by (Ostro, 2004):

$$RR = e^{\beta(C-C_0)} \tag{1}$$

where C is the annual mean concentration of PM$_{10}$, $C_0$ is the baseline PM$_{10}$ concentration under which no health effect is expected (10 μg m$^{-3}$), and β is a coefficient (0.0008) based on the concentration-response factor estimated by epidemiological studies (Soares et al. 2022).

For long-term exposure to PM2.5, the Relative Risk for cardiopulmonary mortality and lung cancer and for age groups over 30-years-old is estimated by (Ostro 2004):

$$RR = \left[\frac{C+1}{C_0+1}\right]^{\beta} \tag{2}$$

where C is the annual mean concentration of PM$_{2.5}$, $C_0$ is the baseline PM$_{2.5}$ concentration (3 μg m$^{-3}$), and β is a coefficient equal to 0.15515 for cardiopulmonary mortality and equal to 0.23218 for lung cancer.

Having the Relative Risk, the population attributable fraction (AF) is then estimated as (Ostro, 2004):

$$AF = \frac{RR-1}{RR} \tag{3}$$

The attributable fraction is defined as the proportional reduction in a disease (e.g. all-cause mortality, cardiopulmonary mortality or lung cancer) for the exposed population that would occur if ambient concentrations were reduced to target values (Chalvatzaki et al., 2019; Soares et al., 2022).



## 2.2 Methodology

As discussed in Proestakis et al. (2024) the CALIOP-based fine-mode and coarse-mode components of atmospheric dust are
established in quality-assured profiles of backscatter coefficient at 532 nm, extinction coefficient at 532 nm, and mass
concentration with the original L2 horizontal and vertical resolution of 5 km and 60 m respectively, along the CALIPSO orbit
path. Accordingly, the atmospheric fine-mode and coarse-mode dust products are processed to a four-dimensional, multiyear,
and near-global Level 3 (L3) CDR of monthly-mean backscatter coefficient at 532 nm, extinction coefficient at 532 nm, and
mass concentration, on a regular grid with 1°×1° spatial resolution spanning between 70° S and 70° N, maintaining the original
60 m vertical resolution of CALIOP, and for the entire lifetime of the CALIPSO mission. However, the L3 fine-mode and
coarse-mode atmospheric dust CDR lacks the spatial and temporal resolution necessary to effectively address the core scientific
objectives of the study, for two primary reasons.

First, the likelihood that the center of a large city or megacity coincides with or is in close proximity to the center of a L3
1°×1° grid cell is relatively low, which leads to the introduction of spatial biases. Therefore, a more suitable L3 CDR for the
fine-mode and coarse-mode components of atmospheric dust must be established. To achieve this, for each large city or
megacity, a 100 km radius surrounding the city center is defined. Within this area, all CALIOP-based L2 5 km quality-assured
profiles of fine-mode and coarse-mode backscatter coefficient at 532 nm, extinction coefficient at 532 nm, and mass
concentration are averaged. The selection of a 100 km radius is supported by the mesoscale aerosol variability in the lower
troposphere (Pappalardo et al., 2010), the CALIPSO Selective Iterated BoundarY Locator (SIBYL) feature detection
algorithm, which employs horizontal averaging up to 80 km along the CALIPSO orbital track (Vaughan et al., 2009), and the
near-zero swath of CALIOP with a footprint of approximately 100 m on the Earth's surface (Omar et al., 2013). In addition,
while consideration of a maximum CALIPSO overpass distance of less than 100 km of the city center may yield slight
improvements in terms of biases (Amiridis et al., 2013; Omar et al., 2013; Pappalardo et al., 2010; Proestakis et al., 2019;
Schuster et al., 2012; Proestakis et al., 2019), this approach would come at the expense of larger uncertainties due to the
reduced sample size of L2 5 km profiles for fine-mode and coarse-mode backscatter coefficients at 532 nm, extinction
coefficients at 532 nm, and mass concentration. Figure 7 illustrates the applied L2-to-L3 spatial-averaging methodology using
the Beijing, China megacity case. The CALIPSO overpasses for Beijing, both during the daytime and nighttime, within a
maximum distance of 100 km from the city center, are displayed using red and blue granules, respectively (Fig. 7a). The map
overlays white circles indicating the locations of the four AERONET long-term monitoring stations: Beijing, Beijing-RADI,
Beijing-PKU, and Beijing-CAMS. The mean dust mass concentration profiles of total (Fig. 7b), coarse-mode (Fig. 7c), and
fine-mode (Fig. 7d) for the period from December 2006 to November 2022 (16-years) reveal elevated levels of dust particulate
matter, particularly near the surface. These profiles are compared to the WHO annual mean and 24-hour mean AQG thresholds
for $PM_{10}$ and $PM_{2.5}$, additionally included in the figure. More specifically, for the case of the Beijing, China megacity the
annual mean total, coarse-mode, and fine-mode dust mass concentration profiles reach values of 115.95, 38.91, and 80.22
$\mu g \cdot m^{-3}$, respectively, significantly surpassing the WHO annual mean AQG limits for both $PM_{10}$ and $PM_{2.5}$ particulate matter
exposure.

(a)          CALIPSO orbits                    (b)   Total Dust Mass Concentration




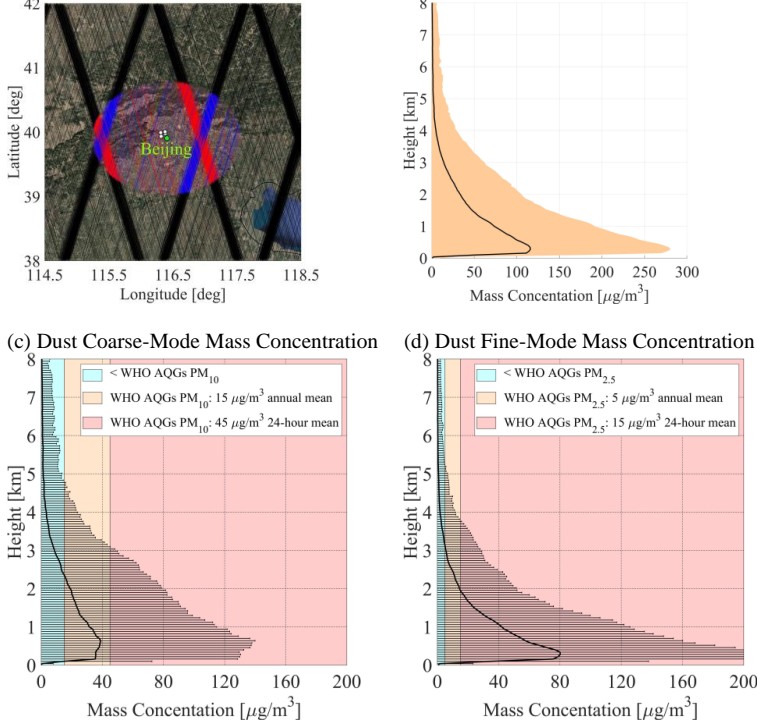

Figure 7: CALIPSO daytime (red lines) and nighttime (blue lines) overpasses within 100 km distance from the Beijing
megacity (Fig.7a), mean mass concentration (μg/m³) profiles of dust (Fig.7b), coarse-mode dust (Fig.7c), and fine-mode dust
(Fig.7d), for the period extending between 06/2006 and 07/2023. The figure provides for the cases of the coarse-mode (Fig.7c)
and fine-mode (Fig.7d) mean dust mass concentration profiles the corresponding World Health Organization (WHO) air
quality guidelines (AQG) for $PM_{10}$ and $PM_{2.5}$ respectively, and for both the cases of 24-hour-mean and annual-mean particulate
matter exposure. Layer background: © Google Maps.


Second, a considerable number of large cities and megacities are located in regions characterized by persistent cloud coverage.
More specifically, according to the United Nations (UN) Population Division of the Department of Economic and Social
Affairs (UN, 2018a, b, 2019) fourteen out of eighty-one large cities and megacities (~17.3 %) are located over the broader
Indian subcontinent (e.g. Karachi, Hyderabad, Ahmedabad, Mumbai, Surat, Pune, Bengaluru, Chennai, Delhi), the Bay of
Bengal (e.g. Kolkata, Dhaka, Bangkok), and the Gulf of Guinea (e.g. Lagos, Kinshasa). These regions experience extensive
cloud coverage (King et al., 2013; Pincus et al., 2012; 2023) for prolonged periods of time, due to the west African monsoon
circulation (Parker et al., 2005) and to Asian monsoonal activity (Dey et al., 2004; Vinoj et al., 2014). Consequently, an
increased number of CALIPSO overpasses is required to enhance the statistical representativeness of the EO-based fine-mode
and coarse-mode atmospheric dust CDR, mitigating data gaps caused by cloud contamination and improving the reliability of
the dataset. To achieve this, the L3 CDR is processed at a different temporal resolution, transitioning from a monthly-mean to
seasonal-mean representation of backscatter coefficient at 532 nm, extinction coefficient at 532 nm, and mass concentration
profiling of the atmosphere. These profiles are seasonally grouped into December–January–February (DJF), March–April–
May (MAM), June–July–August (JJA), and September–October–November (SON), covering the period from December 2006
to November 2022, thus encompassing 16 full years of CALIPSO observations.



It should be emphasized that the broad size distribution of mineral dust particles suspended in the atmosphere, ranging from less than 0.1 µm to more than 100 µm in diameter (Mahowald et al., 2014; Weinzierl et al., 2017; Van der Does et al., 2018; Ryder et al., 2018; 2019), results into frequent inconsistencies in terms of terminology of dust size classes. The general consensus is that the classification has to follow the distinct fine-mode and coarse-mode classes apparent in the size distribution of airborne dust (Seinfeld and Pandis, 2006; Whitby, 1978). However, substantial discrepancies exist in the adopted boundary

separating the two modes (Adebiyi et al., 2023), with frequent applied classification diameters including 1 µm (Mahowald et al., 2014; Mamouri and Ansmann, 2014, 2017; Ansmann et al., 2017), 2 µm (Spurny, 1998; Whitby, 1978; Willeke and Whitby, 1975), 2.5 µm (Seinfeld and Pandis, 2006; Zhang et al., 2013; Pérez García-Pando et al., 2016), 4 µm (Rajot et al., 2008), and 5 µm (Kok et al., 2017; Adebiyi and Kok, 2020). The applied separating diameter of 1 µm in the EO-based dust CDR follows the parametrizations provided by the extensive chamber laboratory experiments performed by Sakai et al. (2010)

and Järvinen et al. (2016), defining the fine-mode as the submicrometer-mode (including the Aitken and accumulation subclasses) and the coarse-mode as the supermicrometer-mode (including the coarse, super-coarse, and giant dust subclasses). The available parametrization constrain of 1 µm diameter decoupling the dust fine and coarse-modes in terms of EOs results in inconsistencies in terms of available WHO AQG, applying separating diameter of 2.5 µm in the definition of $PM_{2.5}$ and $PM_{10}$ classes. Thus, in absence of WHO AQG provided for $PM_{1.0}$ as separating diameter between the two dominant dust modes,

the present study considers the annual-mean and 24-hour mean $PM_{2.5}$ and $PM_{10}$ thresholds for the EO-based fine-mode dust and coarse-mode dust classes.

   Figure 8 shows the total (first row), coarse-mode (second row), and fine-mode (third row) dust in terms of seasonal-mean extinction coefficient at 532 nm (left column) and seasonal-mean mass concentration profiles (right column), for the period 12/2006-11/2022 and for the Beijing megacity of China. In addition, the shown black line indicates the seasonal-mean PBL

height as provided by the ECMWF ERA5 while the white cycles in the dust extinction coefficient at 532 nm figures correspond to the PBL DOD at 532 nm for the case of the total (Fig.8a), coarse-mode (Fig.8c), and fine-mode (Fig.8e) atmospheric dust. This hovmoller approach of time evolution of mass concentration vertical distribution analysis allows several interesting characteristics of atmospheric dust to be revealed. For instance, for the case of the Beijing-China megacity the seasonal-mean coarse-mode and fine-mode atmospheric dust profiles reveal that the predominant dust transport within the free troposphere

occurs during the MAM and JJA seasons (Husar et al., 2001; Che et al., 2014, 2015; Proestakis et al., 2019). This phenomenon aligns with the activation of the major natural dust emission sources located to the west of Beijing, including the Taklimakan Desert encompassed by the Tarim Basin and the vast Gobi Desert spanning across northern China and southern Mongolia (Zhang et al., 1997). The dust emissions are driven by the region's favorably topography allowing development of strong cyclonic systems over the Mongolian Plateau (Sun et al., 2001; Gong et al., 2006, Liu et al., 2008, Bory et al., 2003; Yu et al.,

2008), facilitating the eastward transport of dust across the mainland China and the broader northern Pacific Ocean (Shaw, 1980; Duce et al., 1980; Uno et al., 2001; 2009; Zhang et al., 2003; Huang et al., 2008). Furthermore, while the wave-like seasonality of long-range dust transport is evident in both coarse-mode and fine-mode dust profiles within the free troposphere, a more persistent presence of dust is observed within the PBL, particularly in the fine-mode component of dust. This finding supports the hypothesis of a substantial additional anthropogenic dust contribution to the total atmospheric dust load (Penner

et al., 1994; Tegen and Fung, 1995; Ginoux et al., 2012), attributed to extensive industrial and human activities (Moulin and Chiapello, 2006; Chen et al., 2018, Ginoux et al., 2012, Tegen et al., 1996). The timeseries of seasonal coarse-mode and fine-mode PBL DOD at 532 nm and mean mass concentrations within the PBL facilitate addressing the core scientific questions of the study, with respect to PBL dust changes over the last two decades, the possible presence of statistically significant trends, identification of cities experiencing dust concentrations exceeding WHO AQG, and cities foreseen to exceed WHO AQG in

the near-future.




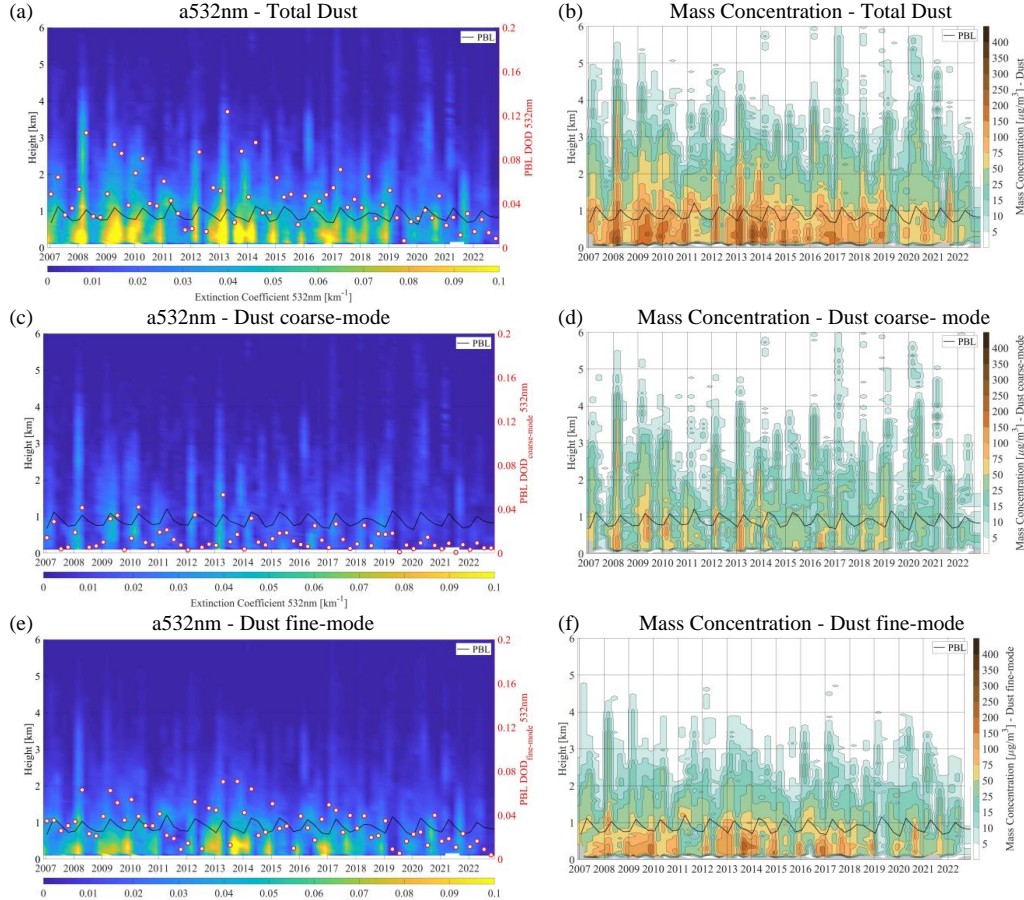

Figure 8: Dust (first row), coarse-mode dust (second row), and fine-mode dust (third row) seasonal-mean extinction coefficient
at 532 nm profiles (left column) and mass concentration profiles (right column) as established on the basis of CALIPSO
overpasses within 100 km radius from the Beijing megacity area and for the temporal period extending between DJF/2006 and
SON/2022. The black line indicates the seasonal-mean PBL height as provided by the ECMWF ERA5 while the white cycles
in the dust extinction coefficient at 532 nm figures correspond to the PBL DOD at 532 nm for the case of the total (Fig.8a),
coarse-mode (Fig.8c), and fine-mode (Fig.8e) atmospheric dust.


## 3. Consistency assessment of increasing/decreasing aerosol load tendencies

The present section aims to provide insight into the consistency of the EO-based fine-mode and coarse-mode components of
the atmospheric dust load (Sect.2.2) to reproduce features reported by ground-based AERONET observations and products of
the fine-mode and coarse-mode components of the total aerosol load (Sect.2.1.3). Towards this objective, EO-based PBL
$DOD_{fine-mode}$ and $DOD_{coarse-mode}$ at 532 nm products and AERONET-based SDA $AOT_{fine-mode}$ and $AOT_{coarse-mode}$ at 500 nm are
compared. Prior to the intercomparison, the observational quality-assured datasets are pre-processed/aggregated into seasonal
mean temporal resolution, covering the period extending between winter 2006 (DJF/2006) and autumn 2022 (SON/2022). The
intercomparison is performed for the large-cities/megacities of the world where AERONET observations cover at least half of
the EO-based period in terms of observations (> 7 yrs), allowing for more robust consistency assessment.



As indicative case of the intercomparison, the megacity of Beijing-China is presented and discussed. More specifically, figure 9 illustrates the seasonal-mean EO-based PBL $DOD_{coarse-mode}$ (Fig.9a) and $DOD_{fine-mode}$ (Fig.9b) at 532 nm products based on CALIPSO overpasses within a 100 km radius of the Beijing-China megacity. As expected, the magnitude of $DOD_{coarse-mode}$ variation exhibits seasonal dependence, with higher values typically recorded during MAM and JJA, coinciding with the high
seasonality of dust aerosol generation and transport related to activation of the Taklimakan and Gobi deserts (Husar et al., 2001; Liu et al., 2008; Uno et al., 2009; Proestakis et al., 2018). The observed trends indicate a statistically significant change in both coarse-mode and fine-mode optical depth of dust within the PBL over the period 12/2006-11/2022. Linear regression analysis yields slopes of -0.0007 $yr^{-1}$ and -0.0015 $yr^{-1}$ for the coarse-mode and fine-mode of dust, respectively. The intercept values of 0.0196 and 0.0427 suggest the baseline $DOD_{coarse-mode}$ and $DOD_{fine-mode}$ levels at the beginning of the study period
(DJF/2006). The statistical significance of the apparent negative trends computed at the significance level of 5% (both positive in the case of Beijing megacity) confirms whether the observed changes are robust and unlikely to be due to random variability. The significant decreasing trends reported in our study are consistent with the spatiotemporal variations in dust emissions and transport over East Asia and China as documented in the literature. More specifically, a well-documented decline in the overall frequency of moderate dust events and severe dust events, originating from the dust sources of Taklimakan Desert (Tarim
Basin), the Gurbantünggüt Desert (Junggar Basin), the Turpan Basin, and the Gobi Desert, since the late 1970s has been documented (Gong et al., 2006; Zhang et al., 2006; Wu et al., 2018). The observed decline in natural dust activity over East Asia in recent years (Proestakis et al., 2018; Yu et al., 2020) is primarily attributed to a reduction in strong wind days (Wu et al., 2022; Zhou et al., 2024), which has been identified as the dominant factor contributing to lower dust emissions. The weakening of surface winds has been closely linked to large-scale atmospheric circulation changes, including the Arctic
Oscillation and North Atlantic Oscillation weakening the East Asian trough and Siberian High (Gong et al., 2006; Wang et al., 2023) and the amplification of Arctic warming leading to weakening of the Polar Vortex (An et al., 2018; Gong et al., 2006; Zhang et al., 2006; Liu et al., 2020; Wu et al., 2022), weakening dust emissions and decreasing the frequency of sand and dust storms in the broader East Asia region (Zou et al., 2004; An et al., 2018; Fan et al., 2014; Liu et al., 2020; Wu et al., 2022). In addition, increased precipitation and soil moisture, along with rising vegetation coverage and leaf area index, have also played
a significant role in suppressing dust emissions over arid and semiarid regions by stabilizing surface soils (Shi et al., 2007; Peng and Zhou, 2017; Kraaijenbrink et al., 2021; Zhou et al., 2024).

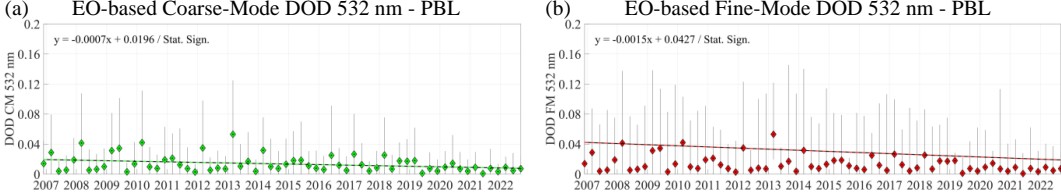

Figure 9: EO-based seasonal-mean PBL coarse-mode DOD at 532 nm (Fig.9a) and fine-mode DOD at 532 nm (Fig.9b) and
reported trends, computed on the basis of CALIPSO overpasses within 100 km radius from the Beijing megacity area and for the temporal period extending between DJF/2006 and SON/2022.

The statistically significant negative trends of EO-based PBL $DOD_{coarse-mode}$ and $DOD_{fine-mode}$ at 532 nm over the Beijing-China megacity (Fig. 9) are consistent with the negative trends in AERONET $AOT_{coarse-mode}$ and $AOT_{fine-mode}$ at 532 nm derived from
SDA retrievals (Fig. 10) on the basis of sunphotometer measurements conducted by the Beijing, Beijing-RADI, Beijing-PKU, and Beijing-CAMS AERONET stations, operating in the proximity of the city center. The observed statistically significant negative AERONET AOT trends indicate a decline in the total aerosol load, including though both dust and non-dust aerosol load components. Moreover, the reported AERONET trends (Table 3) are in agreement with several scientific studies reporting



AOD decrease over the broader eastern China region mainly after 2010 (Hsu et al., 2012; Zhao et al., 2017; Sogacheva et al., 2018; Zheng et al., 2018; Proestakis et al., 2018; Gupta et al., 2022), following a series of strict air quality regulatory control measures enforced by the Chinese government, including among others, the Air Pollution Prevention and Control Action Plan (2013–2017), the Blue Sky Protection Campaign (2018–2020), the China VI vehicle emission standards, and decrease in terms of particulate matter ($PM_{2.5}$ and $PM_{10}$) concentration limits.

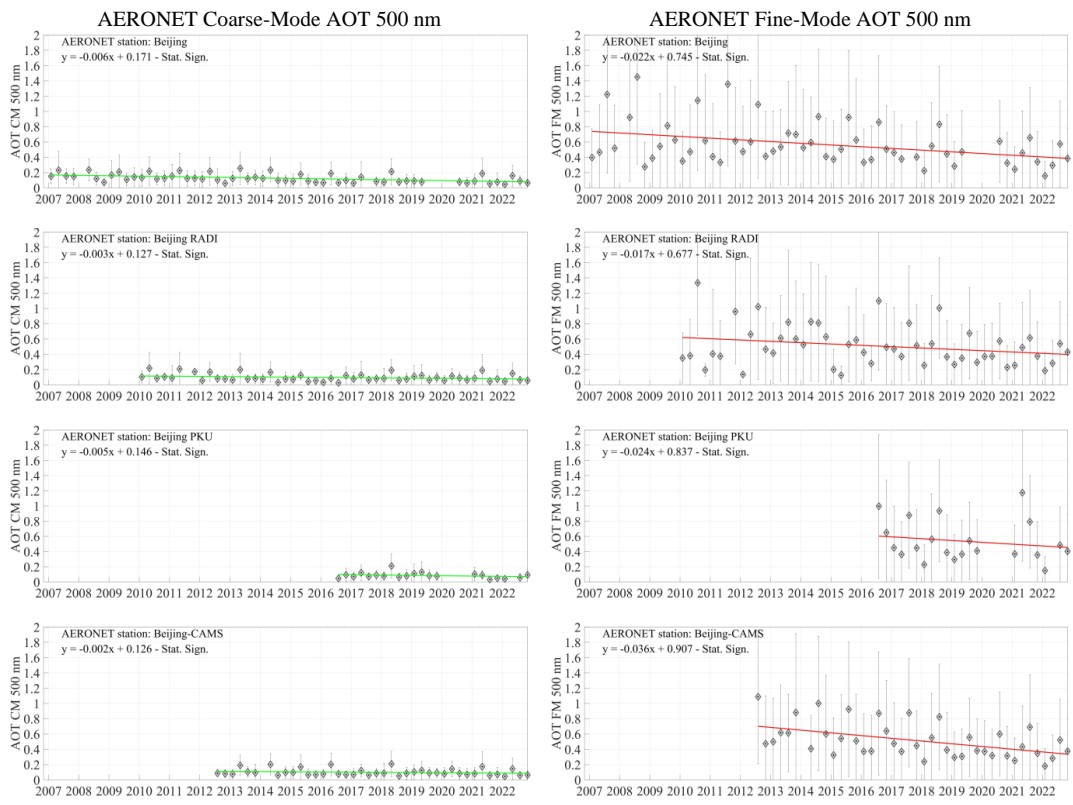

Figure 10: AERONET-based SDA seasonal-mean coarse-mode AOT at 500 nm (Fig. 10-left panel) and fine-mode AOT at 500 nm (Fig. 10-right panel) and reported trends, for the cases of Beijing (Fig. 10-first row), Beijing-RADI (Fig. 10-second row), Beijing-PKU (Fig. 10-third row), and Beijing-CAMS (Fig. 10-fourth row) AERONET stations.

Table 3: Overall tendencies of EO-based PBL fine-mode and coarse-mode DODs at 532 nm components, computed on the basis of CALIPSO overpasses within 100 km radius from the Beijing megacity area and for the temporal period extending between DJF/2006 and SON/2022, and AERONET fine-mode and coarse-mode AOTs at 500 nm for the Beijing, Beijing-RADI, Beijing-PKU, and Beijing-CAMS AERONET stations, including slope ($S_{fit}$) and intercept ($I_{fit}$) of a linear regression fit and statistical significance at 95% confidence level.

| | Coarse-Mode | | | Fine-Mode | | |
|---|---|---|---|---|---|---|
| | Slope | Interception | Statistical Significance | Slope | Interception | Statistical Significance |
| EO-based[1] | -0.0007 | 0.0196 | ✓ | -0.0015 | 0.0427 | ✓ |
| Beijing[2] | -0.006 | 0.171 | ✓ | -0.022 | 0.745 | ✓ |
| Beijing RADI[2] | -0.003 | 0.127 | ✓ | -0.017 | 0.677 | ✓ |
| Beijing PKU[2] | -0.002 | 0.126 | ✓ | -0.036 | 0.907 | ✓ |
| Beijing CAMS[2] | -0.005 | 0.146 | ✓ | -0.024 | 0.837 | ✓ |





1: DOD at 523 nm / 2: AOT at 500 nm.

The intercomparison between EO-based PBL DOD products for coarse and fine-modes at 532 nm and corresponding AERONET AOT retrievals, used as reference, is crucial for enhancing the reliability and confidence in the spatial and temporal patterns observed. In addition, within the framework of the EARLINET well-established dust component separation two-step

POLIPHON technique (Mamouri and Ansmann, 2014, 2017), the conversion of extinction coefficient profiles at 532 nm into dust mass concentration profiles is conducted using an assumed characteristic dust particle density ($\rho_d$) of 2.6 g cm$^{-3}$ (Ansmann et al., 2012) and regional-dependent dust extinction-to-volume concentration conversion factors derived from AERONET-EARLINET synergy (Ansmann et al., 2019). Given that the EO-based PBL dust mass concentration (MC) products for fine and coarse-modes exhibit similar spatiotemporal patterns with the corresponding EO-based DOD products (Proestakis et al.,

2024), the intercomparison with reference AERONET measurements and retrievals strengthens the credibility of conclusions drawn regarding dust mass concentration levels and their temporal evolution, as in the case of large cities and megacities of the world globally. Figure 11 provides the seasonal-mean EO-based PBL dust mass concentration for coarse-mode (Fig. 11a) and fine-mode (Fig. 11b) aerosols over the Beijing-China megacity, derived from CALIPSO observations within a 100 km radius and spanning the period from DJF 2006 to SON 2022. The shaded areas in both subfigures represent the WHO annual

mean AQG thresholds for PM$_{10}$ (15 µg m$^{-3}$; coarse-mode proxy) and PM$_{2.5}$ (5 µg m$^{-3}$; fine-mode proxy), providing a benchmark for evaluating the severity of dust exposure. In both cases of coarse-mode and fine-mode dust mass concentration, the long-term mean exceeds the WHO-recommended air quality thresholds, with values as high as 24.1 ± 23.9 µg m$^{-3}$ and 61.9 ± 35.7 µg m$^{-3}$, respectively. The observed trends indicate a statistically significant negative (decreasing) change in both coarse-mode and fine-mode load of dust within the PBL -in terms of mass concentration- over the period 12/2006-11/2022. Linear

regression analysis yields slopes of -1.8 µg m$^{-3}$ yr$^{-1}$ and -2.7 µg m$^{-3}$ yr$^{-1}$ for the coarse-mode and fine-mode of dust, respectively. The intercept values of 48.6 µg m$^{-3}$ and 84 µg m$^{-3}$ suggest the baseline PBL MC$_{coarse-mode}$ and MC$_{fine-mode}$ levels at the beginning of the study period (DJF/2006) over Beijing. The statistical significance of the apparent negative trends computed at the significance level of 5% (both positive in the case of Beijing megacity) confirms whether the observed changes are robust and unlikely to be due to random variability.


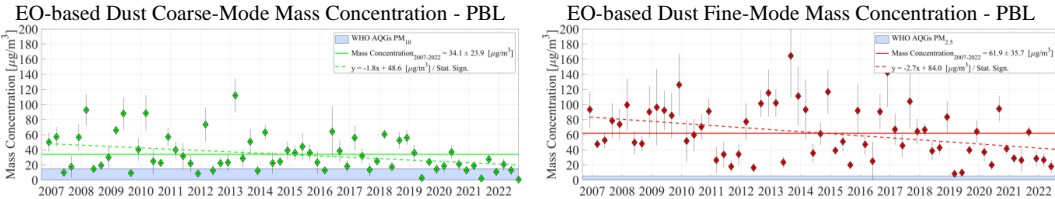

Figure 11: EO-based seasonal-mean PBL coarse-mode dust mass concentration (Fig.11a) and fine-mode dust mass concentration (Fig.11b) and reported trends, computed on the basis of CALIPSO overpasses within 100 km radius from the Beijing megacity area and for the temporal period extending between DJF/2006 and SON/2022. The shaded areas denote the

WHO AQG for PM$_{10}$ (Fig.11a) and PM$_{2.5}$ (Fig.11b) in terms of annual-mean particulate matter exposure.

Expanding the analysis to the entire dataset of the world's large cities and megacities yields insight into the spatial and temporal patterns observed over the highly urbanized and industrialized areas at a global scale. Towards this objective, figure 12a illustrates the geographical distribution of large cities and megacities of the world with available long-term (more than 6 yrs)

AERONET fine-mode AOT at 500 nm product (in total 34 cases) used in the present assessment analysis. More specifically, the figure provides a global map displaying the spatial distribution of EO-based fine-mode DOD at 532 nm, overlaid with the locations of the cities where analysis of trend's intercomparison is feasible. The color of each city location signifies the level



of agreement between the EO-based and AERONET trends, where locations shown in green (red) color indicate cases in agreement (disagreement) regarding the direction of fine-mode aerosol trends provided by satellite remote sensing observations and ground-based measurements. According to the intercomparison, a positive linear correlation of the derived trends is observed (R = 0.47) accompanied by low mean absolute bias (MBE = 0.004 yr$^{-1}$). Figure 12b summarizes the available for the assessment cases, comparing the EO-based PBL fine-mode DOD at 532 nm (y-axis) with the AERONET fine-mode AOT at 500 nm trends (x-axis). The figure is divided into four quadrants labeled ($A_1$), ($A_2$), ($B_1$), and ($B_2$), which classify the trends into positive and negative categories. The first quadrant ($A_2$) represents cities where both trends are positive, suggesting increasing fine-mode aerosols, while the third quadrant ($B_1$) represents cities where both trends are negative, indicative of decreasing fine-mode aerosols. The second ($A_1$) and fourth ($B_2$) quadrants correspond to mixed trend behavior, where one observational dataset shows an increasing trend while the other shows a decreasing trend. Figure 12c is a confusion matrix summarizing the level of agreement between the two observational datasets, the EO-based fine-mode DOD at 532 nm trends and the AERONET-based fine-mode AOT at 500 nm trends. The matrix is divided into four domains corresponding to the scatter plot quadrants (Fig. 12b), with each region showing the number of cases in that specific quadrant and their percentage of the total. The total accuracy (TA) of the trends characterized by identical sign is quantified to ~61.8%, computed as the sum of $B_1$ and $A_2$ quadrants to the total number of cases. According to the assessment, the majority of the data points (52.9%) fall in the third quadrant ($B_1$), indicating a prevalent decreasing trend in fine-mode aerosols across both datasets. The first quadrant ($A_2$) has 8.8% of the sites showing agreement on increasing trends. The remaining quadrants ($A_1$) and ($B_2$) contain 38.2% of the total cases, highlighting instances where trends are less consistent between the two datasets.

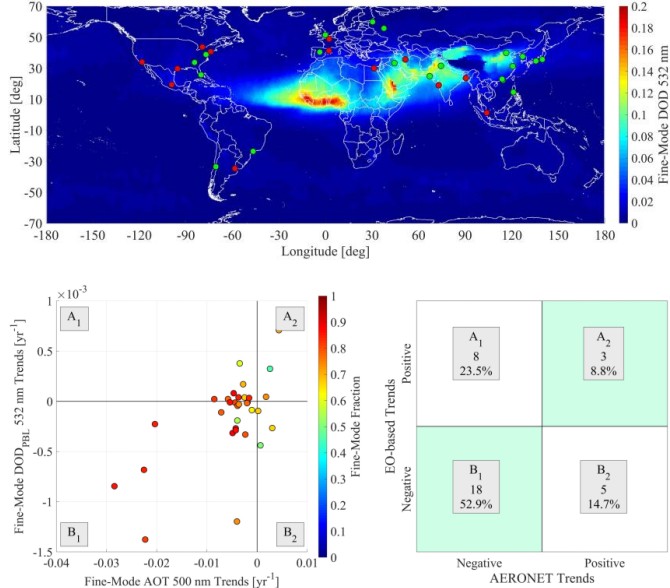

Figure 12: Geographical location of the large cities of the world for which both EO-based temporal trends of fine-mode PBL DOD at 532 nm and AERONET-based temporal trends of fine-mode AOT at 500 nm are calculated (Fig.12-upper panel). Green and red dots denote world-cities with similar and different trend signs, respectively. Scatter plot between EO-based temporal trends of fine-mode PBL DOD at 532 nm and AERONET-based temporal trends of fine-mode AOT at 500 nm (Fig.12-lower left). The color bar indicates the AERONET fine-mode fraction. Confusion matrix of possible trend signs among the two datasets (Fig.12-lower right).





Overall, this statistical analysis of the apparent trends between the CALIPSO-based PBL DOD$_{fine-mode}$ at 532 nm product and the AERONET AOT$_{fine-mode}$ at 500 nm retrievals, emphasizing the degree of agreement between different observational datasets. The clustering of the majority of the data in the lower left quadrant suggests a general decline in fine-mode aerosols, which could be attributed in addition to regional meteorological conditions to regulatory measures and enforced policies aiming to reduce anthropogenic emissions in urban and highly industrialized areas and to mitigate aerosol-related induced disorders on human health (Jin et al., 2016; Turnock et al., 2016; WHO, 2016; UNEP, 2016; Zhao et al., 2017; Zheng et al., 2018; Sogacheva et al., 2018; Abera et al., 2020; Ganguly et al., 2020; Gao et al., 2021; Gupta et al., 2022). It is important though to emphasize that the present intercomparison serves as an indirect consistency assessment between the CALIPSO-based PBL DOD$_{fine-mode}$ at 532 nm product and AERONET AOT$_{fine-mode}$ at 500 nm retrievals, respectively. The analysis does not constitute a direct validation or an evaluation process for several reasons, primarily due to the fundamental distinction that the two datasets represent different physical quantities. More specifically, AERONET measurements and retrievals (AOT, AOT$_{coarse-mode}$, and AOT$_{fine-mode}$) provide information on the total aerosol load in terms of column-integrated optical properties at 500 nm, especially during daytime illumination conditions, spanning the entire atmospheric column from the Earth's surface to the top of the atmosphere. In contrast, the satellite-based derived products (DOD, DOD$_{coarse-mode}$, and DOD$_{fine-mode}$) represent the column-integrated mean extinction coefficient at 532 nm obtained during both daytime and nighttime, characterizing the supermicrometer and submicrometer components of atmospheric dust, within the planetary boundary layer from the Earth's surface to the PBL height. Consequently, due to the spatiotemporal inhomogeneity of aerosol fields, especially due to the mesoscale variability of aerosol residing within the lower troposphere, CALIOP and AERONET rarely sample the same air volumes (Pappalardo et al., 2010; Schuster et al., 2012). These fundamental differences between the intercomparison datasets, as extensively discussed in Proestakis et al. (2024), additional discrepancies between the coarse-mode and fine-mode of EO-based PBL DODs and AERONET AOTs arise from various factors, as extensively discussed in Proestakis et al. (2024). Discrepancies related to the distinct characteristics of CALIOP (Hunt et al., 2009; Winker et al., 2009) and sunphotometer measurements (Holben et al., 1998), discrepancies due to the retrieval algorithms of AEROET (Eck et al., 1999; O'Neill et al., 2001a, b, 2003) and EO-based products (Vaughan et al., 2009; Kim et al., 2018; Kar et al., 2019; Vaughan et al., 2019; Zeng et al., 2019), the applied quality assurance criteria (Marinou et al., 2017; Tackett et al., 2018), and the dust decoupling technique (Shimizu et al., 2004; Mamouri and Ansmann, 2014, 2017; Tesche et al., 2009).

## 4. Atmospheric dust and Air Quality - 16 years of Earth Observations (2007-2022)

The present section aims to address the scientific questions: which urban areas experience fine-mode and coarse-mode dust mass concentrations within the PBL exceeding WHO AQG and is it feasible to identify statistically significant trends? Identifying large cities and megacities where PBL dust PM$_{2.5}$ and PM$_{10}$ concentrations exceed WHO AQG enhances the necessary baseline for environmental health risk assessments, since such information is critical for implementing targeted mitigation strategies, for public health interventions, and for both climate modeling and air quality management. Moreover, addressing the question of statistically significant trends presence in dust concentration changes during the past decades is equally important since it enhances the potential to provide essential information whether conditions are changing over time, allowing policymakers to assess whether dust-related air quality is improving, deteriorating, or remaining stable, particularly in the context of desertification, urbanization, shifting weather patterns, and under the ongoing climate change.

To date, towards addressing these scientific questions numerous dust model-based and EO-based studies have investigated the long-term spatial and temporal variability and the tendencies of the atmospheric dust load, from local to planetary scales. Logothetis et al. (2021) studied the long-term variability of dust optical depth (DOD) over a 15-year period (2003 – 2017), using the MIDAS satellite dataset (Gkikas et al., 2021; 2022). Their findings indicate positive trends over the Arabian Peninsula and western Sahara, while declining tendencies were observed in the eastern Sahara, the Mediterranean, the Thar Desert and



eastern China. Declining DOD trends in East Asia were also reported by Zhao et al. (2023) (2007-2019) and Zhao et al. (2024) (2007-2021), based on CAMS numerical products. These findings were consistent with those of Korras-Carraca et al. (2021), who processed the MERRA-2 aerosol products (2001-2019) and of Proestakis et al. (2018) who performed analysis on long-term CALIOP observations. The reduction in dust loads' intensity in East Asia over recent decades (since 2000) is further supported by Shin et al. (2023), who reported the predominance of negative trends in major Chinese, Korean and Japanese cities where AERONET sun-photometers are in operation. Recent studies (Wang et al., 2021; Wang et al., 2021; Du et al., 2023) have also reported a weakening of dust activity in deserts and desertified areas across the broader region, resulting in a reduced atmospheric dust burden. Ouma et al. (2024) showed that particulate matter with an aerodynamic diameter up to 2.5 μm (PM$_{2.5}$), as simulated in the MERRA-2 reanalysis, exhibited an increasing trend over the period 1980-2021 in Africa. This positive tendency was driven by the temporal variability of mineral particles. Across the African continent, the highest positive rates were observed in the southern, eastern and northern sectors, while almost neutral or slightly negative trends were recorded in the western and central parts, respectively. Hsu et al. (2012), relying on SeaWiFs observations, reported a neutral AOD trend in the western Sahara and a weak increasing tendency in the eastern Sahara, over a period spanning from 1997 to 2010. Notable positive AOD tendencies have been reported by Klingmüller et al. (2016) in dust-rich areas of the Arabian Peninsula between 2000 and 2015, based on MODIS observations. Notaro et al. (2015) demonstrated a strong contrast in dust optical depth (DOD) anomalies over Saudi Arabia between the periods 1998–2005 and 2007–2013, with negative and positive anomalies, respectively. These patterns were linked to drought conditions in the Fertile Crescent. The predominance of increasing AODs in most parts of the Arabian Peninsula has also been discussed by Che et al. (2019), who relied on both reanalysis (MERRA-2) and satellite (MODIS) aerosol products over a 16-year period (2001-2016). The upward trend of aerosol optical depth during the first decade of the 21$^{st}$ century has been further confirmed by several studies (Hsu et al., 2012; de Meij et al., 2012; Pozzer et al., 2015; Wei et al., 2019) relying either on numerical simulations or observations. The Mediterranean is frequently under the impact of dust outbreaks (Gkikas et al., 2016), due to its proximity to the most active deserts worldwide (Ginoux et al., 2012) and the prevailing atmospheric circulation favoring dust advection from the Sahara Desert and the Arabian Peninsula (Gkikas et al., 2015). Salvador et al. (2022) found an increase in both the frequency of occurrence and intensity of African dust outbreaks affecting the Iberian Peninsula and the Balearic Islands (1948-2020). Other satellite-based studies (de Meij et al., 2012; Hsu et al., 2012; Klingmüller et al., 2016) have shown negative DOD trends in the western Mediterranean opposite to the positive ones observed in the eastern sector during the early 21st century. This dipole pattern was not evident in Marinou et al. (2017), who demonstrated the predominance of negative DOD trends across the Mediterranean based on CALIPSO spaceborne retrievals (2007-2015). Similar findings were drawn by Logothetis et al. (2021) over an extended period (2003-2017). In the eastern Mediterranean, slightly increasing DOD trends were found during winter (2000-2017), consisting of positive trends before 2010 and negative trends thereafter (Shaheen et al., 2021). For the same region, and over a longer period (2000-2020), a similar DOD variability has been revealed for the high-dust season (April-July) over 2000-2017, followed by a neutral trend till 2020 (Shaheen et al., 2023). Aryal and Evans (2022) investigated the frequency of occurrence and the intensity of dust events in the western United States using PM measurements from the IMPROVE network (2001-2020). Their results indicate a notable decline in the frequency of high-concentration dust events during spring and summer, while low-concentration dust events have become more frequent. Additionally, the intensity of spring dust events has decreased for both high- and low-concentration cases. Tong et al. (2017), relying on the same ground-based measurements, but for an earlier period (1988 – 2011), demonstrated an intensification of dust activity in southwestern United States. Gasso and Torres (2019) reported a significant increase of dust activity in central Patagonia (South America) between 1964 and 2017, with an acceleration after 2007, based on both ground-based and satellite-derived observations. Finally, Che et al. (2024) reported a reduction in the total amount of dust emitted in Australia between 1980 and 2020, based on MERRA-2 simulations.



However, the assessment of the long-term spatial and temporal variability and the tendencies of the atmospheric dust load with focus on the fine-mode and coarse-mode components within the PBL over densely populated and heavily industrialized urban areas of the world is characterized by significant challenges, hampering the implementation of the affirmations trends to reflect into and assess the environmental risk factor dust aerosols pose for human health. The current consensus asserts that the total atmospheric dust load is the cumulative result of natural dust and anthropogenic dust, a distinction grounded in dust sources

and emission mechanisms (Penner et al., 1994; Tegen and Fung, 1995; Ginoux et al., 2012). Natural dust entrainment into the atmosphere is driven by several meteorological phenomena and physical processes, such as low-level jets (LLJs; Fiedler et al., 2013), haboobs (Knippertz et al., 2007), dust devils (Koch and Renno, 2005), and pressure gradients (Klose et al., 2010), which develop primarily over arid and sparsely vegetated desert regions (Prospero et al., 2002). Anthropogenic dust originates directly or indirectly from human activities, such as transportation, infrastructure projects, and construction activities (Moulin

and Chiapello, 2006; Chen et al., 2018), soil surface degradation and modifications of terrestrial surfaces, including deforestation and overgrazing (Ginoux et al., 2012), expansion of urban areas and agricultural land management (Tegen et al., 1996). To date, several scientific studies have highlighted that the anthropogenic dust contribution to the total atmospheric dust load is substantial. Ginoux et al. (2012) estimated that approximately 25% of the global atmospheric dust burden results from anthropogenic activities, a percentage that in highly urbanized, densely populated, and industrialized regions of

developing nations, where rapid economic growth drives extensive land-use changes, anthropogenic dust entrainment into the atmosphere can comprise up to 70% of the total airborne dust load (Huang et al., 2015; Chen et al., 2019). Implementation of atmospheric aerosol models usually employ static land cover classifications of dust sources with focus on arid regions (Ginoux et al., 2001), neglecting though anthropogenic dust emissions (Ginoux et al., 2012; Huang et al., 2015; Chen et al., 2019), leading to underestimations due to the missing anthropogenic dust emissions of total dust load suspended into the atmosphere

over highly urbanized areas of the Earth (Proestakis et al., 2018; 2024; Papachristopoulou et al., 2022). In addition, implementation of passive remote sensing techniques to quantify the aerosol load and investigate the induced disorders on human health frequently leads to overestimations and ambiguities, due to the retrieval of the total column-integrated aerosol load and not the aerosol load specifically within the PBL (McGrath-Spangler and Denning, 2013; Luo et al., 2014). Moreover, the observational-based aerosol load in terms of optical depth frequently reports on the cumulative fine-mode and coarse-mode

aerosol load and not to the distinct components, hampering the capacity to decouple the induced disorders on human health to the fine-mode and coarse-mode separately (i.e., De Longueville et al., 2010; Deroubaix et al., 2013; Katra et al., 2014; Prospero et al., 2014; Querol et al., 2019).

The EO-based CDR employed in this study leverages multiple complementary observational capabilities and methodological developments enabling to tackle these challenges (Proestakis et al., 2024). First, it utilizes the capacity of CALIOP to retrieve

vertically resolved aerosol profiles (Hunt et al., 2009), which allows for accurate quantification of the aerosol load within the PBL (Zhang et al., 2006; Tsikoudi et al., 2025). Second, it exploits the distinct optical properties of non-spherical dust particles, and more specifically on the particulate depolarization ratio, for both the total dust (Gobbi et al., 2000; Sugimoto et al., 2003; Shimizu et al., 2004; Esselborn et al., 2009; Freudenthaler et al., 2009; Ansmann et al., 2011; Tesche et al., 2011; Wiegner et al., 2011; Mamouri et al., 2013; Baars et al., 2016; Veselovskii et al., 2016; Hofer et al., 2017; Filioglou et al., 2020; Floutsi

et al., 2023) and the submicrometer and supermicrometer components (Sakai et al., 2010; Järvinen et al., 2016). This facilitates the application of the two-step POLIPHON algorithm (Mamouri and Ansmann, 2014, 2017), which enables the decoupling of fine- and coarse-mode dust contributions. Third, the orbital characteristics of the CALIPSO mission (Winker et al., 2010) provide the capability to monitor the four-dimensional (spatial and temporal) distribution of atmospheric dust over multi-year periods (Amiridis et al., 2013; Marinou et al., 2017; Proestakis et al., 2018).

Collectively, these elements enable an in-depth assessment of the long-term variability, spatial distribution, and temporal evolution of the fine- and coarse-mode components of atmospheric dust within the PBL over densely populated and industrialized urban areas worldwide, as discussed in the framework of the Beijing-China megacity (Fig.11). More specifically,



expanding the Beijing-China PBL fine-mode and coarse-mode dust analysis to the entire dataset of the world's large cities and megacities (UN, 2018a, 2019) yields insight into the spatial and temporal patterns observed over the highly urbanized and

industrialized areas a global scale, shown in Figure 13. The figure is organized into three rows and two columns. Each row represents a different aerosol component: the total atmospheric load of dust (top row), the coarse-mode dust (middle row), and the fine-mode of dust (bottom row). The left column shows the mean PBL mass concentration for each dust type, while the right column illustrates the corresponding temporal trends, both for the large cities and megacities of the world and on the basis of EOs over the period from 12/2006 to 11/2022. Cities shown in (cycle) rhombus shape indicate (non) statistically

significant trends. Cities shown in rhombus shape indicate statistically significant trends.

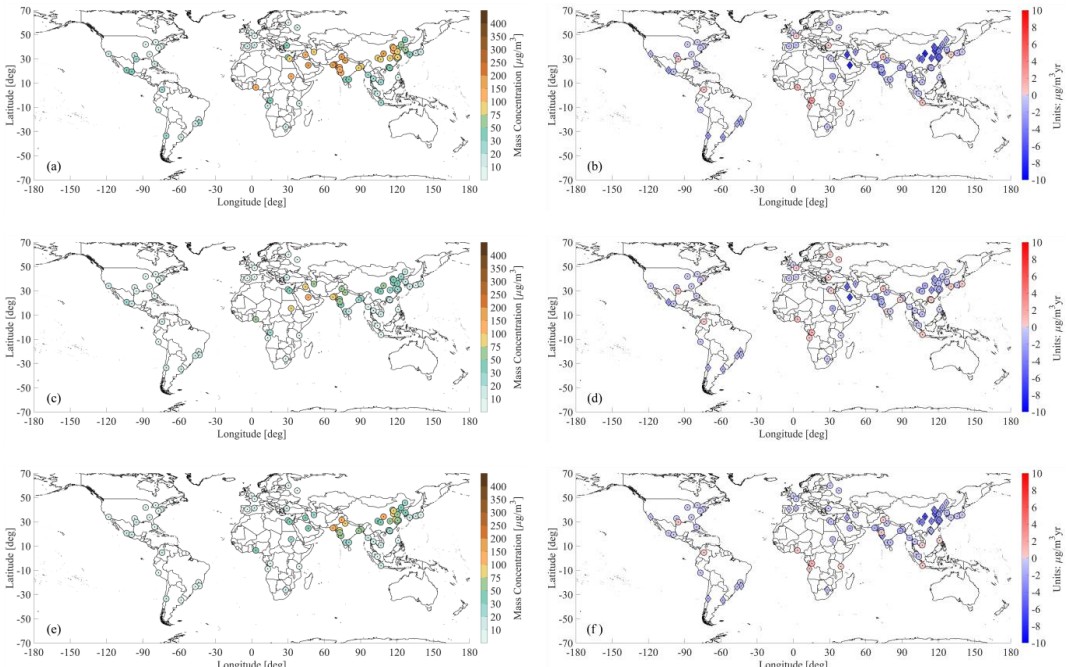

Figure 13: Dust (first row), coarse-mode dust (second row), and fine-mode dust (third row) PBL mass concentration (left column) and PBL mass concentration trends (right column) as established on the basis of CALIPSO overpasses within 100

km radius from each world-city with population greater than 5 million as reported by the United Nations (UN; 2018a, 2019) and on the basis of the temporal period DJF/2006-SON/2022.

Regarding the coarse-mode fraction of dust aerosols confined within the PBL, mass concentrations in 40 out of 81 major cities and megacities worldwide (~49.4%), with populations exceeding 5 million as of 2018 (UN, 2018a, b, 2019), surpass the WHO

annual mean AQG for annual mean $PM_{10}$ concentrations (15 µg m⁻³; Table 1). The general geographical-distribution pattern is that large cities and megacities across North America, South America, South Africa, Europe, and the Indonesia/Indochina region generally exhibit relatively low coarse-mode dust concentrations within the PBL, remaining below WHO recommended AQG for annual mean $PM_{10}$ thresholds. In contrast, large urban centers in the Middle East, the Indian subcontinent, East Asia, North Africa, and the Sahel-Central Africa region frequently record coarse-mode dust levels that significantly surpass these

limits. The ten cities with the highest PBL-level coarse-mode dust mass concentrations include Riyadh, Saudi Arabia (149.94 ± 86.68 µg m⁻³), Hyderabad, India (121.22 ± 78.35 µg m⁻³), Baghdad, Iraq (89.08 ± 57.30 µg m⁻³), Khartoum, Sudan (87.63 ± 42.68 µg m⁻³), Karachi, Pakistan (85.95 ± 55.08 µg m⁻³), Xi'an, China (72.22 ± 57.15 µg m⁻³), Surat, India (60.25 ±



64.20 µg m⁻³), Ahmedabad, India (59.79 ± 58.60 µg m⁻³), Lagos, Nigeria (59.64 ± 84.33 µg m⁻³), and Delhi, India (58.20 ± 55.09 µg m⁻³). Collectively, these ten urban areas are home to over 105 million people, whose health is adversely impacted by

elevated exposure to coarse-mode dust aerosols. An additional interesting feature is that the coarse-mode fraction of dust aerosols confined within the PBL in terms of mass concentrations, computed on the basis of EOs between 12/2006 and 11/2022, exhibit negative trends in 60 out of 81 major cities and megacities worldwide (~74.1%). Among cities exceeding WHO AQG for PM$_{10}$, statistically significant decreasing trends are observed in Beijing, Tianjin, Suzhou, Hangzhou, and Wuhan (China), Baghdad (Iraq), Riyadh (Saudi Arabia), Santiago (Chile), Guadalajara (Mexico), and Tehran (Iran). The observed decline in

airborne coarse-mode dust aerosols near the surface may result in reduced health risks for these populations, although this decreasing trend must be considered in the context of absolute mass concentration levels, which may still pose concerns. The remaining 21 out of 81 major cities and megacities worldwide (~25.9%) exhibit positive trends in the coarse-mode fraction of dust aerosols confined within the PBL, in terms of mass concentration. Although none of these urban areas display statistically significant increasing trends, indicating that the observed tendencies may lack robustness or consistency over time, such trends

must still be interpreted and considered in the context of absolute mass concentration levels. Absence of statistically significant trends in cities where coarse-mode dust concentrations within the PBL exceed WHO AQG safety thresholds for PM$_{10}$ may suggest persistent exposure levels that continue to pose health risks in the coming years, potentially reflecting a lack of effective or adequately implemented air quality management policies. Regarding the mass concentration of the coarse-mode fraction of dust aerosols confined within the PBL, analytical description for each of the large city or megacity across the world as reported

by the UN (UN, 2018a, 2019), is provided in table 4.

With respect to the fine-mode fraction of dust aerosols confined within the PBL, mass concentrations in 71 out of 81 major cities and megacities worldwide (~87.7%), each with populations exceeding 5 million as of 2018 (UN, 2018a, b, 2019), exceed the WHO annual mean AQG for annual mean PM$_{2.5}$ concentrations (5 µg m⁻³; Table 1). Unlike the more regionally in terms of continents concentrated pattern observed for coarse-mode dust, exceedances of WHO-recommended AQG for PM$_{2.5}$

thresholds related to fine-mode dust are observed in major urban areas distributed globally, indicating a more widespread and pervasive issue. The ten cities with the highest PBL-level fine-mode dust mass concentrations include Karachi, Pakistan (113.49 ± 43.77 µg m⁻³), Xian, China (104.40 ± 77.67 µg m⁻³), Lahore, Pakistan (101.39 ± 43.39 µg m⁻³), Hyderabad, India (98.68 ± 36.45 µg m⁻³), Delhi, India (98.14 ± 42.44 µg m⁻³), Jinan, China (91.05 ± 47.74 µg m⁻³), Bombay, India (85.01 ± 50.26 µg m⁻³), Ahmedabad, India (82.92 ± 57.32 µg m⁻³), Nanjing, China (79.58 ± 49.10 µg m⁻³), and Tianjin, China (76.17 ±

43.92 µg m⁻³). Collectively, these ten urban areas are home to over 123 million people, whose health is adversely and highly impacted by elevated exposure to the inhalable fine-mode component of dust aerosols. An additional interesting feature is that the fine-mode fraction of dust aerosols confined within the PBL in terms of mass concentrations, computed on the basis of EOs between 12/2006 and 11/2022, exhibit negative trends in 70 out of 81 major cities and megacities worldwide (~86.4%). Among cities that exceed the WHO AQG for PM$_{2.5}$, statistically significant decreasing trends have been observed in Bombay

and Pune (India), Baghdad (Iraq), Santiago (Chile), Barcelona and Madrid (Spain), Johannesburg (South Africa), Sao Paulo and Rio de Janeiro (Brazil), Buenos Aires (Argentina), Los Angeles (USA), and several cities in China, including Shanghai, Beijing, Chongqing, Tianjin, Shenzhen, Chengdu, Nanjing, Hong Kong, Dongguan, Foshan, Xian, Hangzhou, Shenyang, Haerbin, Suzhou, Qingdao, Dalian, and Jinan. The observed reduction in airborne fine-mode dust aerosols within the PBL may lead to a decrease in health risks for these populations. However, this declining trend should be evaluated in the context of the

absolute mass concentration levels, which may still present potential concerns, in the case of exceeding WHO AQG safety thresholds for PM$_{2.5}$. The remaining 10 out of 81 major cities and megacities worldwide (~12.3%) exhibit positive trends in the fine-mode fraction of dust aerosols confined within the PBL, in terms of mass concentration, however only the urban case of Luanda-Angola is characterized by statistically significant increasing tendency. In addition, in this city case, the PBL fine-mode mass concentration of 7.02 ± 7.7 µg m⁻³ resides within the WHO annual mean AQG safety thresholds for PM$_{2.5}$. It should

be emphasized that absence of statistically significant trends in cities where fine-mode dust concentrations within the PBL




exceed WHO AQG safety thresholds for PM$_{2.5}$ reflects persistent exposure levels that continue to pose health risks in the coming years, potentially implying a lack of effective or adequately implemented air quality management policies. Regarding the mass concentration of the fine-mode fraction of dust aerosols confined within the PBL, analytical description for each of the large city or megacity across the world as reported by the UN (UN, 2018a, 2019), is provided in table 5.

It is essential to emphasize that in order to approach the scientific question on the multifaceted role of airborne dust as an environmental risk factor for human health, requirement is the simultaneous interpretation of (i) the coarse-mode and fine-mode components of particulate matter (PM) from atmospheric dust within PBL, and (ii) their respective increasing or decreasing trends, in relation to the population size of large cities and megacities globally. To this end, Figure 14 presents the mean mass concentrations of coarse-mode and fine-mode dust PM within the PBL across major urban centers as a function of

population (left panel), along with the corresponding temporal trends derived from satellite EOs over the period 12/2006 to 11/2022 (right panel). Cities represented by rhombus symbols indicate statistically significant trends, whereas circular symbols denote non-significant trends. Moreover, the light-orange shaded regions in the upper left and lower right quadrants of both the left and the right panels indicate urban areas where WHO annual mean AQG are exceeded for PM$_{2.5}$ and PM$_{10}$, respectively. The red-shaded area highlights cities where both PM$_{2.5}$ and PM$_{10}$ concentrations surpass the WHO AQG.

It is estimated that out of approximately 807.9 million people residing in the world's 81 largest cities and megacities (average population, 2007–2022; UN, 2018a, 2019), the dust hazard consisted to a greater or lesser degree an environmental risk factor for the health of ~701.4 millions of people. More specifically, in these 81 large cities and megacities of the world:

- approximately 91.7 million live in urban areas where both coarse-mode and fine-mode PM levels are below the WHO AQG for PM$_{2.5}$ and PM$_{10}$, respectively.

- approximately 310.8 million live in cities where either the fine-mode or coarse-mode dust PM component exceeds the respective WHO AQG for PM$_{2.5}$ and PM$_{10}$, respectively.

- approximately 390.6 million reside in areas where both coarse-mode and fine-mode PM concentrations exceed the WHO AQG for PM$_{2.5}$ and PM$_{10}$, respectively.

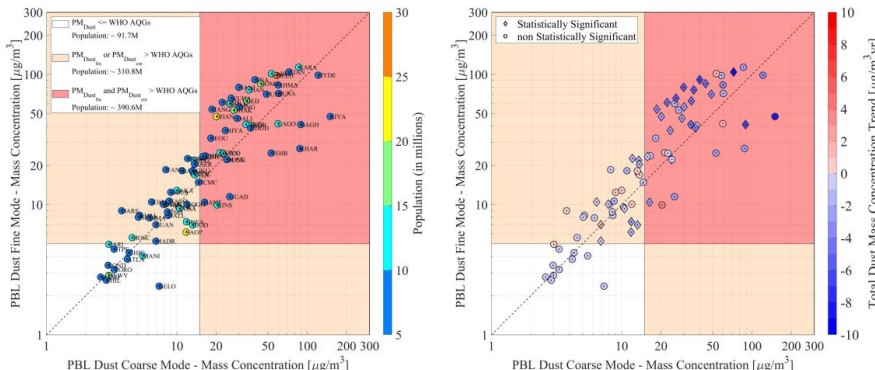

Figure 14: (left) Large cities and megacities dust coarse-mode and fine-mode mean mass concentrations within the PBL and their population. Colored areas delineate WHO annual mean AQG for PM$_{10}$ and PM$_{2.5}$. (right) Increasing/decreasing trends of dust coarse-mode and fine-mode mass concentration of large cities and megacities within the PBL. Cities shown as diamonds (cycles) correspond to (non) statistically significant trends. The analysis is based on CALIOP observations of the temporal period 2007-2022.






**5. Atmospheric dust and Air Quality - insight into the third decade of the 21$^{st}$ century**

Based on sixteen years of EOs (December 2006–November 2022), this study has: (i) quantified the mass concentration of the

coarse-mode and fine-mode atmospheric dust components within the PBL, where the main anthropogenic activity takes place; (ii) identified increasing and decreasing temporal trends on the observational datasets and products; and (iii) considered the statistical significance of these trends across the world's largest urban centers. This section aims to extend the analysis by addressing the following scientific questions: How are the mass concentrations of coarse-mode and fine-mode atmospheric dust within the PBL expected to evolve in the near future? Which highly industrialized and densely populated large cities and

megacities worldwide are projected to exceed the WHO AQG thresholds for PM$_{10}$ and PM$_{2.5}$, respectively, due to anticipated changes in the PBL dust load of the coarse-mode and fine-mode components of atmospheric dust? Addressing these questions on the atmospheric dust hazard is essential for anticipating future air quality challenges, supporting the development of targeted mitigation strategies, and protecting public health in rapidly growing urban environments.

As a representative case for the projection analysis, the megacity of Beijing, China, is utilized. More specifically, figure 15

presents the seasonal mean (denoted by white cycles) of EO-derived PM$_{coarse}$ (Fig.15a) and PM$_{fine}$ (Fig.15b) components of total atmospheric dust load within the PBL, based on CALIPSO overpasses within a 100 km radius of the Beijing metropolitan area. Over the 16-year period from December 2006 to November 2022 (highlighted by the light blue shaded area), the mean mass concentrations of coarse-mode and fine-mode atmospheric dust within the PBL were quantified at 34.20 ± 24.05 μg m$^{-3}$ (Fig.15a) and 62.62 ± 36.44 μg m$^{-3}$ (Fig.15b), respectively. Both values exceed the WHO's annual-mean AQG thresholds

(Table 1). Furthermore, statistically significant decreasing trends were observed in both fine-mode and coarse-mode dust concentrations, as derived from EO data. These negative tendencies are corroborated by ground-based sunphotometer AERONET measurements, specifically the SDA-derived fine-mode and coarse-mode AOT at 500 nm, timeseries established utilizing long-term data from the Beijing, Beijing-RADI, Beijing-PKU, and Beijing-CAMS stations (Fig.10; Table 3). Moreover, based on the 2007–2022 observational timeframe and the derived trends, future projections of PBL coarse-mode

and fine-mode dust mass concentrations beyond the observation period are presented. In figure 15, these projections are depicted as red trend lines, with the surrounding light red shaded areas denoting the associated confidence intervals. It has to be emphasized that the projections represent expected PBL dust concentrations under the assumption of continuation of 2007–2022 trends. However, the reliability of these projections decreases with increasing time beyond the observational temporal period. To account for increasing uncertainty, the projection period is constrained to the point at which (i) the confidence

interval first intersects zero (yielding non-physical negative concentration values), or (ii) the confidence interval variability equals or exceeds the variability characterizing the original EO dataset, (iii) the confidence interval reaches values as high as the projected values themselves. For the indicative case of the Beijing metropolitan area, projections of the PBL coarse-mode and fine-mode dust mass concentration reports with relatively high confidence values as high as 14.67 ± 14.86 μg m$^{-3}$ (2026) and 28.04 ± 28.05 μg m$^{-3}$ (2029), respectively, corresponding to reduction of ~57.1% and ~55.2% for the two dust aerosol

mode components.

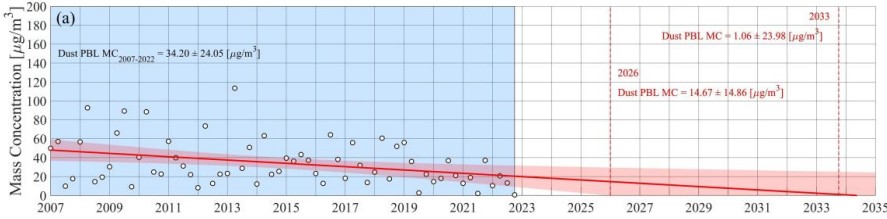





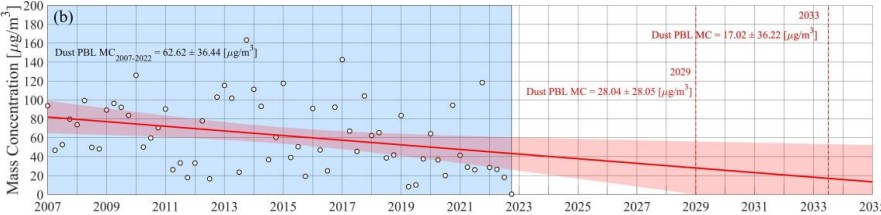

Figure 15: Beijing megacity dust coarse-mode (a) and fine-mode (b) mass concentration within the PBL. Cycles in white colour correspond to seasonal mean mass concentration values for the CALIPSO period 2007-2022 (shaded area in light blue).

Projection into the temporal period beyond 2022 and up to the year 2035 is shown in terms of linear regression extension and 95% confidence interval for predictions.

Regarding the coarse-mode fraction of dust aerosols confined within the PBL, mass concentrations in 38 out of 81 major cities and megacities worldwide (~46.9%), with populations exceeding 5 million as of 2018 (UN, 2018a, b, 2019), surpass the WHO

annual mean AQG for annual mean $PM_{10}$ concentrations (15 µg m$^{-3}$; Table 1). The ten cities with the highest PBL-level coarse-mode dust mass concentrations in the years following the EO-based temporal period include Hyderabad, India ($86.83 \pm 78.37$ µg m$^{-3}$; 2032), Lagos, Nigeria ($77.64 \pm 77.70$ µg m$^{-3}$; 2031), Lahore, Pakistan ($70.78 \pm 40.26$ µg m$^{-3}$; 2032), Guangzhou, China ($67.72 \pm 54.31$ µg m$^{-3}$; 2033), Karachi, Pakistan ($61.31 \pm 55.08$ µg m$^{-3}$; 2032), Riyadh, Saudi Arabia ($61.23 \pm 60.07$ µg m$^{-3}$; 2027), Khartoum, Sudan ($60.24 \pm 42.94$ µg m$^{-3}$; 2033), Dongguan, China ($56.96 \pm 56.96$ µg m$^{-3}$; 2032), Kinshasa, Democratic

Republic of Congo ($50.21 \pm 46.80$ µg m$^{-3}$; 2032), and Xian, China ($47.83 \pm 48.12$ µg m$^{-3}$; 2029). Collectively, these ten urban areas in the year of projection it is foreseen to be home to over 160.6 million people, whose health will be adversely impacted by elevated exposure to coarse-mode dust aerosols. It is projected that, in the coming years, the number of large cities and megacities exhibiting coarse-mode dust aerosol concentrations within the PBL that (i) exceed the WHO 24-hour mean $PM_{10}$ AQG safety thresholds will decrease from 13 to 11, and (ii) the number of case that fall below the WHO annual mean $PM_{10}$

AQG safety thresholds will increase from 41 to 43. In the cases of Delhi, India (-25.6%; 2028), Tehran, Iran (-48.7%; 2027; statistically significant trend), Baghdad, Iraq (-62%; 2025; statistically significant trend), Pune (-18.6%; 2029) and Surat (-30.1%; 2026) in India the PBL coarse-mode dust concentration is foreseen to fall from exceeding WHO 24-hour mean AQG to exceeding the annual mean AQG for $PM_{10}$, with reference the 2007-2022 temporal period. Opposite is the tendency of the large urban areas of Guangzhou, China (216%; 2033), Kinshasa, Democratic Republic of Congo (146.2%; 2032), and

Dongguan, China (136%, 2032) where the increase above the WHO 24-hour mean $PM_{10}$ AQG safety thresholds is projected. Regarding the mass concentration of the coarse-mode fraction of dust aerosols confined within the PBL in the temporal period beyond the observational period, analytical description for each of the large city or megacity across the world as reported by the UN (UN, 2018a, 2019), is provided in table 4.

Table 4: Analytical table of 81 cities (first column, abbreviations second column) with the highest population up to 2018 (UN, 2018a, 2019). The table provides basic statistical analysis outcomes reporting on PBL Dust coarse-mode mass concentration values based on the period 2007-2022, including mean $\pm$ sd (µg m$^{-3}$), increasing/decreasing tendency, and the corresponding statistical significance of the timeseries, in addition to their mean 2007-2022 population. The columns on the right provide insight beyond the EO-based period 2007-2022, including the year when the linear regression extension and the 95%

confidence interval for predictions deviates into lower confidence values and the UN projected population on that specific year. Color coding and exceedances on $PM_{10}$.: Orange color: WHO annual mean AQGs / Red color: WHO 24-hour AQGs.

| Cities and General Information | PBL Dust coarse-mode and Air Quality EO-based period 2007-2022 | PBL Dust coarse-mode and Air Quality Insight beyond the EO-based period 2007-2022 |
| --- | --- | --- |



| City Name | City Acronym | PBL Dust MC 2007-2022 mean ± sd (µg/m³) | Population 2007-2022 (millions) | Trend (µg/m³y) | Statistical Significance | PBL Dust MC future projection mean ± sd (µg/m³) | Year | Population (millions) |
|---|---|---|---|---|---|---|---|---|
| Tokyo | TOKY | 10.34 ± 10.21 | 37.07 | 0.13 | no | 12.57 ± 10.27 | 2032 | 36.36 |
| Delhi | DELH | 58.20 ± 55.09 | 25.68 | -1.09 | no | 43.29 ± 43.04 | 2028 | 37.22 |
| Shanghai | SHAN | 20.14 ± 18.01 | 23.28 | -0.75 | no | 11.64 ± 11.83 | 2026 | 31.05 |
| Mexico City | MEXC | 13.40 ± 10.99 | 20.98 | -0.42 | no | 7.96 ± 8.07 | 2027 | 23.29 |
| Sao Paulo | SAOP | 11.78 ± 11.71 | 20.73 | -0.82 | yes | 0.24 ± 9.00 | 2029 | 23.67 |
| Bombay | BOMB | 43.91 ± 37.71 | 19.23 | -0.87 | no | 31.64 ± 31.31 | 2029 | 24.04 |
| Osaka | OSAK | 8.06 ± 8.24 | 19.22 | -0.11 | no | 6.60 ± 6.56 | 2028 | 18.77 |
| Cairo | CAIR | 37.21 ± 20.94 | 18.69 | 0.36 | no | 43.57 ± 21.02 | 2032 | 26.63 |
| New York | NEWY | 3.00 ± 2.78 | 18.59 | -0.12 | no | 1.70 ± 1.72 | 2025 | 19.15 |
| Beijing | BEIJ | 34.20 ± 24.05 | 18.13 | -1.76 | yes | 14.67 ± 14.86 | 2026 | 22.98 |
| Dhaka | DHAK | 27.29 ± 27.70 | 17.51 | -0.71 | no | 19.04 ± 18.81 | 2026 | 25.36 |
| Hyderabad | HYDE | 121.22 ± 78.35 | 16.57 | -1.92 | no | 86.83 ± 78.37 | 2032 | 24.41 |
| Karachi | KARA | 85.95 ± 55.08 | 14.18 | -1.38 | no | 61.31 ± 55.08 | 2032 | 21.46 |
| Buenos Aires | BUEA | 11.84 ± 14.10 | 14.63 | -1.22 | yes | 0.41 ± 7.36 | 2024 | 15.62 |
| Kolkata | KOLK | 25.49 ± 27.98 | 14.39 | 0.08 | no | 26.88 ± 26.87 | 2031 | 17.97 |
| Istanbul | ISTA | 13.71 ± 13.37 | 13.79 | 0.40 | no | 20.89 ± 13.33 | 2032 | 17.47 |
| Chongqing | CHON | 33.87 ± 43.17 | 13.29 | -1.19 | no | 23.28 ± 23.36 | 2023 | 17.34 |
| Rio de Janeiro | RIOD | 13.22 ± 11.15 | 12.87 | -0.90 | yes | 5.44 ± 5.51 | 2023 | 13.72 |
| Manila | MANI | 5.47 ± 8.54 | 12.77 | -0.15 | no | 4.27 ± 4.34 | 2023 | 14.66 |
| Los Angeles | LOSA | 10.49 ± 7.76 | 12.30 | -0.31 | no | 6.10 ± 6.18 | 2029 | 13.09 |
| Lagos | LAGO | 59.64 ± 84.33 | 12.19 | 1.12 | no | 77.64 ± 77.70 | 2031 | 21.34 |
| Moscow | MOSC | 4.55 ± 6.22 | 11.95 | 0.05 | no | 5.34 ± 5.36 | 2030 | 12.79 |
| Tianjin | TIAN | 35.74 ± 20.54 | 11.87 | -1.43 | yes | 15.53 ± 15.75 | 2029 | 15.56 |
| Guangzhou | GUAN | 21.39 ± 54.55 | 11.62 | 2.56 | no | 67.72 ± 54.31 | 2033 | 16.49 |
| Kinshasa | KINS | 20.39 ± 46.52 | 11.58 | 1.67 | no | 50.21 ± 46.80 | 2032 | 23.75 |
| Shenzhen | SHNZ | 13.54 ± 11.43 | 11.12 | 0.11 | no | 15.44 ± 11.50 | 2032 | 14.84 |
| Paris | PARI | 3.01 ± 3.19 | 10.70 | 0.05 | no | 3.90 ± 3.21 | 2032 | 11.85 |
| Lahore | LAHO | 53.19 ± 40.24 | 10.30 | 0.98 | no | 70.78 ± 40.26 | 2032 | 17.74 |
| Jakarta | JAKA | 9.99 ± 14.17 | 10.13 | 0.42 | no | 17.15 ± 14.11 | 2031 | 12.89 |
| Bangalore | BALO | 22.57 ± 36.50 | 10.08 | 0.31 | no | 26.42 ± 26.42 | 2027 | 15.14 |
| Seoul | SEOU | 18.22 ± 17.75 | 9.89 | -0.31 | no | 14.01 ± 13.89 | 2028 | 10.11 |
| Lima | LIMA | 6.16 ± 7.88 | 9.73 | -0.38 | no | 0.13 ± 7.07 | 2031 | 12.41 |
| Chennai | CHNA | 16.53 ± 29.27 | 9.61 | -0.19 | no | 15.01 ± 15.04 | 2023 | 11.77 |
| Bogota | BOGO | 11.93 ± 46.63 | 9.58 | 0.40 | no | 18.85 ± 46.85 | 2035 | 12.52 |
| Nagoya | NAGO | 8.73 ± 10.52 | 9.31 | -0.62 | yes | 0.21 ± 7.93 | 2028 | 9.46 |
| Bangkok | BANG | 10.96 ± 14.93 | 9.30 | -0.65 | no | 6.67 ± 6.70 | 2021 | 10.72 |
| Chicago | CHIC | 4.31 ± 4.26 | 8.74 | -0.06 | no | 3.47 ± 3.50 | 2029 | 9.34 |
| London | LOND | 2.97 ± 3.92 | 8.61 | -0.07 | no | 2.29 ± 2.27 | 2024 | 9.75 |
| Tehran | TEHR | 52.83 ± 37.58 | 8.54 | -2.00 | yes | 27.09 ± 27.16 | 2027 | 9.95 |
| Chengdu | CHGD | 36.75 ± 44.14 | 8.21 | -1.26 | no | 24.93 ± 24.93 | 2024 | 9.83 |
| Wuhan | WUHA | 26.08 ± 26.67 | 7.90 | -2.28 | yes | 0.70 ± 16.07 | 2026 | 9.12 |
| Nanjing | NANJ | 29.84 ± 22.49 | 7.35 | -0.93 | no | 17.22 ± 17.32 | 2028 | 10.73 |
| Hong Kong | HONG | 12.08 ± 11.36 | 7.25 | -0.14 | no | 9.91 ± 9.93 | 2030 | 7.99 |
| Ho Chi Minh City | HCMC | 14.65 ± 19.63 | 7.27 | -0.66 | no | 3.21 ± 19.76 | 2032 | 11.54 |
| Dongguan | DONG | 24.14 ± 58.44 | 7.14 | 1.88 | no | 56.96 ± 56.96 | 2032 | 8.45 |
| Ahmedabad | AHMA | 59.79 ± 58.63 | 7.06 | -1.01 | no | 46.00 ± 45.86 | 2028 | 9.71 |
| Foshan | FOSH | 15.78 ± 15.98 | 6.88 | 0.72 | no | 28.85 ± 15.94 | 2033 | 8.59 |
| Luanda | LUAN | 6.91 ± 9.10 | 6.73 | 0.43 | no | 14.73 ± 9.05 | 2033 | 13.52 |
| Kuala Lumpur | KUAL | 5.18 ± 12.77 | 6.79 | -0.12 | no | 3.07 ± 12.74 | 2032 | 10.08 |
| Xian | XIAN | 72.22 ± 57.15 | 6.62 | -1.64 | no | 47.83 ± 48.12 | 2029 | 9.86 |
| Hangzhou | HANG | 18.58 ± 20.67 | 6.57 | -1.13 | yes | 0.37 ± 18.29 | 2031 | 9.36 |
| Santiago | SANT | 16.33 ± 14.43 | 6.50 | -1.31 | yes | 0.42 ± 9.25 | 2027 | 7.09 |
| Shenyang | SHYA | 23.55 ± 16.98 | 6.46 | -0.36 | no | 17.19 ± 17.02 | 2032 | 8.75 |
| Baghdad | BAGH | 89.08 ± 57.30 | 6.30 | -5.08 | yes | 33.82 ± 33.63 | 2025 | 8.14 |
| Madrid | MADR | 6.90 ± 8.93 | 6.17 | -0.36 | no | 0.36 ± 8.94 | 2033 | 6.95 |
| Riyadh | RIYA | 149.94 ± 86.68 | 6.13 | -6.89 | yes | 61.23 ± 60.07 | 2027 | 8.20 |
| Toronto | TORO | 3.31 ± 3.00 | 5.81 | -0.01 | no | 3.13 ± 2.98 | 2032 | 6.91 |
| Miami | MIAM | 5.07 ± 5.97 | 5.79 | -0.18 | no | 3.37 ± 3.37 | 2024 | 12.79 |
| Belo Horizonte | BELO | 7.31 ± 7.71 | 5.72 | -0.528 | yes | 3.42 ± 3.47 | 2022 | 6.19 |
| Pune | PUNE | 48.60 ± 45.23 | 5.70 | -0.60 | no | 39.54 ± 39.52 | 2029 | 8.25 |
| Haerbin | HAER | 13.65 ± 10.40 | 5.69 | -0.49 | no | 7.44 ± 7.44 | 2027 | 7.31 |
| Dallas | DALL | 8.56 ± 7.26 | 5.66 | -0.107 | no | 6.78 ± 6.82 | 2031 | 7.14 |
| Surat | SURA | 60.25 ± 64.20 | 5.65 | -1.59 | no | 42.13 ± 42.70 | 2026 | 8.82 |
| Houston | HOUS | 8.98 ± 11.12 | 5.60 | 0.51 | no | 18.28 ± 11.08 | 2033 | 7.45 |
| Philadelphia | PHIL | 2.88 ± 2.53 | 5.57 | 0.029 | no | 3.41 ± 2.54 | 2032 | 6.22 |
| Fukuoka | FUKU | 13.16 ± 13.93 | 5.53 | 0.34 | no | 19.17 ± 13.94 | 2032 | 5.36 |
| Singapore | SING | 8.47 ± 23.24 | 5.46 | -0.22 | no | 4.48 ± 23.38 | 2032 | 5.63 |
| Suzhou | SUZH | 22.29 ± 24.87 | 5.34 | -1.359 | yes | 0.36 ± 22.00 | 2031 | 9.49 |
| Barcelona | BARC | 7.88 ± 7.85 | 5.23 | -0.096 | no | 6.50 ± 6.45 | 2029 | 5.79 |
| Saint Petersburg | STPE | 3.30 ± 4.10 | 5.15 | 0.087 | no | 4.87 ± 4.10 | 2032 | 5.64 |
| Dar es Salaam | DARE | 3.77 ± 4.42 | 5.12 | -0.004 | no | 3.71 ± 3.70 | 2029 | 10.31 |
| Atlanta | ATLA | 4.17 ± 3.92 | 5.11 | -0.087 | no | 3.00 ± 3.01 | 2028 | 6.48 |



| Khartoum | KHAR | 87.63 ± 42.68 | 5.10 | -1.51 | no | 60.24 ± 42.94 | 2033 | 8.94 |
| Qingdao | QING | 30.05 ± 19.35 | 4.99 | -0.52 | no | 20.82 ± 19.34 | 2032 | 6.82 |
| Johannesburg | JOHA | 6.41 ± 6.47 | 4.94 | -0.075 | no | 5.33 ± 5.32 | 2029 | 6.88 |
| Washington, D.C. | WASH | 2.60 ± 2.85 | 4.93 | -0.029 | no | 2.20 ± 2.19 | 2028 | 5.76 |
| Dalian | DALI | 28.92 ± 33.12 | 4.82 | -0.96 | no | 19.40 ± 19.51 | 2024 | 6.22 |
| Yangon | YANG | 8.23 ± 7.99 | 4.81 | -0.086 | no | 6.97 ± 6.96 | 2029 | 6.26 |
| Guadalajara | GUAD | 25.41 ± 32.94 | 4.77 | -2.32 | yes | 0.72 ± 19.66 | 2025 | 5.78 |
| Alexandria | ALEX | 24.24 ± 19.58 | 4.76 | -0.008 | no | 24.11 ± 19.46 | 2032 | 6.69 |
| Jinan | JINA | 39.65 ± 27.17 | 4.58 | -1.24 | no | 21.75 ± 21.86 | 2029 | 6.47 |

Regarding the fine-mode fraction of dust aerosols confined within the PBL, mass concentrations in 38 out of 81 major cities and megacities worldwide (~46.9%), with populations exceeding 5 million as of 2018 (UN, 2018a, b, 2019), surpass the WHO
annual mean AQG for annual mean PM$_{2.5}$ concentrations (5 µg m$^{-3}$; Table 1). The ten cities with the highest PBL-level fine-mode dust mass concentrations in the years following the EO-based temporal period include Lahore, Pakistan (103.41 ± 43.12 µg m$^{-3}$; 2032), Karachi, Pakistan (93.50 ± 43.76 µg m$^{-3}$; 2032), Delhi, India (84.71 ± 42.59 µg m$^{-3}$; 2032), Hyderabad, India (78.43 ± 36.32 µg m$^{-3}$; 2032), Surat, India (75.46 ± 40.93 µg m$^{-3}$; 2032), Lagos, Nigeria (51.48 ± 51.66 µg m$^{-3}$; 2031), Ahmedabad, India (50.65 ± 50.21 µg m$^{-3}$; 2030), Xian, China (41.60 ± 40.55 µg m$^{-3}$; 2024), Bombay, India (39.73 ± 40.03 µg
m$^{-3}$; 2029), and Dhaka, Bangladesh (37.18 ± 37.02 µg m$^{-3}$; 2031). Collectively, these ten urban areas in the year of projection it is foreseen to be home to over 207.7 million people, whose health will be adversely impacted by elevated exposure to fine-mode dust aerosols. It is projected that, in the coming years, the number of large cities and megacities exhibiting fine-mode dust aerosol concentrations within the PBL that (i) exceed the WHO 24-hour mean PM$_{2.5}$ AQG safety thresholds will decrease from 45 to 36, and (ii) that the number of cases that exceed WHO annual mean PM$_{2.5}$ AQG though fall below the 24-hour
mean AQG safety thresholds will decrease from 25 to 24. Regarding the mass concentration of the fine-mode fraction of dust aerosols confined within the PBL in the temporal period beyond the observational period, analytical description for each of the large city or megacity across the world as reported by the UN (UN, 2018a, 2019), is provided in table 5.

Table 5: Analytical table of 81 cities (first column, abbreviations second column) with the highest population up to 2018 (UN,
2018a, 2019). The table provides basic statistical analysis outcomes reporting on PBL dust fine-mode mass concentration values based on the period 2007-2022, including mean ± sd (µg m$^{-3}$), increasing/decreasing tendency, and the corresponding statistical significance of the timeseries, in addition to their mean 2007-2022 population. The columns on the right provide insight beyond the EO-based period 2007-2022, including the year when the linear regression extension and the 95% confidence interval for predictions deviates into lower confidence values and the UN projected population on that specific
year. Color coding and exceedances on PM$_{2.5}$.: Orange color: WHO annual mean AQGs / Red color: WHO 24-hour AQGs.

| Cities and General Information | | PBL Dust fine-mode and Air Quality EO-based period 2007-2022 | | | | PBL Dust fine-mode and Air Quality Insight beyond the EO-based period 2007-2022 | | |
|---|---|---|---|---|---|---|---|---|
| City Name | City Acronym | PBL Dust MC 2007-2022 mean ± sd (µg/m$^3$) | Population 2007-2022 (millions) | Trend (µg/m$^3$y) | Statistical Significance | PBL Dust MC future projection mean ± sd (µg/m$^3$) | Year | Population (millions) |
| Tokyo | TOKY | 9.55 ± 7.96 | 37.07 | -0.134 | no | 7.35 ± 7.36 | 2031 | 36.46 |
| Delhi | DELH | 98.14 ± 42.44 | 25.68 | -0.752 | no | 84.71 ± 42.59 | 2032 | 40.68 |
| Shanghai | SHAN | 47.50 ± 32.55 | 23.28 | -1.778 | yes | 24.16 ± 23.89 | 2028 | 32.04 |
| Mexico City | MEXC | 17.40 ± 11.75 | 20.98 | -0.09 | no | 15.79 ± 11.82 | 2032 | 24.65 |
| Sao Paulo | SAOP | 6.15 ± 6.19 | 20.73 | -0.575 | yes | 0.17 ± 3.46 | 2025 | 22.99 |
| Bombay | BOMB | 85.01 ± 50.26 | 19.23 | -3.22 | yes | 39.73 ± 40.03 | 2029 | 24.04 |
| Osaka | OSAK | 10.04 ± 12.41 | 19.22 | -0.245 | no | 7.56 ± 7.50 | 2025 | 18.92 |
| Cairo | CAIR | 40.72 ± 15.74 | 18.69 | -0.464 | no | 32.43 ± 15.70 | 2032 | 26.63 |
| New York | NEWY | 2.87 ± 2.42 | 18.59 | -0.065 | no | 1.95 ± 1.94 | 2029 | 19.78 |
| Beijing | BEIJ | 62.62 ± 36.44 | 18.13 | -2.448 | yes | 28.04 ± 28.05 | 2029 | 23.99 |
| Dhaka | DHAK | 53.10 ± 40.00 | 17.51 | -0.976 | no | 37.18 ± 37.02 | 2031 | 28.73 |
| Hyderabad | HYDE | 98.68 ± 36.45 | 16.57 | -1.132 | no | 78.43 ± 36.32 | 2032 | 24.41 |
| Karachi | KARA | 113.49 ± 43.77 | 14.18 | -1.118 | no | 93.50 ± 43.76 | 2032 | 21.46 |
| Buenos Aires | BUEA | 7.39 ± 8.96 | 14.63 | -1.075 | yes | - | - | - |
| Kolkata | KOLK | 59.70 ± 39.43 | 14.39 | -1.416 | no | 36.63 ± 36.30 | 2031 | 17.97 |
| Istanbul | ISTA | 16.92 ± 8.87 | 13.79 | -0.193 | no | 13.46 ± 8.88 | 2032 | 17.47 |
| Chongqing | CHON | 23.68 ± 19.52 | 13.29 | -3.251 | yes | 1.09 ± 32.43 | 2027 | 18.85 |





| | | | | | | | |
|---|---|---|---|---|---|---|---|
| Rio de Janeiro | RIOD | 6.98 ± 6.79 | 12.87 | -0.622 | yes | 0.22 ± 3.96 | 2025 | 13.92 |
| Manila | MANI | 4.04 ± 3.89 | 12.77 | 0.131 | no | 6.37 ± 3.87 | 2032 | 17.54 |
| Los Angeles | LOSA | 9.32 ± 5.47 | 12.30 | -0.379 | yes | 4.14 ± 4.07 | 2028 | 12.98 |
| Lagos | LAGO | 41.87 ± 54.47 | 12.19 | 0.577 | no | 51.48 ± 51.66 | 2031 | 21.33 |
| Moscow | MOSC | 5.58 ± 4.52 | 11.95 | -0.159 | no | 3.45 ± 3.44 | 2028 | 12.78 |
| Tianjin | TIAN | 76.17 ± 43.92 | 11.87 | -3.869 | yes | 28.28 ± 28.86 | 2027 | 15.15 |
| Guangzhou | GUAN | 25.07 ± 22.89 | 11.62 | -1.144 | no | 13.48 ± 13.51 | 2025 | 14.87 |
| Kinshasa | KINS | 9.93 ± 37.10 | 11.58 | 1.505 | no | 36.15 ± 36.19 | 2032 | 23.74 |
| Shenzhen | SHNZ | 21.90 ± 18.66 | 11.12 | -1.268 | yes | 10.00 ± 10.09 | 2024 | 13.31 |
| Paris | PARI | 4.96 ± 3.51 | 10.70 | -0.046 | no | 4.13 ± 3.52 | 2032 | 11.85 |
| Lahore | LAHO | 101.39 ± 43.39 | 10.30 | 0.114 | no | 103.41 ± 43.12 | 2032 | 17.74 |
| Jakarta | JAKA | 12.87 ± 13.57 | 10.13 | 0.503 | no | 21.62 ± 13.62 | 2032 | 13.09 |
| Bangalore | BALO | 24.84 ± 23.92 | 10.08 | -0.436 | no | 18.89 ± 18.80 | 2028 | 15.5 |
| Seoul | SEOU | 32.36 ± 31.17 | 9.89 | -0.834 | no | 22.03 ± 22.28 | 2027 | 10.07 |
| Lima | LIMA | 7.92 ± 8.64 | 9.73 | -0.286 | no | 5.12 ± 5.09 | 2024 | 11.36 |
| Chennai | CHNA | 23.68 ± 19.52 | 9.61 | -0.742 | no | 14.30 ± 14.18 | 2027 | 12.91 |
| Bogota | BOGO | 10.07 ± 38.76 | 9.58 | 0.42 | no | 17.33 ± 38.93 | 2032 | 12.52 |
| Nagoya | NAGO | 10.66 ± 13.24 | 9.31 | -0.423 | no | 7.00 ± 6.98 | 2023 | 9.56 |
| Bangkok | BANG | 18.21 ± 16.95 | 9.30 | -0.718 | no | 10.78 ± 10.67 | 2025 | 11.39 |
| Chicago | CHIC | 4.28 ± 2.82 | 8.74 | -0.082 | no | 2.81 ± 2.82 | 2032 | 9.59 |
| London | LOND | 3.43 ± 3.27 | 8.61 | -0.079 | no | 2.41 ± 2.43 | 2027 | 10.01 |
| Tehran | TEHR | 24.88 ± 13.99 | 8.54 | -0.403 | no | 17.67 ± 13.96 | 2032 | 10.41 |
| Chengdu | CHGD | 38.84 ± 31.71 | 8.21 | -2.63 | yes | 15.47 ± 15.96 | 2023 | 9.65 |
| Wuhan | WUHA | 65.65 ± 49.31 | 7.90 | -3.21 | no | 30.73 ± 30.28 | 2025 | 8.98 |
| Nanjing | NANJ | 79.58 ± 49.10 | 7.35 | -4.14 | yes | 31.42 ± 30.81 | 2026 | 10.38 |
| Hong Kong | HONG | 22.59 ± 17.46 | 7.25 | -1.098 | yes | 10.65 ± 10.76 | 2025 | 7.77 |
| Ho Chi Minh City | HCMC | 14.88 ± 32.37 | 7.27 | 0.005 | no | 14.96 ± 32.53 | 2031 | 11.29 |
| Dongguan | DONG | 22.19 ± 21.60 | 7.14 | -1.77 | yes | 0.62 ± 14.21 | 2027 | 7.98 |
| Ahmedabad | AHMA | 82.92 ± 57.32 | 7.06 | -2.06 | no | 50.65 ± 50.21 | 2030 | 10.15 |
| Foshan | FOSH | 23.39 ± 20.47 | 6.88 | -1.169 | yes | 11.83 ± 11.73 | 2024 | 7.71 |
| Luanda | LUAN | 7.02 ± 7.70 | 6.73 | 0.572 | yes | 17.83 ± 7.66 | 2033 | 13.52 |
| Kuala Lumpur | KUAL | 8.29 ± 15.29 | 6.79 | -0.069 | no | 7.73 ± 7.77 | 2022 | 8.41 |
| Xian | XIAN | 104.40 ± 77.67 | 6.62 | -6.698 | yes | 41.60 ± 40.55 | 2024 | 9.01 |
| Hangzhou | HANG | 54.00 ± 43.20 | 6.57 | -2.85 | yes | 25.10 ± 24.95 | 2025 | 8.59 |
| Santiago | SANT | 10.43 ± 9.97 | 6.50 | -1.181 | yes | 0.25 ± 4.44 | 2023 | 6.90 |
| Shenyang | SHYA | 37.08 ± 20.37 | 6.46 | -1.928 | yes | 13.22 ± 13.17 | 2027 | 8.239 |
| Baghdad | BAGH | 41.22 ± 21.47 | 6.30 | -1.835 | yes | 15.75 ± 15.71 | 2028 | 8.85 |
| Madrid | MADR | 5.25 ± 5.37 | 6.17 | -0.306 | yes | 2.68 ± 2.70 | 2023 | 6.75 |
| Riyadh | RIYA | 47.53 ± 26.89 | 6.13 | -1.315 | no | 25.00 ± 25.33 | 2032 | 8.76 |
| Toronto | TORO | 3.18 ± 2.93 | 5.81 | -0.122 | no | 1.85 ± 1.85 | 2025 | 6.49 |
| Miami | MIAM | 8.00 ± 10.23 | 5.79 | -0.464 | no | 0.17 ± 9.54 | 2031 | 6.72 |
| Belo Horizonte | BELO | 2.37 ± 4.28 | 5.72 | -0.194 | no | 0.06 ± 2.90 | 2026 | 6.40 |
| Pune | PUNE | 70.48 ± 47.04 | 5.70 | -2.672 | yes | 35.24 ± 35.49 | 2028 | 8.07 |
| Haerbin | HAER | 20.44 ± 15.17 | 5.69 | -1.025 | yes | 9.29 ± 9.28 | 2025 | 7.07 |
| Dallas | DALL | 8.26 ± 5.51 | 5.66 | -0.159 | no | 5.46 ± 5.43 | 2032 | 7.19 |
| Surat | SURA | 71.46 ± 40.66 | 5.65 | 0.223 | no | 75.46 ± 40.93 | 2032 | 10.14 |
| Houston | HOUS | 12.45 ± 12.68 | 5.60 | 0.40 | no | 19.67 ± 12.62 | 2032 | 7.38 |
| Philadelphia | PHIL | 2.63 ± 2.09 | 5.57 | -0.038 | no | 1.98 ± 1.98 | 2031 | 6.16 |
| Fukuoka | FUKU | 18.20 ± 13.35 | 5.53 | -0.016 | no | 17.93 ± 13.27 | 2032 | 5.36 |
| Singapore | SING | 8.80 ± 13.76 | 5.46 | -0.215 | no | 7.05 ± 7.00 | 2023 | 6.08 |
| Suzhou | SUZH | 60.98 ± 44.28 | 5.34 | -3.427 | yes | 25.43 ± 25.58 | 2025 | 8.59 |
| Barcelona | BARC | 10.08 ± 7.65 | 5.23 | -0.479 | yes | 4.75 ± 4.81 | 2026 | 5.75 |
| Saint Petersburg | STPE | 4.55 ± 4.64 | 5.15 | -0.171 | no | 2.82 ± 2.77 | 2025 | 5.59 |
| Dar es Salaam | DARE | 8.98 ± 10.91 | 5.12 | 0.106 | no | 10.85 ± 10.83 | 2032 | 11.78 |
| Atlanta | ATLA | 3.81 ± 3.13 | 5.11 | -0.028 | no | 3.31 ± 3.15 | 2032 | 6.72 |
| Khartoum | KHAR | 26.97 ± 18.91 | 5.10 | -0.654 | no | 16.75 ± 16.59 | 2030 | 8.02 |
| Qingdao | QING | 56.30 ± 37.04 | 4.99 | -2.377 | yes | 26.29 ± 25.92 | 2027 | 6.42 |
| Johannesburg | JOHA | 10.48 ± 7.34 | 4.94 | -0.44 | yes | 5.02 ± 5.07 | 2027 | 6.66 |
| Washington, D.C. | WASH | 2.78 ± 2.54 | 4.93 | -0.036 | no | 2.22 ± 2.21 | 2030 | 8.98 |
| Dalian | DALI | 45.95 ± 27.24 | 4.82 | -1.749 | yes | 21.24 ± 21.07 | 2029 | 6.77 |
| Yangon | YANG | 18.53 ± 20.07 | 4.81 | -0.856 | no | 11.18 ± 10.97 | 2023 | 5.61 |
| Guadalajara | GUAD | 11.52 ± 14.67 | 4.77 | -0.038 | no | 11.03 ± 10.94 | 2027 | 5.73 |
| Alexandria | ALEX | 22.20 ± 11.62 | 4.76 | -0.24 | no | 17.85 ± 11.64 | 2032 | 6.69 |
| Jinan | JINA | 91.05 ± 47.74 | 4.58 | -5.44 | yes | 27.80 ± 27.61 | 2026 | 6.18 |

It is estimated that out of approximately 1030.2 million people residing in the world's 81 largest cities and megacities (average of the PBL mass concentration fine-mode dust projection year) population (UN, 2018a, 2019), the dust hazard consisted to a greater or lesser degree an environmental risk factor for the health of ~856.5 millions of people. More specifically, in these 81 large cities and megacities of the world:




- approximately 195.4 million will live in urban areas where both coarse-mode and fine-mode PM levels will fall below the WHO annual mean AQG for PM$_{2.5}$ and PM$_{10}$, respectively. This translates to ~113.1% increase in the population number of this category with respect to the reference observational 2007-2022 temporal period.

- approximately 346.8 million will live in cities where either the fine-mode or coarse-mode dust PM component will exceed the respective WHO annual mean AQG for PM$_{2.5}$ and PM$_{10}$, respectively. This translates to ~11.6% increase in the population number of this category with respect to the reference observational 2007-2022 temporal period.

- approximately 509.7 million will reside in areas where both coarse-mode and fine-mode PM concentrations will exceed the WHO annual mean AQG for PM$_{2.5}$ and PM$_{10}$, respectively. This translates to ~30.5% increase in the population number of this category with respect to the reference observational 2007-2022 temporal period.

It should be emphasized that the role of airborne fine-mode and coarse-mode dust aerosol components as environmental risk factors for human health disorders should be examined in the context of the absolute mass concentration levels and their changes with time, with reference WHO annual mean AQG safety thresholds for PM$_{2.5}$ (5 μg/m³) and PM$_{10}$, (15 μg/m³) in parallel with the large cities and megacities population change with time. More specifically, as shown in figure 16, at a continental level, the following atmospheric dust conditions are anticipated (Table 6):

Asia: The projected mass concentration of coarse-mode dust is anticipated to decline by ~25.5% (from 33.80 to 25.18 μg/m³), while fine-mode dust mass concentration is expected to decrease by ~40.3% (from 45.58 to 27.21 μg/m³). Despite these reductions, both components are projected to remain above the WHO annual mean AQG for PM$_{2.5}$ and PM$_{10}$, thereby continuing to pose a health risk, albeit reduced in severity. Notably, with an estimated ~33.3% increase in population (from 478.7 to 638.21 million) across 48 major cities and megacities in Asia (UN, 2018a, 2019), atmospheric dust is likely to remain a significant environmental health hazard, potentially affecting a larger number of individuals, however of lower severity since the per capita dust mass concentration exposure is projected to decrease.

Africa: Both coarse-mode and fine-mode mass concentrations of dust are projected to increase by ~13.5% and ~12.4%, respectively (from 30.78 to 34.94 μg/m³ and from 21.02 to 23.55 μg/m³). This increase coincides with a substantial projected population growth of ~70.8% (from 69.11 to 118.06 million) across the 8 major cities and megacities of the continent (UN, 2018a, 2019). In this context, atmospheric dust is expected to represent an escalating environmental health hazard, both in terms of severity and population affected, due to rising per capita exposure to airborne dust particulate matter.

South America: Both coarse-mode and fine-mode mass concentrations of dust are projected to decline significantly, by ~63.2% and ~47.3%, respectively (from 11.22 to 4.13 μg/m³ and from 7.33 to 3.86 μg/m³), while the population residing in the 7 major cities and megacities of the continent (UN, 2018a, 2019) is expected to grow by ~14.4% (from 79.76 to 91.22 million). In the case of large urban areas located in South America, atmospheric dust is projected to remain an environmental health hazard, potentially impacting a larger number of individuals; however, the associated risk may be of significant lower severity, since the per capita exposure to dust mass concentrations are projected to reside at climatological level, below the WHO annual mean AQG safety thresholds for PM$_{2.5}$ and PM$_{10}$.

North America: Coarse-mode dust mass concentration is projected to drop by ~34.8% (from 7.68 to 5.01 μg/m³), with a more modest ~7.9% decrease in fine-mode dust mass concentration (from 7.21 to 6.64 μg/m³), and an ~18.8% increase in population (from 103.88 to 123.40 million), residing in the 12 major cities and megacities of the continent (UN, 2018a, 2019). While the coarse-mode of dust in terms of mass concentration is projected to fall below WHO annual mean AQG safety threshold PM$_{10}$, the fine-mode of dust in terms of mass concentration is projected to remain close, though exceeding, to the WHO annual mean AQG safety threshold PM$_{2.5}$. Atmospheric dust will remain an environmental health hazard,



though of lower degree compared to the EO-based reference period, however potentially affecting a larger number of individuals.

Europe: Projections over the 6 large urban areas of Europe indicate an ~18.7% reduction in coarse-mode mass concentration of dust (from 4.77 to 3.88 μg/m³) and an ~38.1% decline in fine-mode mass concentration of dust (from 5.64 to 3.49 μg/m³),
with a modest ~10.4% population increase (from 47.81 to 52.77 million). Future projections indicate a high possibility that within the following years both the coarse-mode and fine-mode components of atmospheric dust in terms of mass concentration will fall below the WHO annual mean AQG for $PM_{10}$ and $PM_{2.5}$, respectively.

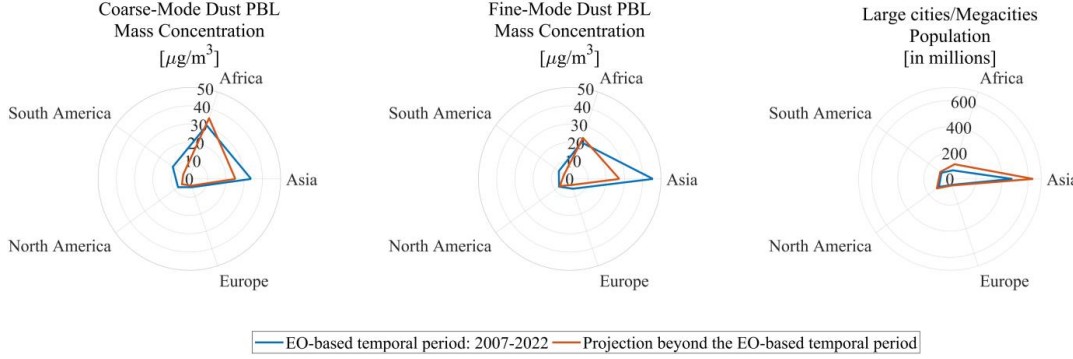


Figure 16: Average continental-scale coarse-mode dust PBL mass concentration (Fig.16-left), fine-mode dust PBL mass concentration (Fig.16-center), and population of the large cities and megacities (Fig.16-right), depicting the long-term EO-based mean conditions (blue line) and for the projection beyond the EO-based temporal period (red line).

Table 6: PBL coarse-mode and fine-mode dust mass concentrations and population of large cities and megacities of the world for the EO-based temporal period (2007-2022) and for the projection beyond the EO-based temporal period, for Asia, Africa, South America, North America, and Europe.

| | PBL coarse-mode dust [μg m⁻³] | | PBL fine-mode dust [μg m⁻³] | | Population (in millions) | |
|---|---|---|---|---|---|---|
| | EO-based temporal period (2007-2022) | Projection beyond the EO-based temporal period | EO-based temporal period (2007-2022) | Projection beyond the EO-based temporal period | EO-based temporal period (2007-2022) | Projection beyond the EO-based temporal period |
| Asia (48) | 33.80 | 25.18 (-25.5%) | 45.58 | 27.21 (-40.3%) | 478.7 | 638.21 (33.32%) |
| Africa (8) | 30.78 | 34.94 (13.52%) | 21.02 | 23.55 (12.4%) | 69.11 | 118.06 (70.83%) |
| S. America (7) | 11.22 | 4.13 (-63.19%) | 7.33 | 3.86 (-47.34%) | 79.76 | 91.22 (14.37%) |
| N. America (12) | 7.68 | 5.01 (-34.77%) | 7.21 | 6.64 (-7.91%) | 103.88 | 123.40 (18.79%) |
| Europe (6) | 4.77 | 3.88 (-18.66%) | 5.64 | 3.49 (-38.12%) | 47.81 | 52.77 (10.37%) |

## 6. Atmospheric dust as environmental risk factor for human health disorders


To date, numerous epidemiological studies have reported associations between elevated levels of airborne dust and adverse health effects. According to the revealed outcomes, coarse mineral particles are considered to pose a low health risk, primarily causing mild skin irritation or allergic reactions, even under conditions of prolonged exposure and high concentrations (Sandstrom, 2008; Pérez García-Pando et al., 2014). However, finer dust particles, particularly those in the $PM_{2.5}$ fraction,
present a greater concern due to their ability to penetrate deep into the respiratory system and reach the alveolar region (Martinelli et al., 2013; Lazaridis, 2023). More specifically, exposure to fine-mode dust has been linked to a range of health outcomes, including, among others, cardiovascular (Kwon et al., 2002; Meng and Lu, 2007; Middleton et al., 2008; Prospero et al., 2008; Sandstrom and Forsberg, 2008; Pérez et al., 2012; De Longueville et al., 2010; Martinelli et al., 2013; Goudie,



2014; Zhang et al., 2016; Achakulwisut et al., 2018; Querol et al., 2019) and respiratory diseases (Kwon et al., 2002; Wiggs et al., 2003; Chen et al., 2004; Veranth et al., 2004; Park et al., 2005; Derbyshire, 2007; Meng and Lu, 2007; Cheng et al., 2008; Yoo et al., 2008; De Longueville et al., 2010; 2013; Leski et al., 2011; Goudie, 2014; Katra et al., 2014; Mueller et al., 2017; Middleton, 2020), as well as an increased risk of lung cancer (Giannadaki et al., 2014; Steenland and Ward, 2014). Having quantified the PBL fine-mode and coarse-mode dust mass concentration over large cities and megacities of the world, the present section aims to translate these specific atmospheric conditions to a health risk associated with population exposure. The characterization is achieved through parametrizations established on the basis of epidemiological studies (Ostro, 2004; Soares et al., 2022) allowing to quantify the dust-exposure (i) relative risk (RR) and (ii) attributable fraction (AF).

For short-term exposure to the coarse-mode of dust, the RR for 41 cities with a 16-year mean mass concentration up to 15 µg m$^{-3}$ was lower than 1.004 (Figure 17a). The latter is linked with an AF lower than 0.4%. These cities exhibit the lower dust mass concentrations than WHO 24-hour mean AQG safety thresholds, thus are characterized by the lower risk expressed by a very small fraction of incidences for all-cause mortality for the exposed population. Higher coarse-mode dust mass concentrations and between 15 and 45 µg m$^{-3}$ were obtained for 27 cities with an RR ranging between 1.004 and 1.030 and an AF ranging between 0.4 - 2.8%. The higher mortality risks for these cities lead to an increased AF for the exposed population which indicates that 2.8% of incidences can be avoided if concentration is reduced to the target value (10 µg m$^{-3}$). Similarly, 13 cities with coarse-mode dust mass concentration higher than 45 µg m$^{-3}$ were associated with an RR higher than 1.03 and up to to 1.118 for the city of Riyadh (Saudi Arabia). These cities correspond to the most air quality burdened urban environments as mortality risks become considerable with 2.8 - 10.6 % of incidences could be avoided with a reduction of dust concentrations to 10 µg m$^{-3}$. Table 7 lists the 10 cities and megacities with the higher coarse-mode dust mass concentrations and the respective RRs and AFs.

For long-term exposure to PM$_{2.5}$, the RR for 11 cities with a 16-year mean fine-mode dust mass concentration lower than 5 µg m$^{-3}$ was below 1.06 and 1.10 for cardiopulmonary mortality and lung cancer respectively (Figure 17b). AF for these cities were the lowest, and less than 6.1% of cardiopulmonary mortality and 8.9% of lung cancer incidences could be avoided with a reduction of fine-mode dust mass concentration to 3 µg m$^{-3}$. In contrast to health risk results for coarse-mode dust, a higher number of cities was obtained with a notable risk for fine dust-mode. This observation is a direct outcome of the stricter WHO AQG for PM$^{2.5}$ compared to PM$^{10}$ (WHO, 2021). In particular, 70 cities (15 < 25 cities < 45 µg m$^{-3}$; 45 cities > 45 µg m$^{-3}$) exhibited concentrations higher than 15 µg m$^{-3}$ which is translated to a cardiopulmonary risk mortality of 1.06 - 1.683 and a lung cancer risk of 1.10 - 2.179. Translating these risks to AFs for the exposed population gives fractions that range up to 40.6% (Karachi) for cardiopulmonary mortality and up to 54.1% (Karachi) for lung cancer. These numbers suggest that a significant number of incidents could be avoided with reduction of fine-mode dust concentrations to the target value, and that the environment in these cities poses a considerable threat for the exposed population. Detailed RRs and AFs values for the cities and megacities with higher fine-mode dust mass concentrations are presented in Tables 8 and 9 for cardiopulmonary and lung cancer respectively.



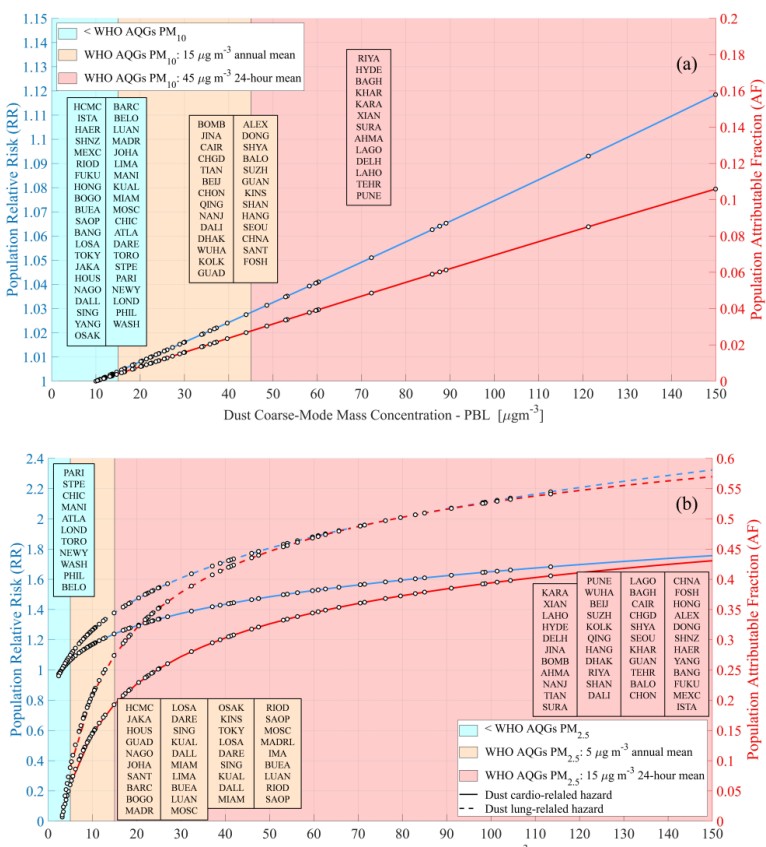

Figure 17: Relative risk and attributable fraction for (a) dust coarse-mode and (b) dust fine-mode. Open circles in each curve represent the estimated index for each city. Color classification was based on WHO AQG using both annual and 24-h mean concentrations.

Table 7: PBL coarse-mode dust mass concentration (mean ± std) for the EO-based temporal period (12/2006-11/2022), population relative risk (RR) mean and rage, and population attribution fraction (AF) mean and range, for the 10 large cities and megacities of the world with higher mass concentrations.

| City | Country | PM (µg/m³) | RR | RR - range | AF | AF - range |
|---|---|---|---|---|---|---|
| Riyadh | Saudi Arabia | 149.94 ± 86.68 | 1.118 | (1.063,1.199) | 0.106 | (0.060,0.166) |
| Hyderabad | India | 121.22 ± 78.35 | 1.093 | (1.056,1.164) | 0.085 | (0.053,0.141) |
| Baghdad | Iraq | 89.08 ± 57.3 | 1.065 | (1.039,1.115) | 0.061 | (0.037,0.103) |
| Khartoum | Sudan | 87.63 ± 42.68 | 1.064 | (1.026,1.101) | 0.060 | (0.026,0.092) |
| Karachi | Pakistan | 85.95 ± 55.08 | 1.063 | (1.037,1.111) | 0.059 | (0.035,0.100) |
| Xian | China | 72.22 ± 57.15 | 1.051 | (1.038,1.100) | 0.049 | (0.037,0.091) |
| Surat | India | 60.25 ± 64.2 | 1.041 | (1.044,1.096) | 0.039 | (0.042,0.087) |
| Ahmedabad | India | 59.79 ± 58.6 | 1.041 | (1.040,1.091) | 0.039 | (0.038,0.083) |
| Lagos | Nigeria | 59.64 ± 84.33 | 1.041 | (1.061,1.113) | 0.039 | (0.058,0.102) |
| Delhi | India | 58.2 ± 55.09 | 1.039 | (1.037,1.086) | 0.038 | (0.035,0.079) |

Table 8: PBL fine-mode dust mass concentration (mean ± std) for the EO-based temporal period (12/2006-11/2022), population relative risk (RR) mean and rage, and population attribution fraction (AF) mean and range, for the 10 large cities and megacities of the world with higher mass concentrations, as cardiopulmonary related dust hazard.



| City | Country | PM (μg/m³) | RR | RR - range | AF | AF - range |
|------|---------|------------|-----|------------|-----|------------|
| Karachi | Pakistan | 113.49 ± 43.77 | 1.683 | (1.562,1.769) | 0.406 | (0.360,0.435) |
| Xian | China | 104.40 ± 77.67 | 1.661 | (1.350,1.810) | 0.398 | (0.259,0.447) |
| Lahore | Pakistan | 101.39 ± 43.39 | 1.654 | (1.518,1.747) | 0.395 | (0.341,0.428) |
| Hyderabad | India | 98.68 ± 36.45 | 1.647 | (1.535,1.729) | 0.393 | (0.348,0.421) |
| Delhi | India | 98.14 ± 42.44 | 1.646 | (1.509,1.739) | 0.392 | (0.337,0.425) |
| Jinan | China | 91.05 ± 47.74 | 1.627 | (1.452,1.736) | 0.385 | (0.311,0.424) |
| Bombay | India | 85.01 ± 50.26 | 1.610 | (1.405,1.729) | 0.379 | (0.288,0.422) |
| Ahmedabad | India | 82.92 ± 57.32 | 1.604 | (1.342,1.738) | 0.376 | (0.255,0.425) |
| Nanjing | China | 79.58 ± 49.10 | 1.593 | (1.377,1.716) | 0.372 | (0.274,0.417) |
| Tianjin | China | 76.17 ± 43.92 | 1.583 | (1.389,1.697) | 0.368 | (0.280,0.411) |

Table 9: PBL fine-mode dust mass concentration (mean ± std) for the EO-based temporal period (12/2006-11/2022), population relative risk (RR) mean and rage, and population attribution fraction (AF) mean and range, for the 10 large cities and megacities of the world with higher mass concentrations, as lung cancer related dust hazard.

| City | Country | PM (μg/m³) | RR | RR - range | AF | AF - range |
|------|---------|------------|-----|------------|-----|------------|
| Karachi | Pakistan | 113.49 ± 43.77 | 2.179 | (1.948,2.349) | 0.541 | (0.487,0.574) |
| Xian | China | 104.40 ± 77.67 | 2.137 | (1.568,2.430) | 0.532 | (0.362,0.588) |
| Lahore | Pakistan | 101.39 ± 43.39 | 2.123 | (1.868,2.305) | 0.529 | (0.465,0.566) |
| Hyderabad | India | 98.68 ± 36.45 | 2.110 | (1.898,2.268) | 0.526 | (0.473,0.559) |
| Delhi | India | 98.14 ± 42.44 | 2.107 | (1.851,2.289) | 0.525 | (0.460,0.563) |
| Jinan | China | 91.05 ± 47.74 | 2.071 | (1.748,2.282) | 0.517 | (0.428,0.562) |
| Bombay | India | 85.01 ± 50.26 | 2.039 | (1.663,2.269) | 0.510 | (0.399,0.559) |
| Ahmedabad | India | 82.92 ± 57.32 | 2.027 | (1.553,2.288) | 0.507 | (0.356,0.563) |
| Nanjing | China | 79.58 ± 49.10 | 2.008 | (1.614,2.243) | 0.502 | (0.381,0.554) |
| Tianjin | China | 76.17 ± 43.92 | 1.988 | (1.635,2.207) | 0.497 | (0.388,0.547) |

## 7. Summary and conclusions

To what extent have the submicrometer (fine-mode) and supermicrometer (coarse-mode) fractions of mineral dust entrained into the atmosphere and confined within the PBL have changed over highly industrialized and densely populated urban areas during the past two decades? Is it feasible to detect statistically significant temporal trends in these changes? Which major urban centers currently experience fine-mode and coarse-mode dust mass concentrations within the PBL that exceed the WHO AQGs (WHO, 2021), and over which large cities and megacities it is foreseen the dust modes to exceed WHO AQGs in the near-future?

The present work provided EO-based quantification of the mass concentration levels of the two modes of dust and insight on the risk the hazard implies on human health. Cornerstone of the analysis is the ESA-LIVAS CDR (Amiridis et al., 2013; 2015; Marinou et al., 2017; Proestakis et al., 2018) that provides satellite-based, four-dimensional, near-global, and multiyear information on the two modes of dust, including mass concentration profiles for the period 12/2006-11/2022 (Proestakis et al., 2024). In a nutshell, the ESA-LIVAS fine-mode and coarse-mode dust CDR is established through (i) a combination of long terms CALIPSO-CALIOP observations (Winker et al., 2010), (ii) laboratory experiments providing parametrizations on the dependence of depolarization ratio on dust PSD (Sakai et al., 2010; Järvinen et al., 2016), and (iii) an EARLINET-established sophisticated approach allowing to decouple the submicrometer and supermicrometer in terms of diameter components of dust, namely the two-step POLIPHON technique (Mamouri and Ansmann, 2014, 2017). The present study focused on the levels of dust fine-mode and coarse-mode mass concentration within the lowest part of the troposphere (PBL), where low air quality strongly influences human health, over the larger cities and megacities of the world, as defined by UN (UN, 2018a, 2019).

First central concluding remark is that atmospheric dust unequivocally poses a significant environmental hazard to public health in a substantial number of global large cities and megacities; conclusion reached on the basis of comparison between the EO-based derived near-surface dust mass concentrations and the WHO AQG safety thresholds. The comparison outcomes





reveal that a significant proportion of major cities and megacities, approximately 49.4% for $PM_{10}$ and 87.7% for $PM_{2.5}$, exceed WHO annual mean AQG safety thresholds for coarse-mode and fine-mode dust aerosol mass concentrations within the PBL, respectively. In addition, with respect to the geographical distribution, cities in the Middle East, Indian subcontinent, East Asia, North Africa, and the Sahel-Central Africa region are particularly affected by both dust aerosol components. Based on population data from the UN (UN, 2018a, 2019), it is estimated that among approximately 807.9 million individuals residing in the world's 81 largest cities and megacities (average population, 2007–2022), ~701.4 million are exposed to airborne dust levels beyond WHO AQGs. More specifically, approximately 91.7 million people live in urban areas where both fine-mode and coarse-mode dust PM concentrations remain below the WHO AQGs for $PM_{2.5}$ and $PM_{10}$, respectively. In contrast, around 310.8 million reside in cities where either the fine-mode or coarse-mode dust PM component exceeds the corresponding WHO AQG thresholds, while approximately 390.6 million live in urban regions where both $PM_{2.5}$ and $PM_{10}$ concentrations surpass the WHO air quality safety recommended limits.

Furthermore, it is of interest that the performed trend analysis carried out over the EO-based temporal period revealed in general declining tendencies in both the PBL coarse-mode and fine-mode components of dust aerosol, observed in ~74.1% and ~86.4% of the urban cities, respectively. However, in most of the cases the trends were characterized by lack of statistical significance. It is discussed and emphasized though that absence of statistically significant trends in urban centers characterized by coarse-mode or/and fine-mode dust mass concentrations within the PBL exceeding WHO AQG safety thresholds may reflect persistent exposure levels that pose and will continue to pose health risks in the coming years, potentially implying lack of effective, or adequately implemented, or even complete absence of air quality management policies, highlighting the necessity for region-specific air quality monitoring and public awareness raising, and intervention of mitigation and adaptation strategies.

Second central concluding remark is that in the years to come atmospheric dust is foreseen to remain an environmental hazard to public health in a substantial number of global large cities and megacities, in general potentially reduced in severity, however affecting a significantly larger number of individuals. It is estimated that approximately 1030.2 million individuals will inhabit the world's 81 largest cities and megacities during the first half of the third decade of the 21$^{st}$ century (UN, 2018a, 2019), population growth of approximately 27.8% compared to the EO-based refernce temporal period. Projections of the PBL fine-mode and coarse-mode mass concentration trends into the temporal period extending beyond the EO-based temporal period indicate that of the estimated future urban population, approximately 856.5 million will be exposed to airborne dust levels potentially posing an environmental-induced human health risk. Specifically, around 195.4 million people (113.1% increase) are expected to live in large urban areas where both $PM_{2.5}$ and $PM_{10}$ levels remain below WHO annual mean AQGs. In contrast, around 346.8 million individuals (11.6% increase) will reside in large cities and megacities where either $PM_{2.5}$ or $PM_{10}$ concentrations exceed their respective WHO annual mean AQGs, while approximately 509.7 million of people (an 30.5% increase) will be exposed to simultaneous exceedances of both dust components.

At a continental scale and large cities and megacities average, future projections reveal distinct trends in atmospheric dust exposure and associated health risks. In Asia, PBL coarse-mode and fine-mode dust mass concentrations are expected to decrease by approximately 25.5% (from 33.80 to 25.18 µg/m³) and 40.3% (from 45.58 to 27.21 µg/m³), respectively. Despite these declines, both fractions are projected to remain above WHO annual mean AQGs for $PM_{10}$ and $PM_{2.5}$, continuing to pose a health risk to an expanding urban population, which is estimated to grow by around 33.3% (from 478.7 to 638.21 million). In Africa, PBL coarse-mode and fine-mode dust levels are anticipated to rise by approximately 13.5% (from 30.78 to 34.94 µg/m³) and 12.4% (from 21.02 to 23.55 µg/m³), respectively, alongside a significant population increase of around 70.8% (from 69.11 to 118.06 million), indicating a likely intensification of the dust hazard. South America is projected to experience substantial reductions in PBL dust levels, with coarse-mode decreasing by approximately 63.2% (from 11.22 to 4.13 µg/m³) and fine-mode by around 47.3% (from 7.33 to 3.86 µg/m³), while the urban population is projected to grow moderately by around 14.4% (from 79.76 to 91.22 million), suggesting lower severity of exposure despite more individuals being potentially



affected. In North America, a decline of approximately 34.8% in coarse-mode (from 7.68 to 5.01 µg/m³) and approximately 7.9% in fine-mode (from 7.21 to 6.64 µg/m³) PBL dust is anticipated, accompanied by an around 18.8% population increase (from 103.88 to 123.40 million); while coarse-mode levels are projected to fall below the WHO AQG for $PM_{10}$, fine-mode is expected to remain slightly above the $PM_{2.5}$ air quality recommendation threshold. Europe presents the most favorable scenario, with projected declines of approximately 18.7% in coarse-mode (from 4.77 to 3.88 µg/m³) and approximately 38.1% in fine-mode (from 5.64 to 3.49 µg/m³) PBL dust, and only a modest population growth of around 10.4% (from 47.81 to 52.77 million), indicating a high probability that both PM fractions will fall below WHO annual mean AQGs, thereby reducing dust-related health risks.

As a next a final step, based on the quantified fine-mode and coarse-mode dust mass concentrations within the PBL over the major cities and megacities worldwide, the study translated the specific atmospheric dust conditions during the EO-based temporal period into associated health risks for the exposed urban populations (e.g. skin irritation, allergic reactions, cardiovascular and respiratory diseases, lung cancer; Martinelli et al., 2013; Giannadaki et al., 2014; Goudie, 2014; Querol et al., 2019; Middleton, 2020; Lazaridis, 2023). However, dust aerosol as cause and health impacts as effects, are not related linearly, but in a more complex way. To capture these effects, the study employed concentration-response functions derived from epidemiological studies (Ostro 2004; Soares et al. 2022). Using the estimated PBL dust mass concentrations for each urban area, two key health risk indicators were calculated: (i) the Relative Risk (RR), which quantifies the increased likelihood of specific health outcomes (e.g., all-cause mortality, lung cancer) due to air pollution exposure, and (ii) the Attributable Fraction (AF), which estimates the proportion of disease incidence in the population that could be prevented if ambient pollutant concentrations were reduced to reference levels. Overall, for short-term exposure to $PM_{10}$, cities with coarse-mode dust levels below 15 µg/m³ showed RR < 1.004 and AF < 0.4%, indicating minimal health impact. Cities with concentrations between 15–45 µg/m³ had RRs up to 1.030 and AFs reaching 2.8%. In the most polluted urban environments (>45 µg/m³) RR exceeded 1.118 with AFs reaching as high as 10.6% (Riyadh-Saudi Arabia). For long-term exposure to $PM_{2.5}$, 11 cities with fine-mode dust concentrations <5 µg/m³ exhibited RR < 1.06 for cardiopulmonary mortality and <1.10 for lung cancer, corresponding to AFs below 6.1% and 8.9%, respectively. However, 70 cities with $PM_{2.5}$ concentrations >15 µg/m³ showed elevated health risks, with RR values reaching as high as 1.683 for cardiopulmonary and as high as 2.179 for lung cancer, translating to AF as high as approximately 40.6% for cardiopulmonary and as high as 54.1 for lung cancer (Karachi-Pakistan).

Closing, among the aerosol species, mineral dust plays a dominant role in the Earth's climate system, affecting significantly anthropogenic activities, as well as humans' health. In this context, the findings of the present study are particularly valuable for data-scarce large cities and megacities, offering Earth Observation based insight into the health hazard associated with airborne dust aerosols. These insights can support evidence-based policymaking, urban planning, and public health interventions. This research has broad implications for global air quality management, especially in rapidly urbanizing regions. The demonstrated effectiveness of targeted regulatory measures, as reflected in the observed reduction of dust concentrations in certain cities, offers a promising path forward. At the same time, the identification of persistent hotspots where dust levels exceed safe thresholds underscores the urgent need for further action. By advancing observational evidence, this study contributes to a deeper scientific understanding of the health risks associated with dust aerosols. The findings highlight the necessity of integrating EO-based dust monitoring into urban health and environmental policy frameworks, facilitating the development of more effective adaptation and mitigation strategies. Ultimately, this work underscores the role of EO science in translating environmental monitoring into tangible public health benefits. It provides a strategic tool for stakeholders and decision-makers, enabling the design of robust interventions to protect human health in the face of accelerating urbanization and climate change.



**Data availability**

The EO-based level 2 pure-dust fine-mode and coarse-mode CDR is available at https://doi.org/10.5281/zenodo.10389741 (Proestakis, 2024). The LIVAS level 2 and level 3 pure-dust CDR is available upon personal communication with Emmanouil Proestakis (proestakis@noa.gr), Eleni Marinou (elmarinou@noa.gr), and/or Vassilis Amiridis (vamoir@noa.gr). The CALIPSO lidar level 1B and level 2 data products are publicly available from the Atmospheric Science Data Center at NASA Langley Research Center (https://earthdata.nasa.gov/eosdis/daacs/asdc, Earthdata, 2023). Population estimates of large cities and megacities over a long time period (1950-2035) is provided by the UN's collection of datasets (https://population.un.org/wup/).

**Author contributions**

EP: conceptualization, methodology, software, data curation, formal analysis, and project administration. KP: conceptualization, methodology, data curation, formal analysis. Thanasis Georgiou: software, data curation, and formal analysis. SEC: conceptualization, methodology, software, data curation, formal analysis, investigation, and visualization. ML: conceptualization and methodology. AG: conceptualization and investigation. IF: conceptualization and validation. IT: conceptualizations. MPP: methodology. VA: conceptualization.

**Competing interests.**

None of the authors has any competing interests.

**Acknowledgements**

Emmanouil Proestakis acknowledges support by the AXA Research Fund for postdoctoral researchers under the project entitled "Earth Observation for Air-Quality – Dust Fine-Mode (EO4AQ-DustFM)". Emmanouil Proestakis and Kyriakoula Papachristopoulou would like to acknowledge the COST Action HARMONIA, CA21119, supported by COST (European Cooperation in Science and Technology).

**Financial support**

This research has been supported by the AXA Research Fund by the AXA Research Fund for postdoctoral researchers under the project entitled "Earth Observation for Air-Quality – Dust Fine-Mode (EO4AQ-DustFM)".

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
