# Peer review of "Atmospheric Dust and Air Quality over large-cities and megacities of the World"

_EGUsphere, 2025_

## Author Comment (AC1)

**Atmospheric Dust and Air Quality over large-cities and megacities of the World**

This is a generally well-written paper on an important topic. I have only a few relatively minor suggested revisions.

The authors would like to thank the reviewer for his time, comments and suggestions. We did our best to incorporate the proposed changes and corrections in the revised manuscript, aiming at improving the presented paper. Following, you will find our responses, one by one to the comments addressed.

Kind regards,
Emmanouil Proestakis et al.

**Reviewer's Comments**

The text is somewhat wordy. It would benefit from a read-through to streamline and eliminate redundancies. For example, the beginning of Section 6 is redundant with the portion of Section 1 that describes the health impacts of coarse and fine particles.

The authors agree with the reviewer.
Following the reviewer's suggestion the beginning text of Section 6 has been removed and replaced as follows:

*from:* "To date, numerous epidemiological studies have reported associations between elevated levels of airborne dust and adverse health effects. According to the revealed outcomes, coarse mineral particles are considered to pose a low health risk, primarily causing mild skin irritation or allergic reactions, even under conditions of prolonged exposure and high concentrations (Sandstrom, 2008; Pérez García-Pando et al., 2014). However, finer dust particles, particularly those in the PM2.5 fraction, present a greater concern due to their ability to penetrate deep into the respiratory system and reach the alveolar region (Martinelli et al., 2013; Lazaridis, 2023). More specifically, exposure to fine-mode dust has been linked to a range of health outcomes, including, among others, cardiovascular (Kwon et al., 2002; Meng and Lu, 2007; Middleton et al., 2008; Prospero et al., 2008; Sandstrom and Forsberg, 2008; Pérez et al., 2012; De Longueville et al., 2010; Martinelli et al., 2013; Goudie, 2014; Zhang et al., 2016; Achakulwisut et al., 2018; Querol et al., 2019) and respiratory diseases (Kwon et al., 2002; Wiggs et al., 2003; Chen et al., 2004; Veranth et al., 2004; Park et al., 2005; Derbyshire, 2007; Meng and Lu, 2007; Cheng et al., 2008; Yoo et al., 2008; De Longueville et al., 2010; 2013; Leski et al., 2011; Goudie, 2014; Katra et al., 2014; Mueller et al., 2017; Middleton, 2020), as well as an increased risk of lung cancer (Giannadaki et al., 2014; Steenland and Ward, 2014).".

*to:* "To date, numerous epidemiological studies report on the adverse effects of airborne dust on human health, with more pronounced the impact of the fine-mode ($PM_{2.5}$) due to the deeper penetration into the respiratory system and the alveolar region (Martinelli et al., 2013; Lazaridis, 2023).".

Along the same lines, the portion of Section 1 that describes the importance of atmospheric dust in terms of its "effects on biogeochemistry, the radiation budget, weather, and climate" (lines 47-66) is too detailed. The study focuses on the health impacts of dust, so the importance of atmospheric dust's impacts on human health should be the focus. Briefly mentioning the important of dust in other fields would be sufficient.

The authors agree with the reviewer.
Following the reviewer's suggestion the beginning text of Section 6 has been removed and replaced as follows:

*from:* "Among the aerosol species resulting in degradation of air quality are mineral dust particles, especially over densely populated and heavily industrialized areas (Papachristopoulou et al., 2022; Proestakis et al., 2024). More specifically, atmospheric dust is recognized as one of the most important aerosol types, both in terms of mass and optical depth, and the dominant component of atmospheric aerosol over large areas of the Earth (Gliß et al., 2021; Kok et al., 2017; 2021; 2023). Once suspended in the atmosphere, dust exerts a multifaceted and complex role in the Earth's climate system, while simultaneously posing considerable challenges to anthropogenic activities. More specifically, upon entering the atmosphere, dust particles are subject to aeolian transport, in many cases over distances of thousands of kilometres downwind (e.g. Prospero, 1999a, 1999b;

Dey et al., 2004; Schepanski et al., 2009; Kanitz et al., 2014; Weinzierl et al., 2016; Marinou et al., 2017; Proestakis et al., 2018; 2024; Ramaswamy et al., 2018; Adebiyi and Kok, 2020; Aslanoğlu et al., 2022; Drakaki et al., 2022; Gkikas et al., 2022). While airborne, dust particles affect several atmospheric processes, spanning from short- (weather) to long- (climate) term temporal scales, via their interactions with the shortwave (SW) and longwave (LW) radiation. Dust aerosols serve as effective cloud condensation nuclei (CCN; Hatch et al., 2008) and/or ice-nucleating particles (INPs; DeMott et al., 2009). Atmospheric dust layers modify clouds' microphysical, macrophysical and optical properties, precipitation patterns, atmospheric stability, cloud formation, lifetime, and coverage (Twomey, 1977; Albrecht, 1989; Rosenfeld et al., 2008). Dust is considered a significant parameter related to aviation safety (Papagiannopoulos et al., 2020; Nickovic et al., 2021; Ryder et al., 2024) while, by reducing the amount of SW radiation reaching the Earth's surface, dust layers affect solar energy production (Kosmopoulos et al., 2018; Masoom et al., 2021; Fountoulakis et al., 2021). Eventually, upon their removal from the atmosphere, through wet or dry deposition (Gao et al., 2003; Hand et al., 2004; Prospero et al., 2010; Mahowald et al., 2011; Van der Does et al., 2018; 2021; Proestakis et al., 2025), dust particles enrich with micro nutrients the marine and terrestrial ecosystems (Okin et al., 2004; Jickells et al., 2005; Li et al., 2018).".

*to:* "Among the aerosol species resulting in degradation of air quality are mineral dust particles, especially over densely populated and heavily industrialized areas (Papachristopoulou et al., 2022; Proestakis et al., 2024). Atmospheric dust is the dominant component of atmospheric aerosol over large areas of the Earth (Gliß et al., 2021; Kok et al., 2017; 2021; 2023). Transported over thousands of kilometres (e.g. Prospero, 1999a, 1999b; Dey et al., 2004; Schepanski et al., 2009; Kanitz et al., 2014; Weinzierl et al., 2016; Marinou et al., 2017; Proestakis et al., 2018; 2024; Ramaswamy et al., 2018; Adebiyi and Kok, 2020; Aslanoğlu et al., 2022; Drakaki et al., 2022; Gkikas et al., 2022), dust interacts with radiation, clouds, and precipitation (Twomey, 1977; Albrecht, 1989; Hatch et al., 2008; Rosenfeld et al., 2008; DeMott et al., 2009), affecting weather, climate, aviation safety, and solar energy production (Kosmopoulos et al., 2018; Papagiannopoulos et al., 2020; Fountoulakis et al., 2021; Masoom et al., 2021; Nickovic et al., 2021; Ryder et al., 2024). Ultimately, upon deposition (Gao et al., 2003; Hand et al., 2004; Prospero et al., 2010; Mahowald et al., 2011; Van der Does et al., 2018; 2021; Proestakis et al., 2025) dust particles enrich with nutrients marine and terrestrial ecosystems (Okin et al., 2004; Jickells et al., 2005; Li et al., 2018).".

The authors never address the fundamental question of why they used remote sensing data, i.e. CALIPSO and AERONET, to analyze the impacts of dust on surface air quality, instead of surface $PM_{10}$ and $PM_{2.5}$ monitors. Do the 81 cities analyzed have PM monitor networks? Were PM monitor data used to, for example, validate the ESA-LIVAS atmospheric dust products? Using surface monitor data would counteract the shortcomings of the remote sensing dataset, such as the impacts of clouds and the coarse 1°x1° spatial resolution. I suspect I know the reasoning for the authors' focus on remote sensing data, but they need to clearly justify their choice in Section 1. This is a glaring omission that will puzzle any air quality experts reading the paper.

The authors appreciate the reviewer's observation and the opportunity given to clarify even more the rationale for using the CALIPSO-based LIVAS climate data record and remote sensing data, instead of surface PM10 and PM2.5 monitors. As such, towards clarifying the above conceptual approach of the authors and the motivation that lies behind the study, and following the reviewer's recommendation and valid comment the following text is included in the manuscript in the Section "Introduction":

"…

To date, in-situ measurements of particulate matter represent the most direct and reliable source of information on ambient air quality (including of the dust aerosol component), allowing for high temporal resolution measurement, low detection thresholds, and the capacity to distinguish between PM10, PM2.5 and even finer size fractions with high precision. Established under networks, such as OpenAQ (https://openaq.org/; last access: 16/09/2025), IQAir (https://www.iqair.com/air-quality-map; last access: 16/09/2025), and SPARTAN (https://www.spartan-network.org/; last access: 16/09/2025), in situ PM measurements are widely used for regulatory purposes, and are considered indispensable, among others, for health impact assessments, epidemiological studies, local air quality management and empowering evidence-based decision-making. More specifically, ground-based in situ monitoring stations provide unparalleled air quality measurements, enabling researchers, policymakers, and the public to track the aerosol load over time, to identify pollution hotspots, evaluate the effectiveness of environmental regulations, and allowing public health studies and air quality forecasting. In the case of the dust aerosol component in situ measurement have significantly contributed through shedding light on dust outbreaks over specific regions in terms of concentrations, phenomenology, and trends, and on dust relation with synoptic and mesoscale meteorology

(Querol et al., 2009; 2013), the contribution to daily PM10 concentrations (Stafoggia et al., 2016), and the broader impact on air quality (Querol et al., 2019).

Nevertheless, despite these significant advantages numerous challenges inherent to the complex nature of in situ measurements of ambient air quality hamper the feasibility of establishing and providing long-term and continuous measurements of high spatial and temporal coverage. More specifically, surface monitoring stations and networks of monitoring stations are not uniformly operational and available across the globe. Even in the case of large cities and megacities, particularly in the case of cities of Africa, parts of Asia, Middle East, and South America, monitoring stations are sparse or even completely absent, while even where networks exist, the provided aerosol load measurements are frequently characterized by non-continuity in terms of temporal coverage due to instrument operation, maintenance, malefactions, or resource limitations. Moreover, different types of instruments and measurement protocols introduce inconsistencies across regions, while spatial representativeness remains limited, as most stations are confined to specific urban environments and may not adequately capture variability within a metropolitan or larger city area.

Towards addressing these formidable challenges air quality monitoring frequently relies on satellite-based earth observations of the aerosol load, offering unique advantages in terms of spatial consistency, global coverage, and the ability to provide long-term, homogeneous datasets across large regions where monitoring networks are incomplete or even completely absent, though with lower accuracy than the accuracy offered by in-situ measurements. However, and despite the increasing number of scientific studies indicating that airborne mineral dust constitutes a significant environmental hazard and risk factor for human health, current knowledge on the dust health impacts, when it comes to incorporating EOs is still characterized by large uncertainties, primarily attributed to three key challenges.

…"

---

## Author Comment (AC2)

**Atmospheric Dust and Air Quality over large-cities and megacities of the World**

The study assesses global urban exposure to fine- and coarse-mode mineral dust within the planetary boundary layer (PBL) using satellite-derived data from the ESA-LIVAS climatology (2006–2022). It quantifies dust mass concentrations, evaluates compliance with WHO air quality guidelines, and estimates associated health risks for populations in 81 major cities and megacities. Future projections suggest continued health threats from dust exposure, especially in Africa and Asia, despite some decreasing trends. In general, the overall work is well supported and professionally prepared.

The authors would like to thank the reviewer for his time, comments and suggestions. We did our best to incorporate the proposed changes and corrections in the revised manuscript, aiming at improving the presented paper. Following, you will find our responses, one by one to the comments addressed.

Kind regards,
Emmanouil Proestakis et al.

**Reviewer's Comments**

1) Attribution of dust concentration trends to regulatory measures lacks empirical support. While the study highlights observed decreasing trends in both fine- and coarse-mode dust mass concentrations over many urban areas, it simultaneously acknowledges that these trends are mostly not statistically significant. Nevertheless, the paper suggests that such reductions reflect the effectiveness of regulatory measures. This connection appears speculative in the absence of a causal analysis. To strengthen this point, it would be necessary to incorporate a policy impact assessment or time-series intervention analysis demonstrating how specific air quality policies temporally align with reductions in dust levels.

To begin with, we would like to thank the reviewer for this point, resulting in realizing that this part was not clear possible leading to misinterpretations and in need of clarifications and further polishing. More specifically, with respect to the comment of the reviewer that "the paper suggests that such reductions (in the atmospheric dust load over large cities and megacities) reflect the effectiveness of regulatory measures is speculative" it should be emphasized that the authors completely agree with the reviewer. In addition, it would be to the right direction to elaborate a bit more on the rationale behind the aforementioned reflections.

In general, to date, several organization reports and scientific studies document a causal relationship between air quality regulatory policies/measures and air quality improvements. Regulatory frameworks, such as technological transitions, imposition of emission-related fees, strict regulations, and establishments of air quality management, monitoring and regulating schemes implemented in numerous countries across the globe have resulted in reduction of aerosol emissions to mitigate the deterioration of air quality and to address associated adverse health impacts. For instance, the World Health Assembly (WHA69) outlined in 2016 a global strategy towards decreasing mortality linked to air pollution, to improve urban air quality, and promote clean energy adoption (WHO, 2016). In the United States, initial legislations addressing air pollution were enacted in 1970, and subsequent reports by the Environmental Protection Agency (EPA) indicate an approximate 40% decline in aerosol-related emissions and pollution across major urban centers (DeMocker, 2003). European Union (EU) member states have a documented reduction of an approximately 29% in aerosol concentrations since 1970, mainly attributed to regulatory measures, interventions, and technological advancements (Turnock et al., 2016), while at present through the ambient air quality directive (AAQD - 2024/2881) EU aims towards even more ambitious air quality achievements. In China policy developments over the last three decades have significantly reduced anthropogenic emissions, particularly during the past ten years (Jin et al., 2016; Zheng et al., 2018) while India introduced the National Clean Air Programme in 2019 with the overarching objective to reduce particulate pollution levels by 20–30% relative to 2017 baselines by 2024, across 122 cities (Ganguly et al., 2020). On the contrary, air pollution regulations across many African cities still remains underdeveloped or even completely absent (Abera et al., 2020), posing substantial challenges to future air quality management given the rapid urban population growth of the continent. In general, a causal relationship between air quality regulatory policies/measures and air quality improvements is apparent.

[Figure]

Figure 01: EO-based (MODIS) AOD (not DOD) trends (per decade) over large cities and megacities of the world, including information on the statistical significance of the trends denoted by the circle radius. (Source: Papachristopoulou et al., 2022).

It has to be emphasized that, even though the regulatory policies/measures regulate anthropogenic activities and as such anthropogenic aerosol emissions with frequently (statistical) significant impact on the total aerosol load (Fig. 01), the authors agree with the argument of the reviewer that in the case of atmospheric dust, such modulation of the total dust load with time is more complicated and that a connection with regulatory imposed measures is partially speculative.

This agreement with the reviewer largely due to the fact that atmospheric dust is the net contribution of both natural and anthropogenic dust emissions. More specifically, anthropogenic activity over erodible soils or areas of little vegetation, involving among others transportation, infrastructure, building and road construction (Moulin and Chiapello, 2006; Chen et al., 2018), deterioration of extended soil surfaces, and changes in land use due to deforestation, grazing (Ginoux et al., 2012), urbanization, and agriculture (Tegen et al., 1996) results in locally emitted anthropogenic dust into the atmosphere. However, such anthropogenic emissions processes lack the strength of natural dust emission/mobilization mechanisms, such as of dust devils (Koch and Renno, 2005; Ansmann et al., 2009), "haboobs" (Knippertz et al., 2007), pressure gradients (Klose et al., 2010), and low-level jets (LLJs; Fiedler et al., 2013), with the anthropogenic dust emissions frequently confined within the PBL and lacking the strength/capacity to penetrate the boundary layer and be released into the free troposphere, such in the case of natural dust mobilization/emission/transport processes. At present, though the current consensus is that atmospheric dust is composed of natural dust and anthropogenic dust, one of the challenging and pressing scientific questions in the case of atmospheric dust, evolves around the relative percentages attributed to each category, natural or anthropogenic (Penner et al., 1994; Tegen and Fung, 1995; Ginoux et al., 2012). For instance, Ginoux et al., 2012 quantified that ~25% of the total dust aerosol load is related to anthropogenic activities, a percentage that recent more studies report as underestimation when it comes to the case of densely populated and heavily industrialized urban areas (such in the case of the present study), where reported anthropogenic dust possible accounts for as much as ~ 70 % of the total dust load (Huang et al., 2015; Chen et al., 2019).

[Figure]

Figure 02: Global maps of statistically significant temporal trends at 95% confidence level at 1◦×1◦ spatial resolution for AOD (a) and DOD (b) MODIS observations during 2003–2017. (Source: Logothetis et al., 2021).

Regulatory policies/measures regulate anthropogenic activities and as such anthropogenic aerosol emissions, extending to anthropogenic activities resulting to emission of anthropogenic dust. As such, impact on anthropogenic dust emissions can be expected. For instance, a recent study by Logothetis et al. (2021) investigated both EO-based AOD and DOD for the period 2003–2017 at a global scale and reported that DOD trends in general follow AOD trends, though frequently are not characterized by similar magnitude of statistical significance (Fig. 02). It is reminded though that when the analysis does not return statistically different conclusions (e.g., p > 0.05) the research can only claim that no evidence of a statistical significance is found, not that there are not statistically significant features in the time-series (Harris et al., 2012). Absence of evidence of statistical significance is not equivalent to evidence of absence of statistical significance (Altman et al., 1996; Wellek et al., 2010).

Since the total atmospheric dust load is the superimposition of natural and anthropogenic dust load, the authors agree with the reviewer that the argument that the reduction of the total atmospheric dust load possible is the outcome of imposed regulatory measures cannot be made with high confidence since it would be to a degree speculative, since such regulation measures would have a more direct impact on the anthropogenic dust emissions and only indirectly to the natural dust emissions (i.e. deforestation). Such conclusion, in order to be strengthen would not only require a policy impact assessment, as the reviewer mentions, but in addition in order to account also for natural dust emissions, longer and more advanced time series analysis would be required, including for instance (among others) patterns of raining, wind, and in general ECVs, to demonstrate how specific air quality policies temporally align with reductions of both natural and anthropogenic dust levels.

As such, towards clarifying the above concept and in agreement with the reviewer's comment, in addition to going through the manuscript and working/removing possible sources leading to this misinterpretation of the outcomes, the following text is included in the beginning in the section "Atmospheric dust and Air Quality - 16 years of Earth Observations (2007-2022)":

"*At this point it should be emphasized that the documented EO-based atmospheric dust trends over the densely populated and heavily industrialized large cities and megacities of the world reported in the present study do not necessary reflect the effectiveness (or not) of regulatory measures as such a causal relationship between air quality regulatory policies/measures and air quality improvements in terms of dust would be to a degree speculative. More specifically, the current consensus is that atmospheric dust is the net contribution of natural dust and anthropogenic dust (Penner et al., 1994; Tegen and Fung, 1995; Ginoux et al., 2012). Natural dust is emitted in the atmosphere by mobilizing mechanisms including mainly dust devils (Koch and Renno, 2005), haboobs (Knippertz et al., 2007), pressure gradients (Klose et al., 2010), and low-level jets (LLJs; Fiedler et al., 2013) over arid regions (Prospero et al., 2002). Anthropogenic dust is emitted into the atmosphere by anthropogenic activities including mainly transportation, infrastructure and construction (Moulin and Chiapello, 2006; Chen et al., 2018), deterioration of soil surfaces and modifications in land use (i.e., deforestation), grazing (Ginoux et al., 2012), urbanization and agriculture (Tegen et al., 1996). To date anthropogenic dust is considered a significant proportion of the total aerosol load though of high spatial and temporal variability, with estimations ranging between ~20% (Ginoux et al., 2012) and as high as ~70% in the case of densely populated and heavily industrialized urban areas (Huang et al., 2015; Chen et al., 2019). Regulatory policies and measures regulate anthropogenic activities and as such anthropogenic aerosol emissions, extending to anthropogenic activities resulting to emission of anthropogenic dust. However, since the total atmospheric dust load is the superimposition of natural and anthropogenic dust and since regulation measures would have a more direct impact on the anthropogenic dust emissions and only indirectly to natural dust emissions, conclusions on reductions of the total atmospheric dust load as outcome of imposed regulatory measures cannot be made with high confidence and would be to a degree speculative.*".*

2) Inconsistencies between EO-based dust mass concentrations and ground-based observations. The paper relies heavily on EO-derived dust mass concentrations from the ESA-LIVAS CDR product. However, it does not provide a systematic validation or cross-comparison with in-situ measurements (e.g., ground-based PM10/PM2.5 data from air quality monitoring stations) in urban centers for any of the cities. This raises concerns about potential mismatches between satellite-derived values and actual exposure levels experienced at the surface, especially given known limitations of satellite products in resolving near-surface conditions over complex urban terrains. A discussion on these discrepancies, or at least quantitative uncertainty bounds, would improve the transparency and reliability of the results. Some of the projections for southern European cities including Madrid or Barcelona, which share quite common atmospheric phenomenology, display very contrasted PM2.5 dust concentrations, in one case unrealistically high, the other unbelievably low. In this line, current (2006-2022) dust values in PM-coarse and PM2.5 are not coincident with experimental values provided extensively in the literature.

The authors appreciate the reviewer's observation and the opportunity given to clarify even more the rationale for using the CALIPSO-based LIVAS climate data record, established in the framework of ESA activities, rather than relying exclusively on ground-based $PM_{10}/PM_{2.5}$ monitoring networks. To begin with, it is essential to emphasize that the authors agree with the reviewer's comment and fully acknowledge that in-situ measurements of particulate matter represent the most direct and reliable source of information on ambient air quality, the reference, allowing for high temporal resolution measurement, low detection limits, and the ability to distinguish between $PM_{10}$, $PM_{2.5}$ and even finer size fractions with high precision. These PM in-situ measurements are well-established reference methods, widely used for regulatory purposes, and are indispensable for health impact assessments, epidemiological studies, and local air quality management. Compared to earth observation satellite-based products, in-situ datasets provide unparalleled accuracy at the surface, direct relevance to human exposure, and the possibility to capture short-term variability such as diurnal cycles or pollution peaks.

Nevertheless, despite these advantages, surface monitoring networks (i.e., OpenAQ, IQAir, SPARTAN) are not uniformly available across the globe. Even in the case of large cities and megacities (such as in the case of the present study), particularly in Africa, parts of Asia, and the Middle East, stations are sparse or even completely absent, and even where networks exist, there are often gaps in temporal coverage due to instrument operation, maintenance, or resource limitations. Moreover, across the globe, different types of instruments and measurement protocols frequently introduce inconsistencies across regions, while spatial representativeness remains limited, as most stations are confined to specific urban environments and may not adequately capture variability even within a metropolitan or larger city area. The authors are aware that the present study would be more robust and accurate if it could rely on dense, continuous in-situ observations for all cities considered, incorporating several in-situ station measurements for each large city and/or megacity. However, in the absence of such comprehensive ground-based in-situ measurements, the present study makes use of satellite-derived earth observation products, with focus on the dust aerosol component, which offer unique advantages in terms of spatial consistency, global coverage, and the ability to provide long-term, homogeneous datasets across large regions where monitoring networks are incomplete or even completely absent, though with lower accuracy than the accuracy offered by in-situ measurements.

The "LIdar climatology of Vertical Aerosol Structure for space-based lidar simulation studies" (LIVAS) Climate Data Record (CDR), developed under the European Space Agency (ESA) primarily on the basis of CALIPSO/CALIOP lidar measurements provides a global, long-term multiyear dataset of dust aerosol vertical distributions, including information on the dust fine-mode and dust coarse-mode components. The EO-based products such as the aforementioned ESA-LIVAS dust CDR, allow for a consistent assessment of dust-related aerosol contributions over extended temporal scales and across multiple urban domains, something not achievable through ground networks alone at present. More importantly, these satellite datasets are not used in isolation: their underlying algorithms and retrieval approaches have been subject to extensive validation and evaluation against ground-based, airborne, and satellite-based measurements in multiple regions during the past two decades and documented uncertainties are available. The approach of the well-established algorithms allowing the establishment of the final ESA-LIVAS dust CDR have followed during the past two decades the approach of -in general- algorithm (technical) developments including validation/evaluation activities to be addressed in separate studies and publications than the studies reporting on the outcomes of the implementation of the final products to address a number of scientific questions of the earth system (Table 01).

More specifically, the POLIPHON (Polarization Lidar Photometer Networking; Tesche et al., 2009; Ansmann et al., 2012, 2019; Mamouri and Ansmann, 2014, 2017) method, initially proposed in terms of equations by Shimizu et al. (2004), constitutes a well-established family of sophisticated techniques developed to separate dust and non-dust aerosol components from the total aerosol load, further extended to separate the fine-mode and coarse-mode dust components in terms of optical properties, incorporating advances allowing retrieving height profiles of dust particle mass, and (among others) dust-related cloud condensation nuclei (CCN) and ice-nucleating particle (INP) concentrations (Mamouri and Ansmann, 2015, 2016; Marinou et al., 2019; Ansmann et al., 2019; Floutsi et al., 2023). POLIPHON mainly focuses on atmospheric dust, leveraging on the optical property of irregular particles to perturb the polarization state of lidar-emitted polarized light pulses (Freudenthaler et al., 2009). The POLIPHON algorithm was established and refined within the framework of extensive experimental campaigns such as the Saharan Mineral Dust Experiment (SAMUM-1 and SAMUM-2; Ansmann et al., 2011; Heintzenberg, 2009) and the Saharan Aerosol Long-range Transport and Aerosol-Cloud-Interaction Experiment (SALTRACE; Weinzierl et al., 2017; Mamouri and Ansmann, 2016, 2017). These campaigns involved ground-based, airborne, and remote sensing observations, with a strong focus on vertical profiling of atmospheric dust (Ansmann et al., 2011; Weinzierl et al., 2017). The method also utilizes climatologically robust relationships between aerosol-type-dependent optical properties and desired environmental and cloud-relevant aerosol parameters, which are derived from the global Aerosol Robotic Network (AERONET) database (Ansmann et al., 2012; Mamouri and Ansmann, 2017; Ansmann et al., 2019; He et al., 2023;

He et al., 2025). Initially, the POLIPHON method was primarily applied to ground-based lidar systems, including those within the European Aerosol Research Lidar Network (EARLINET) (Ansmann et al., 2012) and PollyNET (Baars et al., 2016; Floutsi et al., 2023), using PollyXT systems (Engelmann et al., 2016). Subsequently, its principles have been successfully adapted and applied to satellite lidar systems, such as CALIPSO (Cloud-Aerosol Lidar and Infrared Pathfinder Satellite Observations) (Amiridis et al., 2013; Marinou et al., 2017; Proestakis et al., 2018; 2024; Aslanoğlu et al., 2022) and the ISS-CATS (International Space Station – Cloud Aerosol Transport System) (Proestakis et al., 2024), while at present are applied to EarthCARE (Earth Cloud Aerosol and Radiation Explorer).

Table 01: (Indicative) studies closely related to development, improvements, applications, accuracy, and uncertainties of the lidar-based atmospheric dust algorithms and products (alphabetically listed).

| Ground-based / Air-borne / Satellite-based (In situ / Columnar / Profiling) | Amiridis et al., 2013; 2015 /Ansmann et al., 2011; 2012; 2019 / Floutsi et al., 2023 / Georgoulias et al., 2020 / Groß et al., 2015 / Haarig et al., 2017a, 2017b; 2019 / He et al., 2023; 2025 / Hofer et al., 2017 / Järvinen et al., 2016 / Mamouri and Ansmann, 2014; 2015; 2016; 2017 / Mamali et al., 2018 / Marinou et al., 2017; 2019 / Proestakis et al., 2018; 2024 |
| --- | --- |

As mentioned, the POLIPHON lidar-based dust products, during the past two decades, have undergone through several validation and evaluation activities, aiming not only towards establishing the uncertainties of the products, but also aiming towards further advances. Towards these objectives, extensive validation and evaluation efforts have been undertaken to confirm the quality and reliability of POLIPHON's dust products and algorithm developments, especially upon application to (satellite-based) EOs. More specifically, towards CALIPSO dust retrievals, early validation/evaluation efforts demonstrated improvements in CALIPSO dust extinction retrievals (Amiridis et al., 2013). Upon comparison against concurrent dust-dominant observations derived in part from AERONET measurements and MODIS (Moderate-Resolution Imaging Spectroradiometer) onboard Aqua (as part of A-Train observations), Amiridis et al. (2013) demonstrated (upon application of suitable geographical-dependent parametrization and optical properties ratios) optimization of CALIPSO Saharan dust retrievals. In general, during the past two decades in the framework of the developments included in the family of lidar-based dust product, AERONET observations have been regularly implemented for quality assurance in POLIPHON, providing column-integrated values of total, fine-mode, and coarse-mode optical properties to check consistency with lidar-derived results (Mamouri and Ansmann, 2014; 2017; Hofer et al., 2017; Marinou et al., 2017; Proestakis et al., 2018; 2024; Ansmann et al., 2019), consistency checks crucial given the numerous assumptions in POLIPHON dust retrievals. Such evaluations of the dust products have focused for instance over the vast areas affected by Saharan dust aerosol intrusions (Marinou et al., 2017), over areas in South and East Asia affected by the vast Taklimakan, Gobi, and Thar deserts (Proestakis et al., 2018; 2024), or over the Arabian Peninsula (Mamouri et al., 2013; Nisantzi et al., 2015), demonstrating the consistency of polarization lidar dust products for proper quantification of the atmospheric dust load. This is crucial, due to the fact that AERONET data constitute a global-coverage refence in terms of atmospheric observations, thus AERONET-based quality assurance activities in terms of the performed POLIPHON dust developments have been -and are- central towards determining the uncertainties and establishing climatologically robust conversion parameters that transform lidar-derived aerosol extinction coefficients into microphysical properties like number, surface area, and volume concentrations for dust and non-dust aerosols globally (Mamouri and Ansmann, 2017; Ansmann et al., 2019; He et al., 2023; He et al., 2025).

In addition to establishing the accuracy of POLIPHON dust optical depth (DOD) against AERONET and MODIS dust-dominant aerosol optical depth (AOD) over concurrent observations, several efforts have incorporated lidar-based profiling, towards addressing not only the accuracy of columnar products but in addition the accuracy of the provided vertical structure of dust aerosols in the atmosphere. The efforts have focused mainly (but not restricted) to backscatter and extinction coefficient profiles, addressing the capacity of the algorithms to provide (among others) information on the close-to-the-earth's-surface dust aerosol load. Such activities range from frequently high-quality lidar observations and intercomparisons, as programmed within the framework of EARLINET (Pappalardo et al., 2014) and POLYNET (Baars et al., 2016; Engelmann et al., 2016), to multi-partner extensive experimental campaigns. Quality assurance activities of POLIPHON-based dust products including validation against ground-based lidar systems have been frequently performed. POLIPHON dust algorithms have been applied and tested to observations from PollyXT systems within PollyNET, during campaigns like SAMUM and SALTRACE, and to observations from different multiwavelength Raman/polarization lidar systems (e.g., BERTHA, POLIS, MULIS), and intercompared, reporting overall good agreement in key dust optical properties (Tesche et al., 2009a, 2022; Groß et al., 2015; Haarig et al., 2017a, 2017b). Moreover, ground-based Raman lidar measurements over Cabo Verde during SAMUM were

utilized to validate CALIPSO pure dust observations, initially reporting an underestimation of CALIPSO Level 2 extinction coefficients due to a low lidar ratio assumption (Tesche et al., 2013). These discrepancies were further explored in relation to the presence of large particles not accounted for in the CALIPSO aerosol model (Wandinger et al., 2010). In addition, the consistency between the one-step (dust/non-dust separation) and two-step (fine/coarse dust separation) POLIPHON methods has been demonstrated (Mamouri and Ansmann, 2017). More specifically, although the two-step method involves more assumptions, careful application ensured that the sum of fine and coarse dust backscatter from the two-step method approximately matches the total dust backscatter from the one-step method, serving as a crucial quality assurance check (Mamouri and Ansmann, 2017).

More crucial is that the POLIPHON methods and products have undergone rigorous validations against in situ (mainly airborne) measurements of dust. As discussed, the authors fully acknowledge that in-situ measurements, such the provided by OpenAQ, IQAir, SPARTAN, and the WHO Ambient Air Quality Database, provide unparalleled quality in terms of the provided measurements, enabling researchers, policymakers, and the public to track the aerosol load over time, identifying  pollution hotspots, evaluating the effectiveness of environmental regulations, supporting integration with the scientific community and research, allowing public health studies and air quality forecasting, empowering evidence-based decision-making, and complementing satellite observations, and improving the accuracy of EO-based products, such as the accuracy of POLIPHON dust products. However, despite these strengths, global PM networks have to-date not yet applied towards such a validation of the EO-based dust product for a number of reasons. Despite the high quality of the measurements, frequently the coverage of PM networks is uneven, with many regions, particularly in low-income countries, having sparse or no monitoring stations, which can lead to gaps in data and underrepresentation of the true extent of air pollution, even when it comes to large cities or megacities of the world, such as in the present study. In addition, differences in measurement methods, instrument calibration, and data reporting standards between stations can introduce inconsistencies, complicating direct comparisons across locations, while it may happen that stationary ground-based monitors may not fully capture localized variations in pollution within urban areas or near specific emission sources.

However, these inherited challenges of EO-based products intercomparison against different in-situ measurements do not translate that the algorithms and the dust products have not been assessed against in-situ measurements. When it comes to comparison of the dust products against in-situ measurements and validation of the quality of these products, these necessary activities have still been performed in the framework of extensive experimental campaigns of multi-instrument synergies, airborne in-situ, or chamber experiments.

As examples of validation against in-situ and airborne measurements, the SAMUM-1 and SAMUM-2 campaigns included extensive closure experiments comparing field observations (ground-based, airborne) with modeling results, focusing on the relationship between chemical composition, shape morphology, size distribution, and optical effects of dust (Ansmann et al., 2011). These efforts allowed in addition for direct comparison of AERONET dust products with lidar and in-situ observations for pure dust, showing good agreement for particle extinction and Ångström exponent, while also revealing differences in particle size distributions and absorption properties (Müller et al., 2010). In addition, the foundational work of Sakai et al. (2010) and Järvinen et al. (2016) has been central in establishing and enhancing the parameterization of dust, particularly the backscatter linear depolarization ratio, which is critical for lidar-based aerosol classification schemes like POLIPHON (Mamouri and Ansmann, 2014, 2017; Proestakis et al., 2024). Sakai et al. (2010) and Järvinen et al. (2016) pioneering laboratory chamber experiments provided crucial in-depth understanding and solidified the basis for the two-step POLIPHON methodology of the separation of fine-mode, coarse-mode, and non-dust aerosol components using polarization lidar observations (Mamouri and Ansmann, 2014; 2017). These outcomes additionally enhanced the confidence of, refining the POLIPHON analysis schemes. These combined laboratory efforts, corroborated by recent field observations (Hofer et al., 2017; Mamouri and Ansmann, 2017), have provided a robust foundation for distinguishing fine and coarse dust, thereby ensuring the high quality and reliability of the final dust products and the continuous developments. As examples, during Saharan dust intrusion episodes over Cyprus, vertical profiles of aerosol mass concentration from POLIPHON showed agreement within experimental uncertainty with measurements from optical particle counters (OPCs) on unmanned aerial vehicles (UAVs), especially for the coarse mode (Mamali et al., 2018). Similarly, POLIPHON-derived profiles of dust mass concentration, CCN, and INP concentrations have been successfully compared and validated with independent in-situ measurements, establishing the accuracy of the products (Mamouri and Ansmann, 2016; Marinou et al., 2019; Ansmann et al., 2019). Comparisons of lidar retrievals with airborne in-situ observations during SALTRACE (Barbados) demonstrated the capability of lidars to predict dust properties, with high agreement found for the fine-mode and coarse-mode dust mass concentrations (Weinzierl et al., 2017; Haarig et al., 2019; Samaras et al., 2020). In terms of airborne in-situ observations (Fig. 03), validation of CALIPSO-based fine-mode and coarse-mode pure-dust mass concentration products against airborne in-situ particle size distributions from the FAAM BAe-146 research

aircraft during the AER-D campaign (Ryder et al., 2018) showed good qualitative and quantitative agreement, especially for the coarse-mode pure-dust mass concentration (Proestakis et al., 2024).

[Figure]

Figure 03: AER-D/ICE-D FAAM b920–ISS-CATS underflight on the 7th of August 2015 in the vicinity of Praia, Cabo Verde and satellite-based observations of a Saharan dust advection over the Atlantic Ocean (upper panel), FAAM b920 airborne in situ measurements of mass concentration in the vicinity of Praia on 7 August 2015 based on (averaged) P2 and P7 (a). FAAM b920 in situ (black line) and ISS-CATS (orange line) mass concentration profiles are provided for the total (b), supermicrometer (c), and submicrometer atmospheric aerosol classes (d). (Sources: Ryder et al., 2018; Proestakis et al., 2024).

These multi-faceted validation and evaluation efforts consistently demonstrate the high quality of the final dust products and the continuous developments of the POLIPHON algorithms. The ability to reduce biases in satellite-derived dust optical products and to robustly separate dust from non-dust components, and further distinguish between fine and coarse dust components, support the algorithm's strength. Furthermore, the observed consistency between lidar-derived parameters and independent measurements from photometers, in-situ sensors, and other lidar systems corroborates the reliability of POLIPHON.

These advancements and ongoing validation and evaluation efforts are crucial for a better understanding the dust products. In recognition to the value of the comment and in agreement with the reviewer the following discussion is included in the manuscript:

[revised manuscript text omitted]

3) Overgeneralization of future exposure projections without accounting for uncertainty. The projections of future dust exposure and population impacts are presented with a high degree of numerical precision (e.g., 509.7 million people affected, 113.1% increase), but the uncertainties associated with both future dust concentrations and demographic changes are not adequately discussed. This gives a false sense of certainty in the forecasts. The study would benefit from including scenario-based projections that reflect the range of plausible futures, particularly considering how climate change, land use, and urban development might alter dust emissions and transport.

Based on the LIVAS 4D global dust climate data record, established under ESA activities on the basis of sixteen years of CALIPSO earth observations (12/2006-11/2022), the present study followed the following steps summarized in Fig. 04 in an attempt to address (among others) the question "how are the mass concentrations of coarse-mode and fine-mode atmospheric dust components within the PBL expected to evolve in the near future?".

[Figure]

Figure 04: Conceptual/methodological approach towards addressing one of the scientific questions of the present work, related to provide insight into possible changes on the evolution of the coarse and fine mode components of atmospheric dust components within the PBL over large cities of the world in the near future.

Addressing this question on the atmospheric dust as hazard on human health is of significance towards anticipating future air quality challenges, supporting the development of targeted mitigation strategies, and protecting public health in rapidly growing urban environments. As an example, this necessity is reflected to the latest EU ambient air quality directive (AAQD - 2024/2881) where it is emphasized that it is of significance to analyse the atmospheric aerosol levels in different areas (i.e., urban background locations, air pollution hotspots, industry related locations, traffic related locations) and assess the possible contribution from long-range transport of pollutants – contribution from natural sources, support source apportionment analysis, and enhance the understanding of specific pollutants such as particulate matter.

Towards this objective, timeseries of the (coarse- and fine- modes) of the atmospheric dust load close to the earth's surface in terms of particulate matter future projections beyond the observation period are computed. It has to be emphasized, which has been included in the manuscript for clarity and towards avoiding misinterpretation of the outcome conclusions, that the projections represent expected PBL dust concentrations changes under the assumption of continuation of 2007–2022 trends. Accounting for these forecast-uncertainties associated with future dust concentrations confidence intervals of the projections are computed on the basis of each EO-based timeseries and included in the discussion outcomes (Sect. 5: Fig. 15, Tables 4 & 5). It is emphasized that the reliability of these projections decreases with increasing time beyond the observational temporal period, and if the case they are not constrained the increasing uncertainty of the dust PM projection forecast would result inevitable to unrealistic conclusions.

To account for the impact of the increasing uncertainty and the discussed negative possibility the projection period is constrained to the temporal period prior when:

i.   the confidence interval first intersects zero (yielding non-physical negative concentration values), *or*
ii.  the confidence interval variability equals or exceeds the variability characterizing the original EO dataset, *or*
iii. the confidence interval reaches values as high as the projected values themselves.

For instance, for the indicative case of the Beijing metropolitan area, projections of the PBL coarse-mode and fine-mode dust mass concentration reports, on the basis of the above-mentioned constrains, are provided with relatively high confidence values until the year 2029 (Sect. 5: Fig. 15). These projections are depicted as red trend lines, with the surrounding light red shaded areas denoting the associated confidence intervals.

The year of interrupting the projection varies with the special characteristics of the timeseries established by EO-based observations and products, which vary according to the different characteristics of each urban area. However, the forecast refers to near-future projections, temporarily close to the timeseries of EOs. The timeseries rarely reach the third decade of the 21$^{st}$ century, as reported for each urban area and for each fine- and coarse- mode components in Tables 4 and 5 of Sect.5. An impact attributed to the sixteen years of EOs, which is not long enough as a timeseries to provide insight into a longer period or further decades than the near future.

This reference that the work does not addresses mass concentration changes far beyond the EO-based time-series, not far beyond the near future, comes into argument that a scenario-based projection of policies with reference to different plausible futures would not provide high insight.

A recent study published by Liu et al. (2024) in Nature, investigated future projections of global dust burden from ten bias-corrected initial Climate Models Intercomparison Project Phase Six (CMIP6) models, under four scenarios of shared socioeconomic pathways (SSPs). More specifically, the study reported on future projections, in global bias-corrected dust mass load (DML) over the near-term (2021–2040), medium-term (2051–2070), and long-term (2081–2100), relative to a reference period (2000–2014) under the SSP126, SSP245, SSP370, and SSP585 scenarios. According to the conclusions, reporting on the scenario uncertainty in projections of multidecadal mean DML based on bias-corrected CMIP6 models, for predictions of the near-term, medium-term (2051–2070), long-term (2081–2100). According to the outcomes of the study (Fig. 05), the scenario uncertainty is rarely an important factor in triggering uncertainty, especially in the near-term (2021–2040), while the scenario uncertainty becomes an important source of uncertainty by the end of the century, contributing up to 40% in the projections (Liu et al., 2024; Fig.8). Moreover, the study demonstrated that the bias corrected CMIP6 models in terms of DML significantly deviated among them under different SSPs beyond the near-term (2021–2040) period (Liu et al., 2024; Fig.4). The near-term (2021–2040) period is the period that well-encompasses and well-extends beyond the time-period of dust PM projection of the EO-based timeseries of the presents study, and the period that according to the CMIP6 SSPs projections the different scenarios do not have observable impacts yet.

[Figure]

Figure 05: Proportion of uncertainty of three uncertainty sources (internal variability, model uncertainty and scenario uncertainty) in projections of multidecadal mean DML based on bias-corrected CMIP6 models. The columns show the total variance explained by (left) internal variability, (middle) model uncertainty, and (right) scenario uncertainty for predictions of the (a–c) near-term (2021–2040), (d–f) medium-term (2051–2070), and (g–i) long-term (2081–2100) (Source: Liu et al., 2024; Fig.8).

Finally, it should be mentioned that following the reviewer's recommendation the authors have gone through the manuscript and ensured consistency in all reported approximation values provided, with no higher accuracy than to the first decimal point.

4) Lack of consideration of non-dust PM sources in health risk estimations. The study attributes PM2.5 and PM-coarse related health risks in major cities exclusively to mineral dust exposure, using dust-specific mass concentrations derived from EO data. However, in many urban environments, anthropogenic PM sources (traffic, industry, biomass burning) dominate the PM burden. Without source apportionment or a method to disentangle dust from other PM components, the derived Relative Risk and Attributable Fraction metrics might overestimate the health impacts attributable to dust alone. Clarifying this distinction, or acknowledging the limitations of the approach, would improve the robustness of the health impact assessment.

Indeed, in our approach there is a significant assumption: that the air breathed by the exposed populations comprises entirely of dust particles, either in the fine ($PM_{2.5}$) or in the coarse mode ($PM_{10}$). This was a necessary step in order to apply equations 1 and 2. Nevertheless we support our choice, as the baseline values (10 $\mu g\ m^{-3}$ for $PM_{10}$, 3 $\mu g\ m^{-3}$ for $PM_{2.5}$) include by default the total mass of suspended PM which by looking Tables 4 or 5? are usually lower than the derived dust concentrations in this work. Since the reported estimates are assigned only to dust, then concentrations including all other sources would be much higher. These observations suggest that the derived Relative Risk and Attributable Fraction metrics do not overestimate the health impacts rather than there might be an underestimation. Let's just have an example. In Hopke et al. (2022), which is a review paper, proportion values of several sources from PMF results are reported on a global scale. For $PM_{10}$, dust proportion varies between 1 – 58% and for $PM_{2.5}$ it varies between 2.4 - 20%. Applying these proportions (min – max) to the baseline concentrations for the estimation of RR and AF, and taking as an example Karachi concentration (Tables 7-9) produces the values reported in the following table, and are all higher.

Table 02: R1 - Estimates of RR and AF for assigned dust particle concentrations.

| PM2.5 | | Karachi conc. | RRcardio | RRlung | %AFcardio | %AFlung |
|---|---|---|---|---|---|---|
| Baseline=3 $\mu g\ m^{-3}$ | Assigned basel. | 113.49 $\mu g\ m^{-3}$ | 1.683 | 2.179 | 40.57 | 54.10 |
| 2.4% | 3*0.024=0.072 | | 2.064 | 2.958 | 51.55 | 66.19 |
| 20% | 3*0.2=0.6 | | 1.939 | 2.695 | 48.44 | 62.89 |
| PM10 | | | RR | %AF | | |
| Baseline=10 $\mu g\ m^{-3}$ | Assigned basel. | 85.95 $\mu g\ m^{-3}$ | 1.063 | 5.89 | | |
| 1% | 10*0.01=0.1 | | 1.071 | 6.64 | | |
| 58% | 10*0.58=5.8 | | 1.066 | 6.21 | | |

For specific regional domains, important references for dust occurrence and air quality outcomes, PM composition data or health impacts are missing. Some examples for the southern European case are here:
1. African dust contributions to mean ambient PM10 mass-levels across the Mediterranean Basin, by Querol et al. 2009.
2. Pey, J., Querol, X., Alastuey, A., Forastiere, F., and Stafoggia, M.: African dust outbreaks over the Mediterranean Basin during 2001–2011: PM10 concentrations, phenomenology and trends, and its relation with synoptic and mesoscale meteorology, Atmos. Chem. Phys., 13, 1395–1410, https://doi.org/10.5194/acp-13-1395-2013, 2013.
3. African dust and air quality over Spain: Is it only dust that matters? by Querol et al. 2019.
4. Stafoggia M, Zauli-Sajani S, Pey J, Samoli E, Alessandrini E, Basagaña X, Cernigliaro A, Chiusolo M, Demaria M, Díaz J, Faustini A, Katsouyanni K, Kelessis AG, Linares C, Marchesi S, Medina S, Pandolfi P, Pérez N, Querol X, Randi G, Ranzi A, Tobias A, Forastiere F; MED-PARTICLES Study Group. Desert Dust Outbreaks in Southern Europe: Contribution to Daily PM10 Concentrations and Short-Term Associations with Mortality and

Hospital Admissions. Environ Health Perspect. 2016 Apr;124(4):413-9. doi: 10.1289/ehp.1409164. Epub 2015 Jul 24. PMID: 26219103; PMCID: PMC4829979.

We would like to thank the reviewer for reminding us of these studies, which have been included in the manuscript for providing a more complete view of the high value of ground-based in-situ measurements.

**References included in the "Response to Reviewer"**

Abera, A., Friberg, J., Isaxon, C., Jerrett, M., Malmqvist, E., Sjöström, C., Taj, T., and Vargas, A. M.: Air Quality in Africa: Public Health Implications, Annu. Rev. Publ. Health, 42, 193–210, https://doi.org/10.1146/annurev-publhealth-100119-113802, 2020.

Altman, D. and Bland, J.: Absence of evidence is not evidence of absence, Australian Veterinary Journal, 74, 311–311, https://doi.org/10.1111/j.1751-0813.1996.tb13786.x, 1996.

Amiridis, V., Wandinger, U., Marinou, E., Giannakaki, E., Tsekeri, A., Basart, S., Kazadzis, S., Gkikas, A., Taylor, M., Baldasano, J., and Ansmann, A.: Optimizing CALIPSO Saharan dust retrievals, Atmos. Chem. Phys., 13, 12089–12106, https://doi.org/10.5194/acp-13-12089-2013, 2013.

Amiridis, V., Marinou, E., Tsekeri, A., Wandinger, U., Schwarz, A., Giannakaki, E., Mamouri, R., Kokkalis, P., Binietoglou, I., Solomos, S., Herekakis, T., Kazadzis, S., Gerasopoulos, E., Proestakis, E., Kottas, M., Balis, D., Papayannis, A., Kontoes, C., Kourtidis, K., Papagiannopoulos, N., Mona, L., Pappalardo, G., Le Rille, O., and Ansmann, A.: LIVAS: a 3-D multi-wavelength aerosol/cloud database based on CALIPSO and EARLINET, Atmos. Chem. Phys., 15, 7127–7153, https://doi.org/10.5194/acp-15-7127-2015, 2015.

Ansmann, A., Tesche, M., Knippertz, P., Bierwirth, E., Althausen, D., Müller, D., and Schulz, O.: Vertical profiling of convective dust plumes in southern Morocco during SAMUM, Tellus B, 61, 340–353, https://doi.org/10.1111/j.1600-0889.2008.00384.x, 2009.

Ansmann, A., Tesche, M., Seifert, P., Gross, S., Freudenthaler, V., Apituley, A., Wilson, K. M., Serikov, I., Linne, H., Heinold, B., Hiebsch, A., Schnell, F., Schmidt, J., Mattis, I., Wandinger, U., and Wiegner, M.: Ash and fine-mode particle mass profiles from EARLINET-AERONET observations over central Europe after the eruptions of the Eyjafjallajokull volcano in 2010, J. Geophys. Res.-Atmos., 116, D00U02, https://doi.org/10.1029/2010JD015567, 2011a.

Ansmann, A., Petzold, A., Kandler, K., Tegen, I., Wendisch, M., Müller, D., Weinzierl, B., Müller, T., and Heintzenberg, J.: Saharan Mineral Dust Experiments SAMUM–1 and SAMUM–2: what have we learned?, Tellus B: Chemical and Physical Meteorology, 63, 403–429, https://doi.org/10.1111/j.1600-0889.2011.00555.x, 2011b.

Ansmann, A., Seifert, P., Tesche, M., and Wandinger, U.: Profiling of fine and coarse particle mass: case studies of Saharan dust and Eyjafjallajokull/Grimsvotn volcanic plumes, Atmos. Chem. Phys., 12, 9399–9415, https://doi.org/10.5194/acp-12-9399-2012, 2012.

Ansmann, A., Mamouri, R.-E., Hofer, J., Baars, H., Althausen, D., and Abdullaev, S. F.: Dust mass, cloud condensation nuclei, and ice-nucleating particle profiling with polarization lidar: updated POLIPHON conversion factors from global AERONET analysis, Atmospheric Measurement Techniques, 12, 4849–4865, https://doi.org/10.5194/amt-12-4849-2019, 2019.

Aslanoğlu, S. Y., Proestakis, E., Gkikas, A., Güllü, G., and Amiridis, V.: Dust Climatology of Turkey as a Part of the Eastern Mediterranean Basin via 9-Year CALIPSO-Derived Product, Atmosphere, 13, https://doi.org/10.3390/atmos13050733, 2022.

Baars, H., Kanitz, T., Engelmann, R., Althausen, D., Heese, B., Komppula, M., Preissler, J., Tesche, M., Ansmann, A., Wandinger, U., Lim, J.-H., Ahn, J. Y., Stachlewska, I. S., Amiridis, V., Marinou, E., Seifert, P., Hofer, J., Skupin, A., Schneider, F., Bohlmann, S., Foth, A., Bley, S., Pfuller, A., Giannakaki, E., Lihavainen, H., Viisanen, Y., Hooda, R. K., Pereira, S. N., Bortoli, D., Wagner, F., Mattis, I., Janicka, L., Markowicz, K. M., Achtert, P., Artaxo, P., Pauliquevis, T., Souza, R. A. F., Sharma, V. P., van Zyl, P. G., Beukes, J. P., Sun, J., Rohwer, E. G., Deng, R., Mamouri, R.-E., and Zamorano, F.: An overview of the first decade of Polly(NET): an emerging network of automated Raman-polarization lidars for continuous aerosol profiling, Atmos. Chem. Phys., 16, 5111–5137, https://doi.org/10.5194/acp-16-5111-2016, 2016.

Brooks, I. M. (2003). Finding boundary layer top: Application of a wavelet covariance transform to lidar backscatter profiles. Journal of Atmospheric and Oceanic Technology, 20(8), 1092–1105.

Chen, S., Jiang, N., Huang, J., Xu, X., Zhang, H., Zang, Z., Huang, K., Xu, X., Wei, Y., Guan, X., Zhang, X., Luo, Y., Hu, Z., and Feng, T.: Quantifying contributions of natural and anthropogenic dust emission from different climatic regions, Atmospheric Environment, 191, 94–104, https://doi.org/10.1016/j.atmosenv.2018.07.043, 2018.

Chen, S., Jiang, N., Huang, J., Zang, Z., Guan, X., Ma, X., Luo, Y., Li, J., Zhang, X., and Zhang, Y.: Estimations of indirect and direct anthropogenic dust emission at the global scale, Atmospheric Environment, 200, 50–60, https://doi.org/10.1016/j.atmosenv.2018.11.063, 2019.

DeMocker, M. J.: Benefits and Costs of the Clean Air Act 1990–2020: Revised Analytical Plan For EPA's Second Prospective Analysis, Ind. Econ. Inc. Cambridge, MA, https://www.epa.gov/sites/default/files/2015-07/documents/mainbody51203.pdf (last access: December 2022), 2003.

Engelmann, R., Kanitz, T., Baars, H., Heese, B., Althausen, D., Skupin, A., Wandinger, U., Komppula, M., Stachlewska, I. S., Amiridis, V., Marinou, E., Mattis, I., Linné, H., and Ansmann, A.: The automated multiwavelength Raman polarization and water-vapor lidar PollyXT: the neXT generation, Atmos. Meas. Tech., 9, 1767–1784, https://doi.org/10.5194/amt-9-1767-2016, 2016.

Fiedler, S., Schepanski, K., Heinold, B., Knippertz, P., and Tegen, I.: Climatology of nocturnal low-level jets over North Africa and implications for modeling mineral dust emission, J. Geophys. Res.-Atmos., 118, 6100–6121, https://doi.org/10.1002/jgrd.50394, 2013.

Floutsi, A. A., Baars, H., Engelmann, R., Althausen, D., Ansmann, A., Bohlmann, S., Heese, B., Hofer, J., Kanitz, T., Haarig, M., Ohneiser, K., Radenz, M., Seifert, P., Skupin, A., Yin, Z., Abdullaev, S. F., Komppula, M., Filioglou, M., Giannakaki, E., Stachlewska, I. S., Janicka, L., Bortoli, D., Marinou, E., Amiridis, V., Gialitaki, A., Mamouri, R.-E., Barja, B., and Wandinger, U.: DeLiAn – a growing collection of depolarization ratio, lidar ratio and Ångström exponent for different aerosol types and mixtures from ground-based lidar observations, Atmospheric Measurement Techniques, 16, 2353–2379, https://doi.org/10.5194/amt-16-2353-2023, 2023.

Freudenthaler, V., Esselborn, M., Wiegner, M., Heese, B., Tesche, M., Ansmann, A., MüLLER, D., Althausen, D., Wirth, M., Fix, A., Ehret, G., Knippertz, P., Toledano, C., Gasteiger, J., Garhammer, M., and Seefeldner, M.: Depolarization ratio profiling at several wavelengths in pure Saharan dust during SAMUM 2006, Tellus B: Chemical and Physical Meteorology, 61, 165–179, https://doi.org/10.1111/j.1600-0889.2008.00396.x, 2009.

Ganguly, T., Selvaraj, K. L., and Guttikunda, S. K.: National Clean Air Programme (NCAP) for Indian cities: Review and outlook of clean air action plans, Atmos. Environ., 8, 100096, https://doi.org/10.1016/j.aeaoa.2020.100096, 2020.

Ginoux, P., Prospero, J. M., Gill, T. E., Hsu, N. C., and Zhao, M.: Global-scale attribution of anthropogenic and natural dust sources and their emission rates based on MODIS Deep Blue aerosol products, Reviews of Geophysics, 50, https://doi.org/10.1029/2012RG000388, 2012.

Groß, S., Freudenthaler, V., Schepanski, K., Toledano, C., Schäfler, A., Ansmann, A., and Weinzierl, B.: Optical properties of long-range transported Saharan dust over Barbados as measured by dual-wavelength depolarization Raman lidar measurements, Atmospheric Chemistry and Physics, 15, 11067–11080, https://doi.org/10.5194/acp-15-11067-2015, 2015.

Haarig, M., Ansmann, A., Gasteiger, J., Kandler, K., Althausen, D., Baars, H., Radenz, M., and Farrell, D. A.: Dry versus wet marine particle optical properties: RH dependence of depolarization ratio, backscatter, and extinction from multiwavelength lidar measurements during SALTRACE, Atmos. Chem. Phys., 17, 14199–14217, https://doi.org/10.5194/acp-17-14199-2017, 2017a.

Haarig, M., Ansmann, A., Althausen, D., Klepel, A., Gross, S., Freudenthaler, V., Toledano, C., Mamouri, R.-E., Farrell, D. A., Prescod, D. A., Marinou, E., Burton, S. P., Gasteiger, J., Engelmann, R., and Baars, H.: Triple-wavelength depolarization-ratio profiling of Saharan dust over Barbados during SALTRACE in 2013 and 2014, Atmos. Chem. Phys., 17, 10767–10794, https://doi.org/10.5194/acp-17-10767-2017, 2017b.

Haarig, M., Walser, A., Ansmann, A., Dollner, M., Althausen, D., Sauer, D., Farrell, D., and Weinzierl, B.: Profiles of cloud condensation nuclei, dust mass concentration, and ice-nucleating-particle-relevant aerosol properties in the Saharan Air Layer over Barbados from polarization lidar and airborne in situ measurements, Atmospheric Chemistry and Physics, 19, 13773–13788, https://doi.org/10.5194/acp-19-13773-2019, 2019.

Harris, A. H. S., Fernandes-Taylor, S., and Giori, N.: "Not statistically different" does not necessarily mean "the same": the important but underappreciated distinction between difference and equivalence studies, J Bone Joint Surg Am, 94, e29, https://doi.org/10.2106/JBJS.K.00568, 2012.

He, Y., Yin, Z., Ansmann, A., Liu, F., Wang, L., Jing, D., and Shen, H.: POLIPHON conversion factors for retrieving dust-related cloud condensation nuclei and ice-nucleating particle concentration profiles at oceanic sites, Atmospheric Measurement Techniques, 16, 1951–1970, https://doi.org/10.5194/amt-16-1951-2023, 2023.

He, Y., Choudhury, G., Tesche, M., Ansmann, A., Yi, F., Müller, D., and Yin, Z.: Extended POLIPHON dust conversion factor dataset for lidar-derived cloud condensation nuclei and ice-nucleating particle concentration profiles, EGUsphere, 1–22, https://doi.org/10.5194/egusphere-2025-2666, 2025.

Heintzenberg, J.: The SAMUM-1 experiment over Southern Morocco: overview and introduction, Tellus B: Chemical and Physical Meteorology, 61, 2–11, https://doi.org/10.1111/j.1600-0889.2008.00403.x, 2009.

Hofer, J., Althausen, D., Abdullaev, S. F., Makhmudov, A. N., Nazarov, B. I., Schettler, G., Engelmann, R., Baars, H., Fomba, K. W., Mueller, K., Heinold, B., Kandler, K., and Ansmann, A.: Long-term profiling of mineral dust and pollution aerosol with multiwavelength polarization Raman lidar at the Central Asian site of Dushanbe, Tajikistan: case studies, Atmos. Chem. Phys., 17, 14559–14577, https://doi.org/10.5194/acp-17-14559-2017, 2017.

Hopke et al., 2020: Hopke P.K., Dai Q., Li L., Feng Y. (2020). Global review of recent source apportionments for airborne particulate matter. Science of the Total Environment 740, 140091.

Huang, J. P., Liu, J. J., Chen, B., and Nasiri, S. L.: Detection of anthropogenic dust using CALIPSO lidar measurements, Atmospheric Chemistry and Physics, 15, 11653–11665, https://doi.org/10.5194/acp-15-11653-2015, 2015.

Järvinen, E., Kemppinen, O., Nousiainen, T., Kociok, T., Möhler, O., Leisner, T., and Schnaiter, M.: Laboratory investigations of mineral dust near-backscattering depolarization ratios, Journal of Quantitative Spectroscopy and Radiative Transfer, 178, 192–208, https://doi.org/10.1016/j.jqsrt.2016.02.003, 2016.

Jin, Y., Andersson, H., and Zhang, S.: Air pollution control policies in China: A retrospective and prospects, Int. J. Environ. Res. Pu., 13, 1219, https://doi.org/10.3390/ijerph13121219, 2016.

Klose, M., Shao, Y., Karremann, M. K., and Fink, A. H.: Sahel dust zone and synoptic background, Geophysical Research Letters, 37, https://doi.org/10.1029/2010GL042816, 2010.

Knippertz, P., Deutscher, C., Kandler, K., Mueller, T., Schulz, O., and Schuetz, L.: Dust mobilization due to density currents in the Atlas region: Observations from the Saharan Mineral Dust Experiment 2006 field campaign, J. Geophys. Res.-Atmos., 112, D21109, https://doi.org/10.1029/2007JD008774, 2007.

Koch, J. and Renno, N. O.: The role of convective plumes and vortices on the global aerosol budget, Geophys. Res. Lett., 32, L18806, https://doi.org/10.1029/2005GL023420, 2005.

Li, Z., Guo, J., Ding, A., Liao, H., Liu, J., Sun, Y., Wang, T., Xue, H., Zhang, H., and Zhu, B.: Aerosol and boundary-layer interactions and impact on air quality, National Science Review, 4, 810–833, https://doi.org/10.1093/nsr/nwx117, 2017.

Liu, H., Pan, X., Lei, S., Zhang, Y., Du, A., Yao, W., Tang, G., Wang, T., Xin, J., Li, J., Sun, Y., Cao, J., and Wang, Z.: Vertical distribution of black carbon and its mixing state in the urban boundary layer in summer, Atmos. Chem. Phys., 23, 7225–7239, https://doi.org/10.5194/acp-23-7225-2023, 2023.

Liu et al. 2024: Liu, J., Wang, X., Wu, D., Wei, H., Li, Y., and Ji, M.: Historical footprints and future projections of global dust burden from bias-corrected CMIP6 models, npj Clim Atmos Sci, 7, 1, https://doi.org/10.1038/s41612-023-00550-9, 2024.

Logothetis, S.-A., Salamalikis, V., Gkikas, A., Kazadzis, S., Amiridis, V., and Kazantzidis, A.: 15-year variability of desert dust optical depth on global and regional scales, Atmospheric Chemistry and Physics, 21, 16499–16529, https://doi.org/10.5194/acp-21-16499-2021, 2021.

Mamali, D., Marinou, E., Sciare, J., Pikridas, M., Kokkalis, P., Kottas, M., Binietoglou, I., Tsekeri, A., Keleshis, C., Engelmann, R., Baars, H., Ansmann, A., Amiridis, V., Russchenberg, H., and Biskos, G.: Vertical profiles of aerosol mass concentration derived by unmanned airborne in situ and remote sensing instruments during dust events, Atmospheric Measurement Techniques, 11, 2897–2910, https://doi.org/10.5194/amt-11-2897-2018, 2018.

Mamouri, R. E. and Ansmann, A.: Fine and coarse dust separation with polarization lidar, Atmospheric Measurement Techniques, 7, 3717–3735, https://doi.org/10.5194/amt-7-3717-2014, 2014.

Mamouri, R. E. and Ansmann, A.: Estimated desert-dust ice nuclei profiles from polarization lidar: methodology and case studies, Atmos. Chem. Phys., 15, 3463–3477, https://doi.org/10.5194/acp-15-3463-2015, 2015.

Mamouri, R. E., Ansmann, A., Nisantzi, A., Kokkalis, P., Schwarz, A., and Hadjimitsis, D.: Low Arabian dust extinction-to-backscatter ratio, Geophys. Res. Lett., 40, 4762–4766, https://doi.org/10.1002/grl.50898, 2013.

Mamouri, R.-E. and Ansmann, A.: Potential of polarization lidar to provide profiles of CCN- and INP-relevant aerosol parameters, Atmospheric Chemistry and Physics, 16, 5905–5931, https://doi.org/10.5194/acp-16-5905-2016, 2016.

Mamouri, R.-E. and Ansmann, A.: Potential of polarization/Raman lidar to separate fine dust, coarse dust, maritime, and anthropogenic aerosol profiles, Atmos. Meas. Tech., 10, 3403–3427, https://doi.org/10.5194/amt-10-3403-2017, 2017.

Marinou, E., Amiridis, V., Binietoglou, I., Tsikerdekis, A., Solomos, S., Proestakis, E., Konsta, D., Papagiannopoulos, N., Tsekeri, A., Vlastou, G., Zanis, P., Balis, D., Wandinger, U., and Ansmann, A.: Three-dimensional evolution of Saharan dust transport towards Europe based on a 9-year EARLINET-optimized CALIPSO dataset, Atmos. Chem. Phys., 17, 5893–5919, https://doi.org/10.5194/acp-17-5893-2017, 2017.

Marinou, E., Tesche, M., Nenes, A., Ansmann, A., Schrod, J., Mamali, D., Tsekeri, A., Pikridas, M., Baars, H., Engelmann, R., Voudouri, K.-A., Solomos, S., Sciare, J., Groß, S., Ewald, F., and Amiridis, V.: Retrieval of icenucleating particle concentrations from lidar observations and comparison with UAV in situ measurements, Atmospheric Chemistry and Physics, 19, 11315–11342, https://doi.org/10.5194/acp-19-11315-2019, 2019.

Moulin, C. and Chiapello, I.: Impact of human-induced desertification on the intensification of Sahel dust emission and export over the last decades, Geophysical Research Letters, 33, https://doi.org/10.1029/2006GL025923, 2006.

Müller, D., Weinzierl, B., Petzold, A., Kandler, K., Ansmann, A., Müller, T., Tesche, M., Freudenthaler, V., Esselborn, M., Heese, B., Althausen, D., Schladitz, A., Otto, S., and Knippertz, P.: Mineral dust observed with AERONET Sun photometer, Raman lidar, and in situ instruments during SAMUM 2006: Shape-independent particle properties, Journal of Geophysical Research: Atmospheres, 115, https://doi.org/10.1029/2009JD012520, 2010.

Nisantzi, A., Mamouri, R. E., Ansmann, A., Schuster, G. L., and Hadjimitsis, D. G.: Middle East versus Saharan dust extinction-to-backscatter ratios, Atmos. Chem. Phys., 15, 7071–7084, https://doi.org/10.5194/acp-15-7071-2015, 2015.

Pappalardo, G., Amodeo, A., Apituley, A., Comeron, A., Freudenthaler, V., Linne, H., Ansmann, A., Boesenberg, J., D'Amico, G., Mattis, I., Mona, L., Wandinger, U., Amiridis, V., Alados-Arboledas, L., Nicolae, D., and Wiegner, M.: EARLINET: towards an advanced sustainable European aerosol lidar network, Atmos. Meas. Tech., 7, 2389–2409, https://doi.org/10.5194/amt-7-2389-2014, 2014.

Penner, J. E., Charlson, R. J., Hales, J. M., Laulainen, N. S., Leifer, R., Novakov, T., Ogren, J., Radke, L. F., Schwartz, S. E., and Travis, L.: Quantifying and Minimizing Uncertainty of Climate Forcing by Anthropogenic Aerosols, Bulletin of the American Meteorological Society, 75, 375–400, https://doi.org/10.1175/1520-0477(1994)075<0375:QAMUOC>2.0.CO;2, 1994.

Pey, J., Querol, X., Alastuey, A., Forastiere, F., and Stafoggia, M.: African dust outbreaks over the Mediterranean Basin during 2001–2011: PM10 concentrations, phenomenology and trends, and its relation with synoptic and mesoscale meteorology, Atmospheric Chemistry and Physics, 13, 1395–1410, https://doi.org/10.5194/acp-13-1395-2013, 2013.

Pikridas, M., Bezantakos, S., Močnik, G., Keleshis, C., Brechtel, F., Stavroulas, I., Demetriades, G., Antoniou, P., Vouterakos, P., Argyrides, M., Liakakou, E., Drinovec, L., Marinou, E., Amiridis, V., Vrekoussis, M., Mihalopoulos, N., and Sciare, J.: On-flight intercomparison of three miniature aerosol absorption sensors using unmanned aerial systems (UASs), Atmos. Meas. Tech., 12, 6425–6447, https://doi.org/10.5194/amt-12-6425-2019, 2019.

Proestakis, E., Amiridis, V., Marinou, E., Georgoulias, A. K., Solomos, S., Kazadzis, S., Chimot, J., Che, H., Alexandri, G., Binietoglou, I., Daskalopoulou, V., Kourtidis, K. A., de Leeuw, G., and van der A, R. J.: Nine-year spatial and temporal evolution of desert dust aerosols over South and East Asia as revealed by CALIOP, Atmospheric Chemistry and Physics, 18, 1337–1362, https://doi.org/10.5194/acp-18-1337-2018, 2018.

Proestakis, E., Gkikas, A., Georgiou, T., Kampouri, A., Drakaki, E., Ryder, C. L., Marenco, F., Marinou, E., and Amiridis, V.: A near-global multiyear climate data record of the fine-mode and coarse-mode components of atmospheric pure dust, Atmospheric Measurement Techniques, 17, 3625–3667, https://doi.org/10.5194/amt-17-3625-2024, 2024.

Querol, X., Pey, J., Pandolfi, M., Alastuey, A., Cusack, M., Pérez, N., Moreno, T., Viana, M., Mihalopoulos, N., Kallos, G., and Kleanthous, S.: African dust contributions to mean ambient PM10 mass-levels across the Mediterranean Basin, Atmospheric Environment, 43, 4266–4277, https://doi.org/10.1016/j.atmosenv.2009.06.013, 2009.

Querol, X., Pérez, N., Reche, C., Ealo, M., Ripoll, A., Tur, J., Pandolfi, M., Pey, J., Salvador, P., Moreno, T., and Alastuey, A.: African dust and air quality over Spain: Is it only dust that matters?, Science of The Total Environment, 686, 737–752, https://doi.org/10.1016/j.scitotenv.2019.05.349, 2019.

Ryder, C. L., Marenco, F., Brooke, J. K., Estelles, V., Cotton, R., Formenti, P., McQuaid, J. B., Price, H. C., Liu, D., Ausset, P., Rosenberg, P. D., Taylor, J. W., Choularton, T., Bower, K., Coe, H., Gallagher, M., Crosier, J., Lloyd, G., Highwood, E. J., and Murray, B. J.: Coarse-mode mineral dust size distributions, composition and optical properties from AER-D aircraft measurements over the tropical eastern Atlantic, Atmospheric Chemistry and Physics, 18, 17225–17257, https://doi.org/10.5194/acp-18-17225-2018, 2018.

Sakai, T., Nagai, T., Zaizen, Y., and Mano, Y.: Backscattering linear depolarization ratio measurements of mineral, sea-salt, and ammonium sulfate particles simulated in a laboratory chamber, Appl. Optics, 49, 4441–4449, https://doi.org/10.1364/AO.49.004441, 2010.

Samaras, S., Böckmann, C., Haarig, M., Ansmann, A., Walser, A., and Weinzierl, B.: Retrieval of microphysical dust particle properties from SALTRACE lidar observations: Case studies, Atmospheric Chemistry and Physics Discussions, 1–41, https://doi.org/10.5194/acp-2020-459, 2020.

Seibert, P., Beyrich, F., Gryning, S. E., Joffre, S., Rasmussen, A., & Tercier, P. (2000). Review and intercomparison of operational methods for the determination of the mixing height. Atmospheric Environment, 34(7), 1001–1027.

Shimizu, A., Sugimoto, N., Matsui, I., Arao, K., Uno, I., Murayama, T., Kagawa, N., Aoki, K., Uchiyama, A., and Yamazaki, A.: Continuous observations of Asian dust and other aerosols by polarization lidars in China and Japan during ACE-Asia, Journal of Geophysical Research: Atmospheres, 109, https://doi.org/10.1029/2002JD003253, 2004.

Stafoggia, M., Zauli-Sajani, S., Pey, J., Samoli, E., Alessandrini, E., Basagaña, X., Cernigliaro, A., Chiusolo, M., Demaria, M., Díaz, J., Faustini, A., Katsouyanni, K., Kelessis, A. G., Linares, C., Marchesi, S., Medina, S., Pandolfi, P., Pérez, N., Querol, X., Randi, G., Ranzi, A., Tobias, A., Forastiere, F., and the MED-PARTICLES Study Group: Desert Dust Outbreaks in Southern Europe: Contribution to Daily PM10 Concentrations and Short-Term Associations with Mortality and Hospital Admissions, Environmental Health Perspectives, 124, 413–419, https://doi.org/10.1289/ehp.1409164, 2016.

Stull, R. B.: Mean boundary layer characteristics, in: An introduction to boundary layer meteorology, pp. 1–27, Springer, 1988.

Tegen, I. and Fung, I.: Contribution to the atmospheric mineral aerosol load from land surface modification, Journal of Geophysical Research, 100, 18,707-18,726, https://doi.org/10.1029/95jd02051, 1995.

Tegen, I., Lacis, A. A., and Fung, I.: The influence on climate forcing of mineral aerosols from disturbed soils, Nature, 380, 419–422, https://doi.org/10.1038/380419a0, 1996.

Tesche, M., Ansmann, A., MüLLER, D., Althausen, D., Mattis, I., Heese, B., Freudenthaler, V., Wiegner, M., Esselborn, M., Pisani, G., and Knippertz, P.: Vertical profiling of Saharan dust with Raman lidars and airborne HSRL in southern Morocco during SAMUM, Tellus B: Chemical and Physical Meteorology, 61, 144–164, https://doi.org/10.1111/j.1600-0889.2008.00390.x, 2009a.

Tesche, M., Ansmann, A., Müller, D., Althausen, D., Engelmann, R., Freudenthaler, V., and Groß, S.: Vertically resolved separation of dust and smoke over Cape Verde using multiwavelength Raman and polarization lidars during Saharan Mineral Dust Experiment 2008, Journal of Geophysical Research: Atmospheres, 114, https://doi.org/10.1029/2009JD011862, 2009b.

Tesche, M., Wandinger, U., Ansmann, A., Althausen, D., Mueller, D., and Omar, A. H.: Ground-based validation of CALIPSO observations of dust and smoke in the Cape Verde region, J. Geophys. Res.-Atmos., 118, 2889–2902, https://doi.org/10.1002/jgrd.50248, 2013.

Tesche, M., Gross, S., Ansmann, A., Müller, D., Althausen, D., Freudenthaler, V., and Esselborn, M.: Profiling of Saharan dust and biomass-burning smoke with multiwavelength polarization Raman lidar at Cape Verde | Tellus B: Chemical and Physical Meteorology, 2022.

Turnock, S. T., Butt, E. W., Richardson, T. B., Mann, G. W., Reddington, C. L., Forster, P. M., Haywood, J., Crippa, M., Janssens-Maenhout, G., Johnson, C. E., Bellouin, N., Carslaw, K. S., and Spracklen, D. V.: The impact of European legislative and technology measures to reduce air pollutants on air quality, human health and climate, Environ. Res. Lett., 11, 024010, https://doi.org/10.1088/1748-9326/11/2/024010, 2016.

Wandinger, U., Tesche, M., Seifert, P., Ansmann, A., Müller, D., and Althausen, D.: Size matters: Influence of multiple scattering on CALIPSO light-extinction profiling in desert dust, Geophysical Research Letters, 37, https://doi.org/10.1029/2010GL042815, 2010.

Weinzierl, B., Ansmann, A., Prospero, J. M., Althausen, D., Benker, N., Chouza, F., Dollner, M., Farrell, D., Fomba, W. K., Freudenthaler, V., Gasteiger, J., Groß, S., Haarig, M., Heinold, B., Kandler, K., Kristensen, T. B., Mayol-Bracero, O. L., Müller, T., Reitebuch, O., Sauer, D., Schäfler, A., Schepanski, K., Spanu, A., Tegen, I., Toledano, C., and Walser, A.: The Saharan Aerosol Long-Range Transport and Aerosol–Cloud-Interaction Experiment: Overview and Selected Highlights, Bulletin of the American Meteorological Society, 98, 1427–1451, https://doi.org/10.1175/BAMS-D-15-00142.1, 2017.

Wellek, S.: Testing Statistical Hypotheses of Equivalence and Noninferiority, 2nd ed., Chapman and Hall/CRC, New York, 431 pp., https://doi.org/10.1201/EBK1439808184, 2010.

World Health Organization (WHO): Health and Environment: Draft road map for an enhanced global response to the adverse health effects of air pollution A69/18, Who, 7 pp., https://apps.who.int/iris/handle/10665/252673, 2016.

Zheng, B., Tong, D., Li, M., Liu, F., Hong, C., Geng, G., Li, H., Li, X., Peng, L., Qi, J., Yan, L., Zhang, Y., Zhao, H., Zheng, Y., He, K., and Zhang, Q.: Trends in China's anthropogenic emissions since 2010 as the consequence of clean air actions, Atmos. Chem. Phys., 18, 14095–14111, https://doi.org/10.5194/acp-18-14095-2018, 2018.